# FOXO1-mediated lipid metabolism maintains mammalian embryos in dormancy

Vera A. van der Weijden [1], Maximilian Stötzel[1,2], Dhanur P. Iyer[1,2], Beatrix Fauler [3], Elzbieta Gralinska[4,8], Mohammed Shahraz[5], David Meierhofer [6], Martin Vingron [4], Steffen Rulands [7,9], Theodore Alexandrov [5], Thorsten Mielke [3] & Aydan Bulut-Karslioglu [1] ✉

Mammalian developmental timing is adjustable in vivo by preserving pre-implantation embryos in a dormant state called diapause. Inhibition of the growth regulator mTOR (mTORi) pauses mouse development in vitro, yet how embryonic dormancy is maintained is not known. Here we show that mouse embryos in diapause are sustained by using lipids as primary energy source. In vitro, supplementation of embryos with the metabolite L-carnitine balances lipid consumption, puts the embryos in deeper dormancy and boosts embryo longevity. We identify FOXO1 as an essential regulator of the energy balance in dormant embryos and propose, through meta-analyses of dormant cell signatures, that it may be a common regulator of dormancy across adult tissues. Our results lift a constraint on in vitro embryo survival and suggest that lipid metabolism may be a critical metabolic transition relevant for longevity and stem cell function across tissues.

Dormancy maintains cells in a reversible non-proliferative state and is a major feature of stem cell function[1]. Although mainly explored in the context of adult stem cell activity, dormancy is also employed in early development as a means to preserve early embryos under certain circumstances[2]. Perturbation of dormancy entry and exit leads to exhaustion of stem cell pools, compromised regeneration and risks reproductive success in many mammals. Thus, it is of great importance to identify cellular mechanisms of establishment and maintenance of dormancy, as well as exit from it.

The cellular niche, signalling and metabolism are all involved in the active rewiring of cellular networks during the transition from proliferation to dormancy[3–9]. The mTOR pathway acts as a rheostat to adjust cellular growth and anabolism according to environmental and metabolic signals. mTOR is also a central regulator of dormancy across tissues, and its hyperactivation leads to stem cell exhaustion[10–12].

Previously, we showed that inhibition of mTOR (mTORi) induces a diapause-like dormant state in vitro in embryonic stem cells (ESCs) and mouse pre-implantation embryos (blastocysts)[13]. Mouse blastocysts can be paused via mTORi for up to a few weeks; however, significant embryo loss occurs over time[13]. Importantly, the inner cell mass (ICM) and trophectoderm (TE) show distinct responses to dormancy cues in vitro and in vivo, such as different levels of genomic activity and proliferation[13–15].

Although dormant cells across tissues share common characteristics such as reduced proliferation and energy conservation, how different cells respond to dormancy cues at the onset of this transition and the specific steps taken on this route are unclear. This is largely due to inaccessibility and limited numbers of dormant cells in vivo, and their tendency to spontaneously activate upon isolation from their niche[16]. In this Article, we leveraged our mTORi-mediated in vitro diapause system

[1]Stem Cell Chromatin Group, Department of Genome Regulation, Max Planck Institute for Molecular Genetics, Berlin, Germany. [2]Institute of Chemistry and Biochemistry, Department of Biology, Chemistry and Pharmacy, Freie Universität Berlin, Berlin, Germany. [3]Microscopy and Cryo-Electron Microscopy Facility, Max Planck Institute for Molecular Genetics, Berlin, Germany. [4]Department of Computational Molecular Biology, Max Planck Institute for Molecular Genetics, Berlin, Germany. [5]Structural and Computational Biology, European Molecular Biology Laboratory, Heidelberg, Germany. [6]Mass Spectrometry Facility, Max Planck Institute for Molecular Genetics, Berlin, Germany. [7]Max Planck Institute for the Physics of Complex Systems, Dresden, Germany. [8]Present address: Roche Innovation Center Zurich, Schlieren, Switzerland. [9]Present address: Arnold Sommerfeld Center for Theoretical Physics, Ludwig-Maximilians-Universität München, Munich, Germany. ✉e-mail: aydan.karslioglu@molgen.mpg.de

to gain a time-resolved understanding of the regulation of dormancy entry. By performing time-series proteomics analysis and modelling of the ESC and trophoblast stem cell (TSC) response to mTORi, we identify distinct pathways that are critical for the establishment of a reversibly paused state. We find that the metabolic switch to fatty acid oxidation (FAO) is necessary for dormancy maintenance and identify FOXO1 and L-carnitine as its essential regulators. By supplementing embryo culture with the FAO metabolite L-carnitine, embryo longevity is extended up to 34 days in culture. We propose that facilitating the use of lipid reserves via L-carnitine supplementation enhances in vitro diapause by establishing a deeper dormant state and in addition point to the FOXO1/FAO axis as a regulator of dormancy in adult tissues.

## Results

### Reversible pausing of ESCs, but not TSCs, in vitro

To investigate the tissue-specific restrictions to mTORi-induced pausing, we cultured ESCs and TSCs, which are the in vitro derivatives of the pre-implantation epiblast and TE, in the mTOR inhibitor INK-128 (refs. [17–19]). As expected, ESCs significantly reduced proliferation and established a diapause-like state that retains colony morphology and marker expression of naive mouse ESCs (Extended Data Fig. 1a)[13]. Importantly, ESCs reactivate upon withdrawal of mTORi without compromising stem cell colony morphology (Extended Data Fig. 1a)[13]. In contrast, TSC colony morphology deteriorated over time and spontaneous differentiation was observed under prolonged mTORi and upon reactivation (Extended Data Fig. 1a,b). mTORi treatment reduced the levels of phospho-mTOR and its downstream targets phospho-S6 (target of mTOR complex 1) and phospho-AKT (target of mTOR complex 2) in both ESCs and TSCs, indicating its effectivity (Extended Data Fig. 1c–e). Thus, ESCs and TSCs show intrinsically different capacities to respond to mTOR inhibition and to maintain stemness under prolonged mTORi.

To isolate ESC-specific pathways that may allow their successful entry into pause and that can be used to improve TSC/TE pausing, we performed a time-resolved analysis of global ESC and TSC proteomes during the cellular transition in and out of pausing (denoted 'pause' and 'release' from here onwards) via label-free quantitative mass spectrometry (Fig. 1a and Supplementary Table 1). Principal component (PC) analysis separated the samples primarily by cell type as expected (PC1, 38.8%; Fig. 1b). TSC pause and release samples further separated into two different clusters, while ESCs did not (PC2, 5.5%; Fig. 1b). This suggests that the TSC proteome undergoes irreversible changes upon mTORi treatment, which is in line with the morphological changes and reduced protein expression of early TE markers upon release from mTORi (Extended Data Fig. 1a,b). To further investigate cell type-specific proteome dynamics, we constructed a pseudotime trajectory using the time-series proteome datasets. The pseudotime analysis also revealed distinct trajectories of ESCs and TSCs during pause and release (Fig. 1c,d). For ESCs, pseudotime (x axis) and biological time (y axis) largely correlate during pause and release (Fig. 1c), indicating progressive changes in the proteome. Furthermore, pause and release trajectories anti-correlate and cross paths, indicating reversibility of these changes. These data suggest a model where ESCs go through a defined sequence of events to establish pausing, which upon release are reverted in the opposite order of their establishment, that is, in a mirror-image fashion. The release trajectory does not fully reach the starting point after 48 h, suggesting that reversal to the original molecular state is slower than entry into pause (a complete reversal could potentially take up to 8 days if extrapolating the regression curve) (Fig. 1c). Interestingly, the pause-release samples show less variability than normal ESCs, suggesting that paused pluripotent cells may have more uniform pluripotency characteristics (Fig. 1c). On the contrary, TSCs do not show a clear pseudotime or reversible molecular behaviour (Fig. 1d). Thus, not all stem cells undergo reversible dormancy in response to

mTORi and this may be due to the inability to initiate a non-stochastic dormancy programme.

### Pathway divergence in stem cells in response to mTORi

Since ESCs reversibly pause and TSCs do not; we reasoned that pathways essential for reversible pausing can be isolated by comparing their pathway usage. Instead of comparing pathway usage at each timepoint, we computed a pathway divergence score by following pathway expression (that is, mean protein expression in each pathway) over time in each cell type (Fig. 1e,f and Extended Data Fig. 2a,b).

A divergence score cutoff at 0.0075 yielded 25 divergent pathways between ESCs and TSCs (Fig. 1e,f and Supplementary Table 2). Metabolic pathways dominate the list, among which lipid and amino acid degradation are enriched in ESCs and autophagy in TSCs (Fig. 1e). Since several other metabolic pathways show common patterns (Extended Data Fig. 2c), these results suggest that metabolism of lipids and certain amino acids that show divergent patterns could be of specific relevance for a successful dormancy period.

### Immediate and adaptive changes during pausing transition

To temporally resolve the steps of ESCs pausing, we performed a time-series analysis of individual protein expression, followed by k-means clustering to identify temporal patterns (Fig. 2a and Extended Data Fig. 3a for TSCs). We categorized the clusters as immediate (unidirectional response over time) or adaptive (changing response over time) on the basis of the temporal expression patterns and performed gene ontology (GO) analysis (Supplementary Table 3). Among immediate events we find reduced translation and ribosome biogenesis, which are direct mTOR targets, and chromatin changes (Fig. 2b,c). Regulation of cell cycle, lipid metabolism, and DNA repair are among adaptive responses (Fig. 2b). In contrast to ESCs, TSCs did not show changes in chromatin organization-, DNA repair- or lipid metabolism-related proteins despite downregulating translation and ribosome biogenesis (Fig. 2d). These data suggest that the immediate response to mTORi may be commonly realized in both cell types, yet TSCs may lack the adaptive responses to fully transition into a dormant state.

To investigate our proteome-based predictions of differential chromatin and adhesion properties of paused cells are reflected at the structural level, we performed transmission electron microscopy (TEM) of whole blastocysts at E4.5 (n = 4) or 4 days into diapause (n = 4, at equivalent day of gestation (EDG) 7.5). Pluripotent epiblast cells did show extensive chromatin reorganization in diapause, with accumulation of heterochromatin at the nuclear periphery and denser, punctuated nucleoli, which were not seen in TE cells (Fig. 2e). The nuclei of diapaused embryos appeared 'wrinkled' in TEM images, which was confirmed via immunofluorescence (IF) staining of the nuclear envelope component LAMIN B1 (Fig. 2f,g). The nuclear lamina-associated heterochromatin mark H3K9me2 was enriched in the epiblast compared with the TE in diapause, highlighting the tissue-specific chromatin reorganization (Fig. 2f,g). Quantification of H3K9me2 and LAMIN B1 intensity in single-nucleus cross-sections showed that the H3K9me2 specifically accumulates at the nuclear envelope in epiblast but not TE cells (Fig. 2h,i), corroborating the increased heterochromatinization at the nuclear periphery observed in TEM images. Paused ESCs showed similar H3K9me2 accumulation at the nuclear envelope and recapitulate the chromatin organization of the diapaused embryo (Extended Data Fig. 3b,c). We also observed stress fibres and large focal adhesion complexes in diapaused embryos, as predicted by the proteome analysis (Extended Data Fig. 3d). These results suggest that, unlike TSCs/TE, pluripotent cells successfully undertake genomic and metabolic adaptations during mTORi-induced pausing.

### Enhancing lipid usage boosts embryo survival

Our proteomic analysis points to metabolism, particularly the degradation of lipids and certain amino acids, as a means to establish an

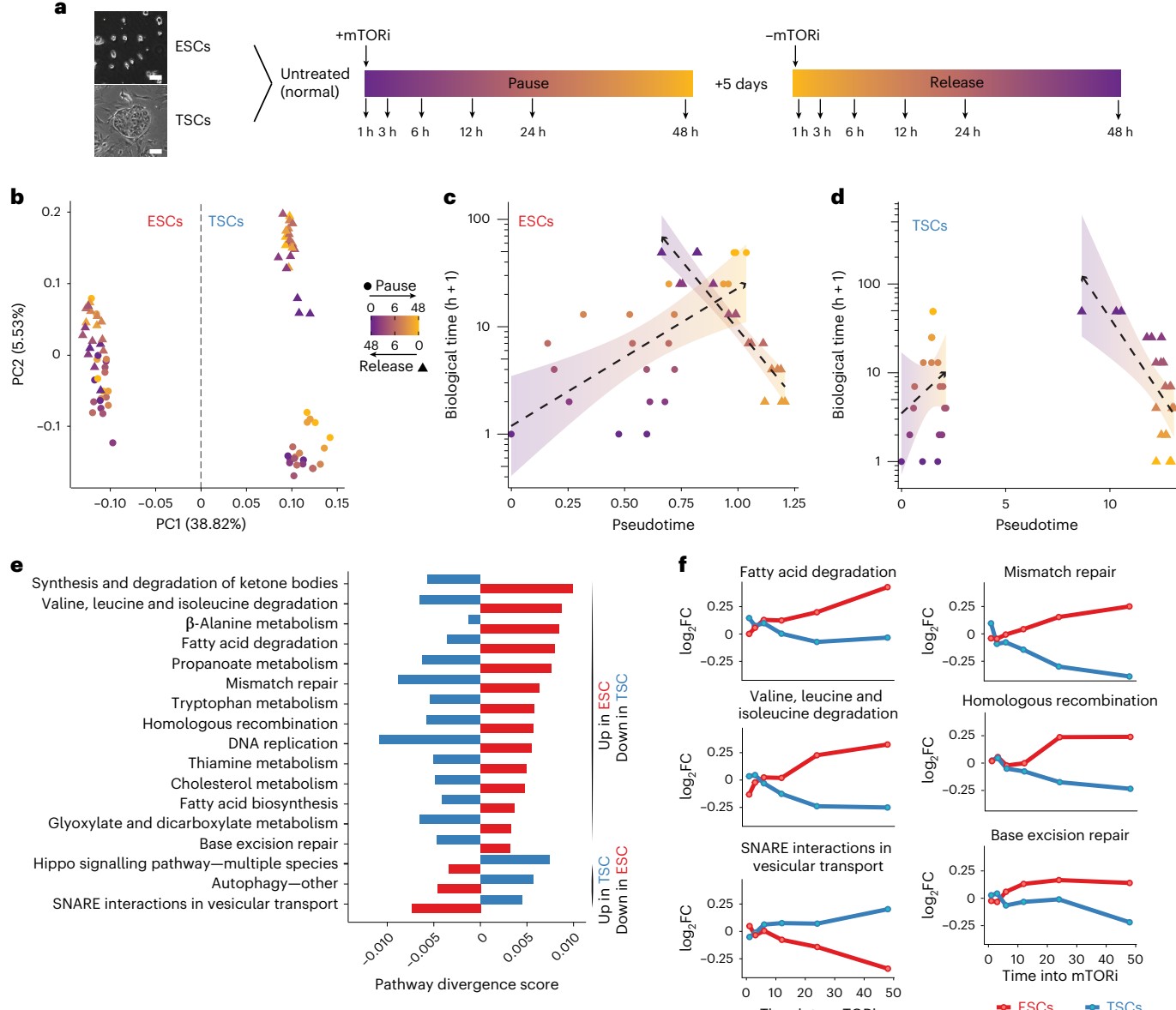

**Fig. 1 | Reversible pausing in ESCs but not in TSCs. a**, Experimental setup: ESCs and TSCs were paused via mTORi for a total of 7 days (samples collected only within the first 48 h), then released. Three biological replicates were collected at the indicated timepoints for proteomics analysis. Representative images of ESC and TSC cultures are shown. Scale bar, 100 μm. **b**, PC analysis of ESC and TSC proteomics data. The colour scale indicates the timepoint during entry into pausing or release from pausing. **c,d**, Pseudotime analysis of ESCs (**c**) or TSCs (**d**), showing the computed pseudotime against the biological time (in hours, +1 for imputation) with a regression curve in the black dotted line and 95% confidence

interval around it. The pseudotime analysis indicates reversibility of pausing of the ESC proteome. **e**, Divergent KEGG pathways during entry into pausing in ESCs and TSCs. The pathway score was computed on the basis of mean expression of pathway proteins over time (Methods). Selected pathways with a divergence score of >0.0075 are displayed (see Supplementary Table 5 for the full list). SNARE, soluble N-ethylmaleimide-sensitive factor attachment protein receptor. **f**, Line plots of log₂FC of mean protein expression (mTORi/control *t* = 0) of selected divergent pathways.

energetically sustainable paused state in ESCs (Figs. 1e,f and 2b). To first assess the metabolic status of ESCs and TSCs under mTORi, we measured glycolysis and basal respiration in real time via Seahorse extracellular acidification rate and oxygen consumption rate assays (Extended Data Fig. 4a). Paused ESCs showed a precipitous decrease in the rates of glycolysis and basal respiration (Extended Data Fig. 4a). Even though TSCs showed a proportionally similar decrease, in absolute energy levels this decrease is milder, since TSCs are less energetic to begin with (Extended Data Fig. 4a). To specifically measure the rate of lipid usage, we quantified FAO using the the FAOBlue reagent in live cells (Extended Data Fig. 4a). We found the basal FAO rate of normal TSCs to be less than ESCs by an order of magnitude (Extended Data Fig. 4a). Paused ESCs showed higher

FAO levels than both normal and mTORi-treated TSCs, even though the overall FAO rate was reduced compared with normal ESCs. Inhibition of the FAO-promoting enzyme CPT1 reduced FAO in paused ESCs (but not TSCs), indicating the active use of this energy pathway (Extended Data Fig. 4a). Generally, ESCs and TSCs decreased the rate of glycolysis and basal respiration under mTORi, yet specifically ESCs seemed to make use of lipids (Extended Data Fig. 4a). To further corroborate this, we performed the Seahorse Mito Fuel Flex test (Extended Data Fig. 4b). In this test, cells are challenged with inhibitors against the three major mitochondrial fuels glucose, glutamine and fatty acids and their use of each source is measured. Paused ESCs showed a clear preference for fatty acids compared with glucose (fatty acid dependency: on average

6.5% in normal versus 47% in paused ESCs; glucose dependency: 14.5% in normal versus 14% in pause, Extended Data Fig. 4b).

To more precisely identify the metabolites consumed by paused ESCs, we next performed bulk metabolomics (Fig. 3a and Supplementary Table 4). Predominantly lipids were differentially enriched. Specifically, paused ESCs accumulated long-chain carnitine-conjugated fatty acids, ceramide and sphingomyelins, but were depleted of short-chain carnitine-conjugated fatty acids (Fig. 3a). FAO provides energy but not many building blocks for proliferation and thus may be an ideal energy pathway in dormant cells. Proteins involved in FAO were upregulated ~24 h after mTORi treatment and downregulated upon release, while glycolysis genes did not show this dynamic pattern (Fig. 3b; only significantly altered proteins are shown). Thus, FAO is probably the main energy pathway in paused pluripotent cells.

Transfer of fatty acids from the cytosol into mitochondria requires their conjugation to carnitine by CPT1 (Fig. 3c)[20]. This step is tightly regulated and acts as a bottleneck in FAO capacity. CPT1 expression increased in ESCs but not in TSCs during mTORi treatment (Fig. 3d). We hypothesized that the fitness and survival of mTORi-diapaused blastocysts may be compromised due to an energy imbalance in the different tissues of the embryo, with efficient FAO in the epiblast and inefficient or absent FAO in the TE. We reasoned that supplementing the embryos with free L-carnitine may enhance FAO in the epiblast and promote it in the TE, thus alleviating the current constraints and maintaining the paused embryos longer in culture. As L-carnitine can be generated within the cells via amino acid breakdown, we cultured embryos in reduced potassium simplex optimized medium (KSOM) devoid of amino acids (Supplementary Table 5). L-carnitine supplementation during mTORi-mediated pausing remarkably enhanced survival duration (Fig. 3e,f; mTORi-only median and maximum survival are 8 and 15 days, respectively, and mTORi + carnitine (mTORi + c) median and maximum survival are 15 and 34 days, respectively). Inhibition of CPT1 activity with etomoxir[21] cancelled the beneficial effect of L-carnitine supplementation, indicating that the benefit is through FAO (Fig. 3e).

Supplementation with carnitine-coupled short-chain fatty acids and free fatty acids in culture media was mostly toxic to embryos and did not enhance pausing, except adipoyl-L-carnitine (Extended Data Fig. 4c). As short-chain carnitines can be generated from branched-chain amino acids, and this amino acid degradation pathway is specifically enriched in paused ESCs (Fig. 1f)[22], we also tested whether branched-chain amino acid supplementation enhances pausing (Fig. 3e). Isoleucine and valine significantly enhanced pausing, although not to the extent of direct L-carnitine supplementation. Inhibition of fatty acid uptake by SLC27A with Grassofermata[23] or synthesis by FASN with Orlistat[24] did not alter pausing maxima, suggesting that stored cellular lipids are used in paused cells (Fig. 3c,e). Supplementation with the citric acid cycle intermediate α-ketoglutarate or vitamin C as antioxidant also did not significantly increase pausing maxima, ruling out the contribution of these factors downstream of FAO (Extended Data Fig. 4c). Taken together, L-carnitine supplementation strikingly enhances mTORi-induced in vitro diapause through FAO.

To find out how carnitine supplementation benefits embryo pausing, we first probed the links between FAO and the mTOR and AMPK pathways. Enhancing or blocking FAO, with L-carnitine supplementation or CPT1i treatment, did not alter mTOR and S6 phosphorylation compared with mTORi-only embryos, indicating that FAO is downstream of mTOR activity (Extended Data Fig. 4d,e). Phospho-AKT slightly increased in L-carnitine-supplemented embryos compared with mTORi only, which suggests a feedback between lipid metabolism and mTOR complex 2 activity (Extended Data Fig. 4f). Phosphorylation of AMPK, which is a stress sensor that is upregulated in response to starvation[25], was significantly reduced in the TE of carnitine-supplemented embryos (Extended Data Fig. 4g). Although pAMPK is also reduced in CPT1i embryos, these do not survive more than 13 days in culture (Fig. 3e). The enzyme ACC converts acetyl-coA to malonyl-coA, which has been shown to inhibit CPT1 (ref. 26). ACC phosphorylation, which inhibits this conversion, was higher in mTORi embryos compared with E4.5, thus mTORi may promote CPT1 activity by lifting its inhibition by malonyl-coA. Overall, carnitine supplementation prolongs the duration of in vitro diapause probably downstream of mTOR inhibition, possibly in feedback with AKT, AMPK and ACC.

## Balanced lipid usage in carnitine-supplemented embryos

To analyse the tissue-specific distribution and usage of lipids in in vivo- and in vitro-diapaused embryos, we next investigated the abundance and distribution of lipid droplets (LDs) via electron microscopy and fluorescence staining. Electron microscopy was performed on single sections along the mural–polar axis of four individual blastocysts each for E4.5 and in vivo diapaused embryos (for 4 days, EDG7.5) (Fig. 4a and Extended Data Fig. 5a). A striking difference was observed in LD usage in the epiblast and TE of embryos in diapause (Fig. 4a). While the epiblast and TE contained several small LDs in normal E4.5 embryos, each TE cell of diapaused embryos in most cases had a single, large LD that almost spanned the width of the cell (Fig. 4a). In contrast to TE, epiblast LDs became slightly smaller in diapause. Through manual annotations of LDs and mitochondria, we find that the total number of LDs per embryo decreased in diapause, while LD size slightly increased (Fig. 4b and Extended Data Fig. 5a–c). More mitochondria were located in proximity of LDs in diapause, pointing to the usage of their lipids (Fig. 4b). Fluorescence imaging of LDs through BODIPY staining confirmed the presence of large LDs in the TE, but not the epiblast in diapause (Fig. 4c). Interestingly, large LDs were found in mural, but not polar TE, indicating a heterogeneous metabolic pattern in the TE tissue.

We next investigated LD abundance and size in carnitine-supplemented embryos in comparison with mTORi-only embryos via BODIPY staining and single-cell quantifications (Fig. 4d–f). As fixing the embryos impaired the BODIPY staining in our hands, a tissue-specific LD quantification was not possible. In general, in vitro-paused embryos, with or without carnitine, had fewer LDs than E4.5 embryos, similar to in vivo diapause (Fig. 4d). The pattern and timing of LD usage was altered with carnitine supplementation. While mTORi-only embryos only switched to consuming LDs after day 3, carnitine-supplemented embryos did so already on day 3 (Fig. 4e,f). Collectively, these data suggest that L-carnitine prolongs in vitro diapause duration probably by modulating lipid usage and balancing energy comsumption in the embryo.

---

**Fig. 2 | Temporal proteome changes during entry into pausing in ESCs. a**, Immediate and adaptive protein expression changes in ESCs. *k*-Means clustering was performed on differentially expressed proteins identified with MetaboAnalyst. **b–d**, GO analysis of upregulated (**b**) and downregulated (**c**) proteins identified in **a**. Immediately upregulated pathways pertain to chromatin organization, whereas metabolic pathways show an adaptive response. GO analysis of proteome changes in TSCs (**d**). The full list of significantly enriched GO terms is provided in Supplementary Table 3. The Benjamini–Hochberg correction was used. *P* value cutoff 0.05 and *q* value cutoff 0.1. **e**, TEM images of selected areas with one or more nuclei in the epiblast and TE of E4.5 and diapaused embryos. The nucleus is denoted with 'N' and visible as the area with an electron dense periphery, while the nucleolus is denoted with 'n'. Scale bar, 2 μm. **f,g**, IF staining of H3K9me2 and Lamin B1 in E4.5 and diapaused (EDG7.5) embryos. Projections at 20× (**f**; scale bar, 50 μm), and close-ups at 63× (**g**; scale bar, 5 μm), are shown. The ICM is indicated by the dashed line. **h,i**, Quantifications (**h**) of the H3K9me2 and Lamin B1 signal intensity along single-cell cross-sections (**i**), normalized for DAPI (*n* = 5–8 cells). ICM and TE cells of E4.5 and EDG7.5 embryos were quantified along the lines shown in **i** (scale bar, 5 μm). The signal intensity of H3K9me2, Lamin B1 and DAPI were quantified per cell, using the multichannel plot profile of the BAR plugin in Fiji-2. The cell size was scaled, and DAPI-normalized H3K9me2 and Lamin B1 intensity were plotted. Data are presented as median with confidence interval.

**A deeper dormant state in carnitine-supplemented embryos**

We have previously shown that mTORi-paused embryos reside in a near-dormant state with low nascent transcription, histone acetylation and translation[13]. Yet, the TE showed higher genomic activity than the ICM[13]. Fatty acid degradation directly contributes to the cellular quantity of acetyl-CoA, thereby providing acetyl donors for histone acetylation[27]. Quiescent cells have been associated with low or high histone acetylation in different contexts, and the histone acetyltransferase MOF

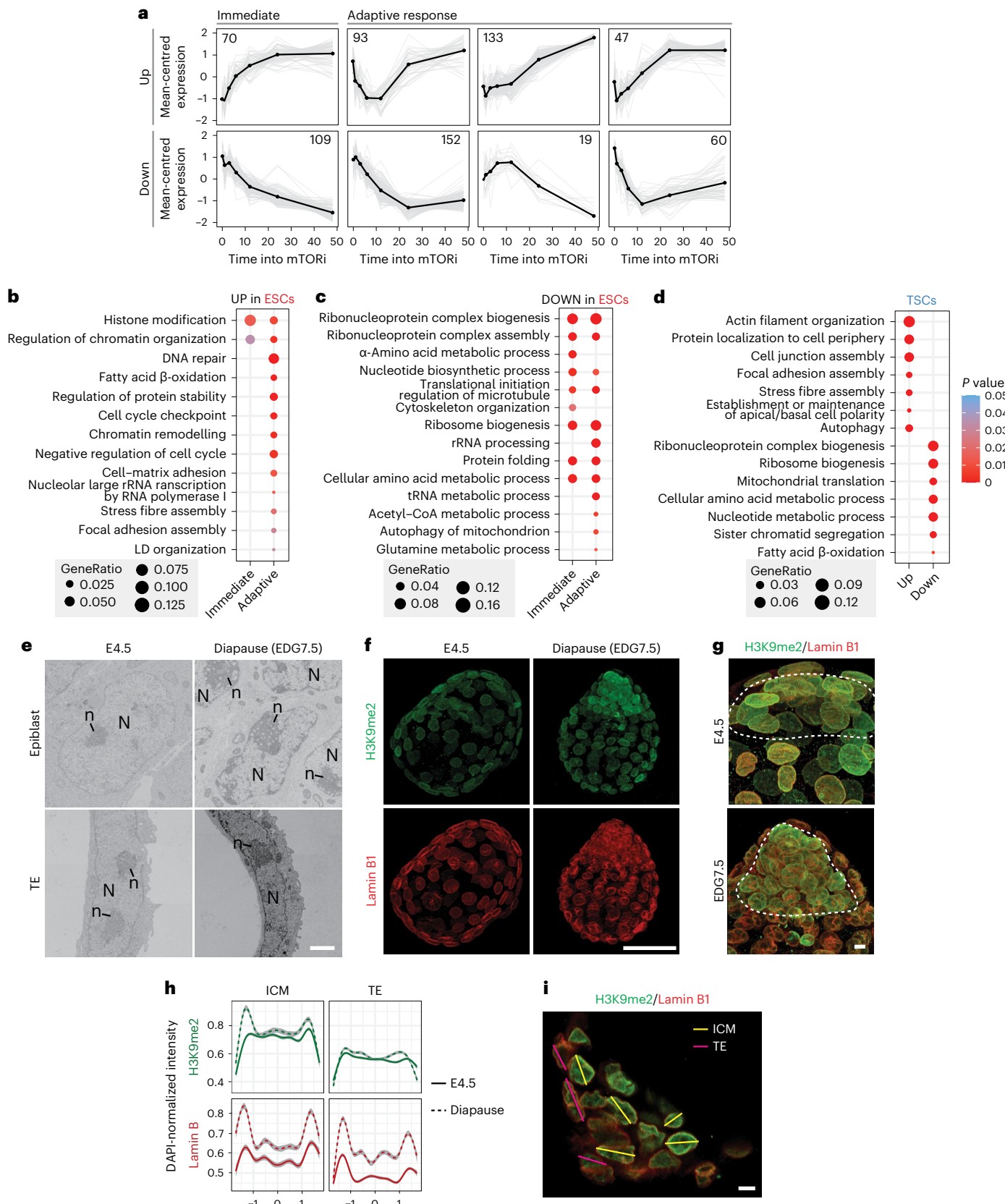

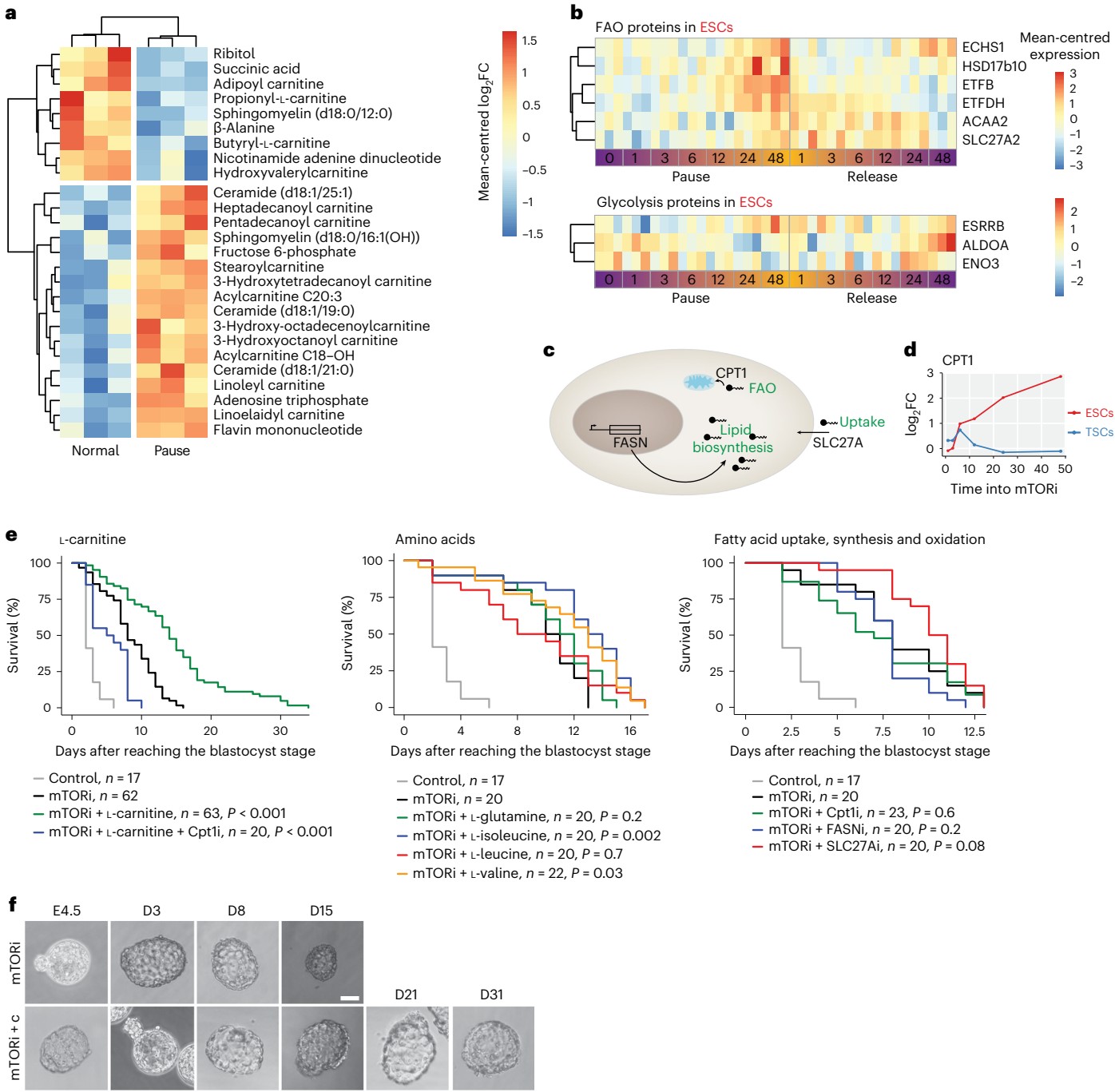

**Fig. 3 | L-carnitine supplementation enhances the duration of developmental pausing in vitro. a**, Differentially enriched metabolites detected by LC–MS/MS ($P < 0.05$ and absolute $\log_2$FC > 0.75) in paused (7 days) versus normal ESCs. Mean-centred data of three biological replicates are shown. **b**, Expression levels of differentially expressed FAO and glycolysis-related proteins in ESCs. Mean-centred protein expression of three biological replicates is shown. **c**, Schematic overview of fatty acid uptake, synthesis and oxidation, in which processes are indicated in green and respective genes in black. **d**, Increased CPT1 protein expression during mTORi in ESCs, but not TSCs. **e**, Embryo survival curves in the shown conditions. $n$ = number of embryos. Statistical test is the G-rho family test of Harrington and Fleming. **f**, Bright field images of mTORi-only and carnitine-supplemented embryos. Scale bar, 50 μm.

was directly implicated in regulating quiescence[28]. To address whether lipid usage influences global histone acetylation levels in the TE and ICM of carnitine-supplemented embryos, we stained the embryos for histone H4 lysine 16 acetylation (H4K16ac) and performed image-based quantification (Fig. 5).

Paused embryos with or without carnitine showed a global decrease in H4K16ac levels, except for a transient increase in D3 mTORi + c embryos (Fig. 5a–d). The ICM showed more dynamic changes in H4K16ac compared with the TE, with gradual reduction in over time

(Fig. 5c,d). On day 15, mTORi + c embryos showed lower H4K16ac levels than mTORi-only embryos and no staining for the proliferation marker Ki67, suggesting that the former may reside in a deeper dormant state (Fig. 5a–d and Extended Data Fig. 6a,b).

To corroborate this deeper dormant state via another measure, we analysed the metabolic status of E4.5, D8 mTORi-only and D15 mTORi + c embryos via single embryo metabolomics through matrix-assisted laser desorption/ionization (MALDI)-imaging mass spectrometry (Fig. 5e,f and Extended Data Fig. 6c). A total of 17 specific

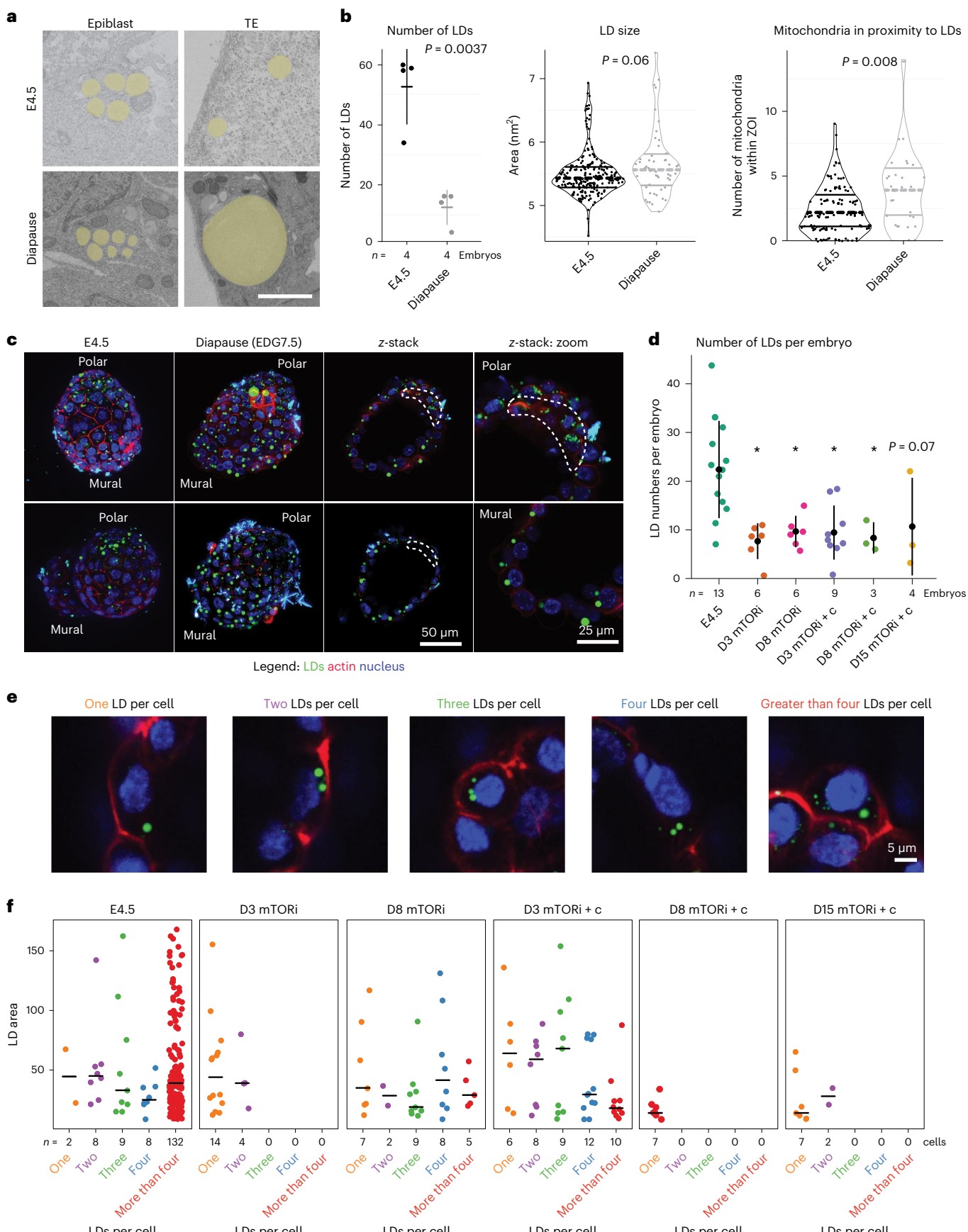

Legend: LDs actin nucleus

**Fig. 4 | L-carnitine facilitates lipid usage during developmental pausing.**
**a**, Electron microscopy images of E4.5 and diapaused embryos. LDs are
highlighted in yellow. Scale bar, 2 µm. **b**, Quantifications of LD abundance and
size and the number of mitochondria in proximity to LDs. Whole-blastocyst
electron microscopy images (single plane) from four individual E4.5 or diapaused
embryos were used. ZOI, zone of interest. Data are presented as median ±
standard deviation. Statistical test is a two-sided Student's t-test. **c**, Visualization
of LDs in E4.5 or diapaused embryos via BODIPY staining (green). Red channel
shows an actin stain (CellMask). **d**, Number of LDs per embryo in the shown

conditions. Live embryos were stained with BODIPY (LDs), CellMask (actin) and
Hoechst (DNA). n is the number of embryos per condition. *P < 0.05, one-way
ANOVA, Dunnett's post hoc test. Data are presented as median ± standard
deviation. **e**, Representative images of cells with one, two, three, four or greater
than four LDs. LDs are shown in green, cell membrane in red and the nucleus in
blue. Scale bar, 5 µm. **f**, LD area for each condition per category, that is, one, two,
three, four and greater than four LDs per cell. n is the number of cells. Horizontal
lines show median values.

metabolites were detected over the background levels. D15 mTORi + c
embryos showed reduced metabolite levels compared with E4.5 and D8
mTORi-only embryos (Fig. 5e and Extended Data Fig. 6c), supporting
the notion of a deeper dormant state.

Reversibility of dormancy is a hallmark of diapause. To test
whether mTORi + c embryos can recover from deep dormancy, we
withdrew mTORi and carnitine from the embryo culture medium and
allowed the embryos to reactivate for 12 or 24 h (denoted as +12/24 h R).
Embryos were not apoptotic, could reactivate and showed high
H4K16ac levels after 12–24 h in both the ICM and TE (Fig. 5a–d and
Extended Data Fig. 6f). Of note, D15 mTORi + c embryos took longer to
reactivate than D8 embryos (Fig. 5b–d). The number of OCT4-positive
cells, which reduced during dormancy, increased after reactivation
(Fig. 5b and Extended Data Fig. 6d). We derived several ESC and TSC
lines from carnitine-supplemented embryos (day 12) expressing
lineage-appropriate markers, further showing the reversibility of
pausing (Fig. 5g,h). Together, these data suggest a deeper dormant,
yet reversible, state in carnitine-supplemented paused embryos.

## FOXO1 is essential for carnitine-enhanced pausing

Forkhead box (FOX) family transcription factors (known as DAF16 in
*Caenorhabditis elegans*) regulate cellular metabolism, nutrient stress,
longevity and lifespan in multiple species[29–31] and have been proposed
to play a role in marsupial and killifish diapause[32,33]. As FOXO1 is a
known regulator of lipid metabolism[34,35], and is gradually upregulated
in mTORi-treated ESCs and TSCs (Extended Data Fig. 7a), we tested
whether it plays a role in the metabolic switch to lipids during in vitro
diapause. Chemical inhibition of FOXO1 with AS1842856 (ref. 36) inter-
fered with pausing in a dose-dependent manner and completely abol-
ished it at 1 µM, with or without carnitine supplementation (Fig. 6a and
Extended Data Fig. 7b, used concentrations are not cytotoxic). Therefore,
L-carnitine-mediated FAO appeared to be dependent on FOXO1 activity.

As FOXO1 is a multifunctional factor involved in the regulation
of lipid metabolism, autophagy and DNA repair, among others[37,38], we
aimed to dissect the contributions of FOXO1 and FAO to diapause. To
this end, we treated embryos with FOXOi, carnitine or CPT1i together
with mTORi and performed single embryo proteomics (Fig. 6b).
Four to five individual embryos were used in each condition. A total of
5,098 proteins were detected in single embryos with low inter-embryo
variability in each condition (Fig. 6b, Extended Data Fig. 7c and
Supplementary Table 6). All treated embryos clustered separately
from E4.5 and formed distinct groups based on condition (Fig. 6b).
Carnitine-supplemented embryos clustered together with mTORi on
PC1, while CPT1i and FOXO1i disturbed this co-clustering.

To find out the specific pathways under the control of CPT1/carnitine
and/or FOXO1, we computed Kyoto Encyclopedia of Genes and Genomes
(KEGG) pathway expression and also performed Gene Set Enrich-
ment Analysis (GSEA) (Fig. 6c,d and Supplementary Tables 7 and 8).
FOXO1 signalling, lysosome function and sphingolipid metabolism
showed dependency on both CPT1 and FOXO1, while CPT1i increased
the expression of peroxisomal genes, possibly in attempt to relocate
FAO from mitochondria to peroxisomes. FOXO1i led to expression
of the originally suppressed metabolic pathways such as tricarbox-
ylic acid cycle (oxidative phosphorylation) and pyruvate metabolism
(Fig. 6d), in accordance with earlier findings showing enhanced glyco-
lytic activity in response to FOXO1i (ref. 39).

To visualize tissue-specific lysosome abundance and autophagy
activity in embryos, we stained for LAMP1 (lysosomal membrane
protein), phospho-ULK (upstream negative regulator of autophagy)
and LC3B (component of autophagosomes). pULK was reduced in
mTORi-only and mTORi + c embryos and increased under FOXO1i, sug-
gesting that the former employ autophagy in a FOXO1-dependent man-
ner (Fig. 6e and Extended Data Fig. 7d). In line with this, we observed
distinct LC3B foci in paused embryos, which were reduced in the TE
under FOXO1i treatment (Fig. 6e and Extended Data Fig. 7d). Lysosome
abundance most significantly fluctuated between the conditions.
mTORi-only embryos accumulated lysosomes in mural TE and carnitine
supplementation, while FOXO1i reversed this trend (Fig. 6e). Lysosomes
may accumulate in mTORi-only pausing due to inefficient FAO. This is
probably alleviated in carnitine-supplemented and FOXO1i-treated
embryos via different routes; the former by enhancing lipid usage and
the latter by upregulation of glycolysis and TCA, both of which gener-
ate energy for the cells (Fig. 6d). The latter is however detrimental for
in vitro pausing as it does not allow establishing of the dormant state
(Fig. 6a). Taken together, we show that both CPT1 and FOXO1 are crucial
for the energy balance of paused embryos and the maintenance of cel-
lular dormancy during diapause.

## Shared dormancy signatures in embryonic and adult tissues

Lipid metabolism has previously been implicated in the maintenance
of tissue-specific stem cells and the balance between cellular prolifera-
tion and dormancy[8,40–43]. We wondered whether adult and embryonic
stem/progenitor cells share a regulatory basis for dormancy induction
and maintenance. For this, we analysed nine publicly available tran-
scriptomics datasets comprising six adult cell types, including muscle
stem cells (MuSCs), neuronal stem cells (NSCs), hair follicle stem cells
(HFSCs), haematopoietic stem cells (HSCs), B cells and hepatic stellate
cells (hepatic). These samples were selected from among studies that

**Fig. 5 | Carnitine supplementation reduces global genomic and metabolic**
**activity. a,b**, IF staining of H4K16ac and OCT4 in E4.5, mTORi (**a**) and mTORi + c
embryos (**b**). Embryos (n = 8–12 per condition) were paused for 3, 8 or 15 days
and or reactivated (R) for 12–24 h in fresh medium without mTORi. For each
condition, a representative z-projection and single z-stack channels are
shown. The ICM is highlighted with dashed white lines. Scale bar, 50 µm.
**c,d**, Quantification of DAPI-normalized H4K16ac intensities in the ICM (OCT4-
positive cells (**c**)) or TE (**d**) in each condition. n is the number of cells used for
quantification. A total of 8–12 embryos were used. Plots show median, *P < 0.05,
one-way ANOVA and Dunnett's post hoc test. **e**, Mean-centred log$_2$FC of

metabolites detected in single embryo MALDI imaging metabolomics analysis
(n = 4–12 embryos per condition per metabolite). **f**, An example image of the
single embryo MALDI imaging, showing the relative intensity of two exemplary
metabolites in green and purple. Scale bar, 100 µm. **g**, ESC and TSC derivation
efficiency of E3.5 and D12 + c embryos. Representative bright field images of ESC
and TSC colonies are shown on the right. Scale bar, 100 µm. **h**, Stemness marker
expression in ESCs and TSCs. Left: alkaline phosphatase (AP) activity (scale bar,
100 µm) and IF for OCT4 and CDX2 are shown (scale bar, 50 µm). Two biological
replicates were performed. Right: area-normalized intensity of OCT4 and CDX2.

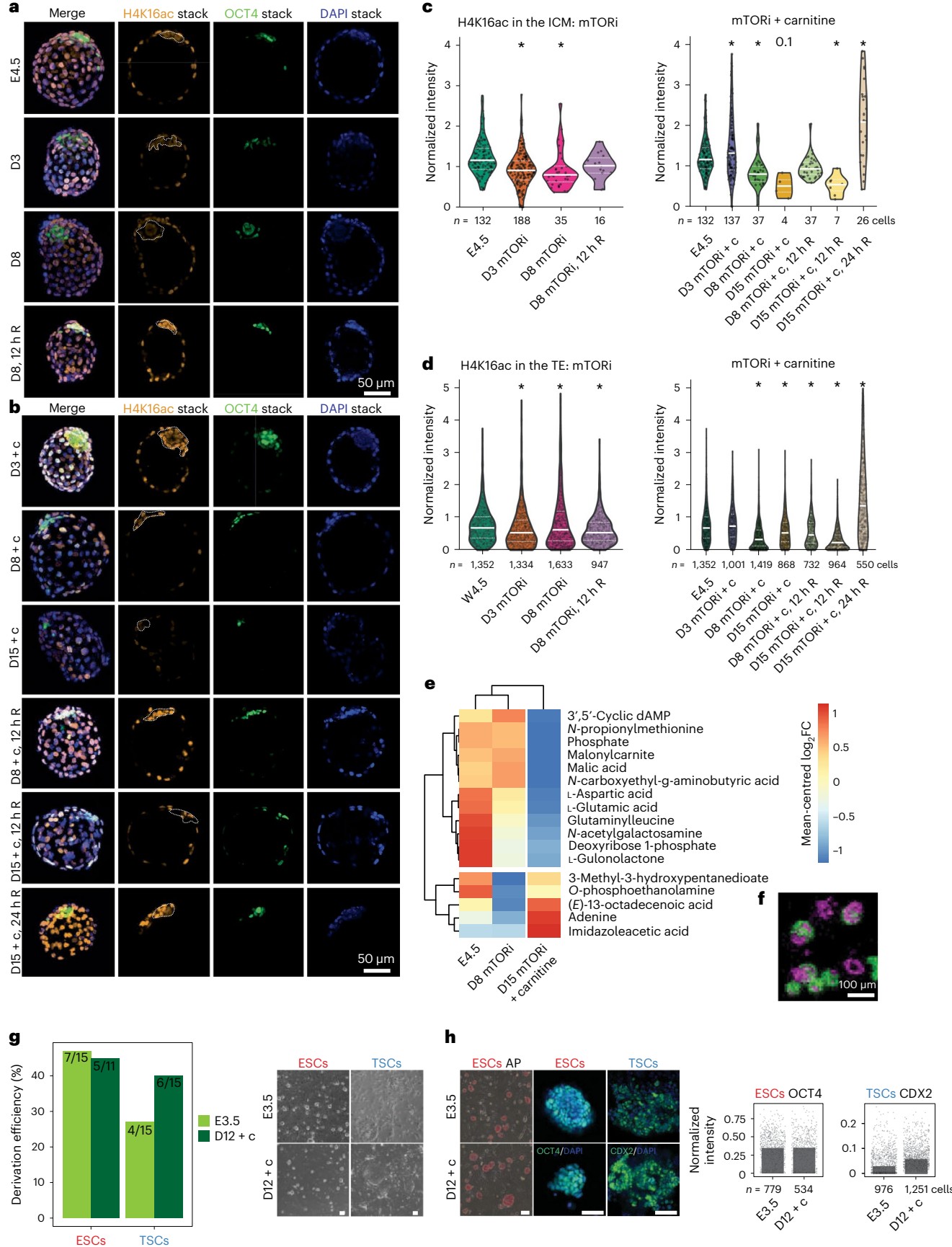

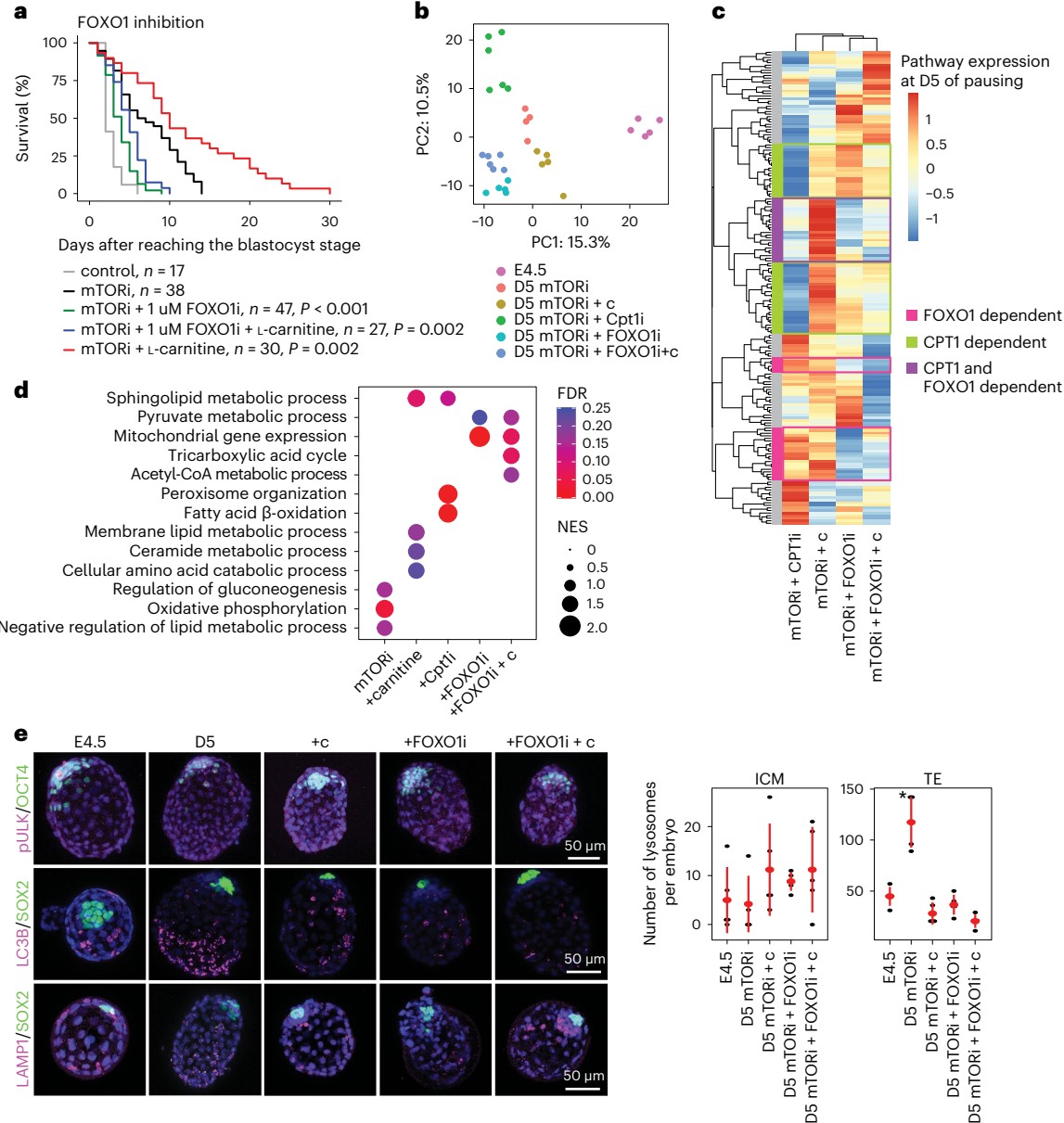

**Fig. 6 | Essential FOXO1 activity for carnitine-enhanced developmental pausing. a**, Embryo survival curves in the shown conditions. FOXO1 inhibition cancels the benefit of carnitine supplementation. *n* is the number of embryos. Statistical test is the G-rho family test of Harrington and Fleming. **b**, PC analysis of single embryo proteomics, *n* = 4–6 embryos per condition. Conditions are indicated in the different colours. See Extended Data Fig. 7c for inter-embryo variability and Supplementary Table 6 for full list of proteins. **c**, KEGG pathway protein expression profiles of D5 mTORi + c, mTORi + c + CPT1i, mTORi + FOXO1i, and mTORi + FOXO1i + c embryos compared with D5 mTORi-only embryos.

**d**, GSEA showing enriched processes in paused embryos. D5 mTORi compared with D5 mTORi + c, all other comparisons are with D5 mTORi. **e**, IF staining for pULK/OCT4, LC3B/SOX2 and LAMP1/SOX2 in each condition. Embryos (*n* = 5 per condition) were paused for 5 days with or without ʟ-carnitine, FOXO1i and FOXO1i + c. Representative embryos' *z*-projections are shown. Scale bar, 50 μm. Quantifications of number of lysosomes per embryo in the ICM and TE in the shown conditions. Data are presented as median ± standard deviation. *\*P < 0.05*, one-way ANOVA with Tukey's post hoc test.

compare quiescent versus activated cells and passed our quality control threshold of sequencing coverage, number of replicates (minimum 2) and sample variability. PC analysis largely separated the proliferating and dormant cells on PC1 (35% variance) irrespective of their tissue of origin (Fig. 7a). To identify commonalities, we retrieved the genes that were significantly upregulated (*n* = 295) or downregulated (*n* = 647) in at least four out of six cell types (Fig. 7b and Supplementary Table 9). In addition, genes that were significantly upregulated in only one of the tissues were identified. Importantly, these follow a similar expression pattern even though they pass the significance threshold only in one tissue, supporting the notion of a common expression pattern across

dormant cells. Among the commonly upregulated genes are FOXO1 and the carnitine transporter SLC22A5, in addition to several lipid metabolism and lysosomal proteins such as LIPA, LIPE, LIPN1, PSAP, ACP2, MTM1, PPT2, PITPNC1 and STARD5 (Fig. 7b). In addition, the natural mTOR inhibitors SESN1 and CASTOR1 were also found (Supplementary Table 9). At the pathway level, lipid metabolism and lysosomes were upregulated, whereas those related to ribosomes and cell division were downregulated in dormant adult stem cells (Fig. 7c). Collectively, these findings point to the importance of FOXO1 and lipid metabolism in the metabolic regulation of embryonic dormancy, with strong indication of the same in adult tissues (Fig. 7d).

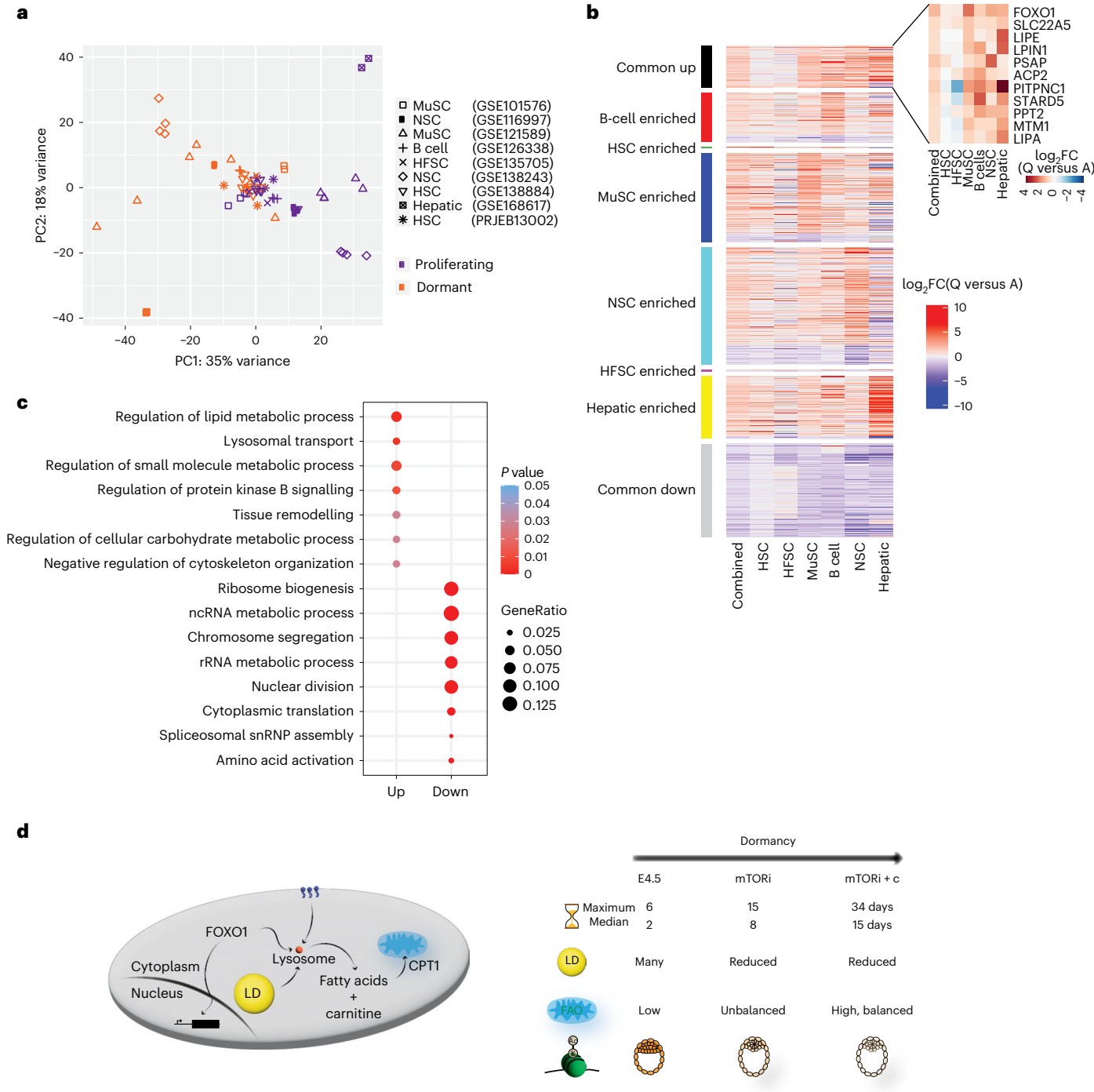

**Fig. 7 | FOXO1 and lipid metabolism genes are commonly upregulated across dormant adult cells. a**, PC analysis of nine publicly available datasets comparing gene expression signatures of quiescent/dormant versus activated cells (n = 2–7 replicates per condition per cell type). Conditions are indicated by the different colours. **b**, Heatmap displaying all genes with a P < 0.05 (as found by DESeq), subdivided into stem cell type-specific expression or common genes defined as those upregulated or downregulated in at least four out of six cell types. Zoom-in heatmap for the expression of FOXO1, SLC22A5, LIPE, LIPN1, PSAP, ACP2, PITPNC1, STARD5, PPT2, MTM1 and LIPA. **c**, GO analysis of commonly upregulated and downregulated genes in dormant stem cells compared with proliferating stem cells. The Benjamini–Hochberg correction was used. P value cutoff: 0.05, q value cutoff: 0.1. A full list of GO terms is provided in Supplementary Table 10. **d**, Model of carnitine/FOXO1-enhanced developmental pausing through balanced FAO and the establishment of a deeper dormant state.

## Discussion

In this study, we investigated the cell type-specific response to mTORi-induced in vitro diapause with the aim of identifying the temporally coordinated events during dormancy transition. We identified FOXO1-dependent lipid metabolism as a critical regulator of embryo longevity during diapause. Enhancing oxidation of lipids via carnitine supplementation prolonged embryo survival by inducing a deeper dormant state. Importantly, this state is reversible and embryos reactivate and give rise to ESCs/TSCs upon further culture. Re-transfer of these embryos into surrogate females to further corroborate the developmental competence of the carnitine-supplemented paused embryos needs to be further explored in future studies.

Morphologically, we noticed rosette-like structures in the ICM of both mTORi and mTORi + c embryos (Fig. 5b, D3 + c embryo). The epiblast of in vivo diapaused embryos is organized into a rosette-like structure around EDG9.5 (equivalent to D6 mTORi in this study), following a transient increase in WNT pathway activity[44]. Here we find that mTORi + c embryos generate rosette-like structures more efficiently and earlier than mTORi only. These results point to a direct link between lipid metabolism and epiblast morphology and suggest that in vitro-diapaused embryos with carnitine supplementation may better recapitulate the ICM morphology of in vivo diapaused embryos.

Lipid consumption has been associated with a naive-to-primed pluripotency transition[45,46]. The early onset of polarization in carnitine-supplemented embryos supports this notion. The diapaused epiblast is thought to reside in a naive pluripotent state; however, only EDG6.5 embryos were used for the transcriptomics profiling underlying this conclusion[15]. Time-series transcriptomics analysis of in vivo- and in vitro-paused embryos is likely to reveal dynamic progression of pluripotency during diapause.

Lipid reserves and FAO appear to play an essential role in regulating the balance of stem cell dormancy and proliferation in several adult tissues[8,40–43], as well as during embryonic diapause across species[33,47–49], although a contrary observation has been made in human cells[28]. Interestingly, enhanced lipid consumption due to dietary restriction increases lifespan in *Drosophila*[50] and *C. elegans*[51]. Similarly, one of the most prominent regulators of lipid metabolism, FOXO1 (DAF16 in *C. elegans*) is associated with longevity[31] and upregulated by nutrient restriction in adipocytes[52]. FOXO1, in turn, promotes lysosome function via the expression of TFEB[39] and thereby lipid metabolism, including sphingolipid metabolism[53–55]. Involvement of mTOR, FOXO1, lysosomes and FAO in both diapause and lifespan regulation points to a partially shared regulatory network between these two phenomena.

## Online content

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

## Methods

### Animal experimentation

All animal experiments were performed according to local animal welfare laws and approved by the local authority Landesamt für Gesundheit und Soziales (license numbers ZH120, G0284/18, G021/19 and G0243/18). Mice were housed in individually ventilated cages (Techniplast) on bedding (S-SELECT-09322) on a 12 h dark/light cycle and fed ad libitum (Ssniff, V1124-300).

### Cell lines and culture conditions

**Mouse ESCs.** E14 ESCs (S. Kinkley Lab, MPIMG) were used. Cells were plated on 0.1% gelatin-coated dishes and grown in Dulbecco's modified Eagle medium (DMEM) high glucose with GlutaMAX medium (Thermo Fisher, 31966047) supplemented with 15% foetal bovine serum (FBS; Thermo Fisher, 2206648RP), 1× non-essential amino acids (Thermo Fisher, 11140-035), 1× β-mercaptoethanol (Thermo Fisher, 21985023), 1× penicillin/streptomycin (Life Technologies, 15140148) and 1,000 U ml$^{-1}$ LIF (homemade) and grown at 37 °C in 20% $O_2$ and 5% $CO_2$.

**Mouse TSCs.** TSCs (M. Zernicka-Goetz Lab, Caltech) were grown on mitotically inactivated mouse embryonic fibroblasts in media containing Roswell Park Memorial Institute 1640 medium with GlutaMAX (Gibco 61870010, Thermo), 20% FBS, 1× β-mercaptoethanol (Thermo Fisher, 21985023), 1× penicillin/streptomycin (Life Technologies, 15140148), 1× sodium pyruvate, 25 ng μl$^{-1}$ FGF4 (R&D Systems, 235-F4-025) and 1 μg ml$^{-1}$ heparin (Sigma, H3149). Cells were depleted off feeders at the time of collection for analysis. Cell lines have not been authenticated but have been sextyped. Cells tested negative for mycoplasma.

### Developmental pausing setup

**Mouse ESC/TSC pausing.** Cells were treated with the mTOR inhibitor INK-128 (MedChem Express, HY-13328) at 200 nM final concentration. Medium was replenished daily.

**In vitro fertilization.** The 8–12-week-old b6d2F1 mice (Janvier Labs or produced in-house at the MPIMG) were superovulated by an intraperitonial injection with pregnant mare serum gonadotrophin (5 IU per 100 μl) on day 0 of the superovulation protocol. On day 2, mice were intraperitonially injection with human chorionic gonadotropin (5 IU per 100 μl). On day 3, mice were killed with $CO_2$ and cervical dislocation, after which the cumulus oocyte complexes were retrieved. Oocytes were fertilized using b6casF1 (MPIMG) sperm. After overnight culture, the obtained two-cell stage embryos were transferred to a fresh drop of KSOM (Merck, MR-107-D).

**Mouse blastocyst pausing in vitro.** Embryos as generated above were used. Starting at E3.5, embryos were cultured in reduced KSOM medium made in-house (Supplementary Table 5). A 2× homemade reduced KSOM medium was prepared and stored for up to 3 months at −80 °C. The medium was filtered through a 0.22 μm filter (Corning, 431118). A 1× reduced KSOM medium was freshly prepared from the 2× frozen stock before each experiment with the addition of $CaCl_2$•$2H_2O$ and bovine serum albumin (BSA) fraction V. The reduced KSOM contains a total nutrient content of 10.4 mM, compared with 13.5 mM in commercially available KSOM[56]. Embryos were cultured in four-well dishes (Nunc IVF multidish, Thermo Fisher, 144444) in a volume of 500 μl reduced KSOM under hypoxic conditions (5% $O_2$ and 5% $CO_2$ at 37°C). Embryo survival was evaluated daily under a stereomicroscopy and collapsed embryos were removed.

**Embryo supplements.** Supplements are listed in Supplementary Table 5. The supplement solvent concentration did not exceed 2%.

**Survival analysis.** Embryo survival curves were plotted in RStudio (version 1.3.1093 with R version 3.6.3) with the survminer (version 0.4.9) package. The survival (version 3.3-1) package was used to extract median and maximum survival values.

**In vivo diapause induction.** In vivo diapause was induced as previously described[57] after natural mating of CD1 mice. Briefly, pregnant females were ovariectomized at E2.5 and afterwards injected every other day with 3 mg medroxyprogesterone 17-acetate (subcutaneously). Diapaused blastocysts were flushed from uteri in M2 media after 4 days of diapause at EDG7.5.

### Global proteomics

**Sample preparation.** Proteomics sample preparation was done following a published protocol with minor modifications[58]. In brief, $5 \times 10^6$ cells were lysed under denaturing conditions in 500 μl of buffer containing 3 M guanidinium chloride (GdmCl), 10 mM tris(2-carboxyethyl) phosphine, 40 mM chloroacetamide and 100 mM Tris−HCl pH 8.5. Lysates were denatured at 95 °C for 10 min by shaking at 1,000 rpm and sonicated for 10 min. A total of 100 μl lysate was diluted with dilution buffer (10% acetonitrile and 25 mM Tris−HCl, pH 8.0, to reach a 1 M GdmCl concentration). Proteins were digested with LysC (Roche; enzyme-to-protein ratio 1:50, mass spectroscopy (MS) grade) shaking at 700 rpm at 37 °C for 2 h. The digestion mixture was diluted again to reach 0.5 M GdmCl, followed by a tryptic digestion (Roche, enzyme-to-protein ratio 1:50, MS grade) and incubation at 37 °C overnight in at 700 rpm. Peptide desalting was performed according to the manufacturer's instructions (Pierce C18 Tips, Thermo Fisher Scientific). Desalted peptides were reconstituted in 0.1% formic acid in water and separated into four fractions by strong cation exchange chromatography (SCX 3 M purification). Eluates were dried in a SpeedVac, dissolved in 5% acetonitrile and 2% formic acid in water, briefly vortexed and sonicated for 30 s before injection to nano liquid chromatography–tandem mass spectometry (LC−MS/MS).

**Run parameters.** LC−MS/MS was carried out by nano-flow reverse phase LC (Dionex Ultimate 3000, Thermo Fisher) coupled online to a Q Exactive HF Orbitrap mass spectrometer (Thermo Fisher), as reported previously[59]. LC separation was performed using a PicoFrit analytical column (75 μm inner diameter (ID) × 50 cm long, 15 μm tip ID; New Objectives) packed in-house with 3-μm C18 resin (Reprosil-AQ Pur, Dr. Maisch).

**Peptide analysis.** Raw MS data were processed with MaxQuant software (v1.6.10.43) and searched against the mouse proteome database UniProtKB with 55,153 entries, released in August 2019. The MaxQuant processed output files can be found in Supplementary Table 1.

**Global proteomics analysis.** The DEP package (version 1.14.0) was used for the global proteomics analysis[60]. Potential contaminants were filtered, unique gene names were generated and only proteins that were quantified in two out of three replicates were included for further analysis. Data were normalized, and missing values were imputed using random draws from a Gaussian distribution centred around a minimal value.

**Pseudotime analysis.** Unique peptide abundances were averaged across replicates. The 500 most variable proteins were selected for downstream analysis. $Z$-score was calculated and used for diffusion maps via the destiny package version 3.8.0 in R version 4.1.0 with parameter n_pcs = 10. Diffusion pseudotime was calculated using the DPT function of the destiny package.

**Pathway divergence analysis.** Differentially expressed genes were identified using the DEP package. $\log_2$ fold change (FC) and adjusted $P$ values during entry into pausing compared with 0 h were computed. KEGG[61] pathways containing at least ten genes were retained for the

divergence analysis. Pathway expression value corresponds to the mean $\log_2 FC$ of proteins in the pathway.

Divergence score was calculated as:

$$\text{Divergence score} =$$
$$m_{ESC}\left(\text{mean } \log_2 FC \text{ pathway expression } \frac{tP}{tN}\right)$$
$$-m_{TSC}\left(\text{mean } \log_2 FC \text{ pathway expression } \frac{tP}{tN}\right)$$

where tP is the pause timepoint, tN is the normal (untreated) timepoint and $m$ denotes the slope of pathway expression over time, computed with linear regression.

Pathways with a divergence score of >0.0075 were considered divergent. Waterfall plots were generated using ggplot2 (version 3.3.5).

### Time-series differential expression analysis

Potential contaminants were removed, and label-free quantitation values were $\log_2$ transformed in Perseus version 1.6.14.0. For each cell type, rows were filtered and only proteins expressed in two out of three samples of at least one timepoint were retained. Values were extracted, and a one-way repeated measures ANOVA with MetaboAnalyst 4.0 was used for time-series analysis[62]. Proteins with an adjusted $P$ value of <0.05 were regarded as differentially expressed.

**Dynamics of differentially expressed proteins.** $k$-Means clustering was used to identify the dynamic behaviour of the differentially expressed proteins. The R stats package 'stats' (version 4.1.0) was used.

**GO term analysis.** The clusterProfiler R package (version 4.0.5) and the Benjamini–Hochberg correction was used, with $P$ value cutoff of 0.05 and $q$ value cutoff of 0.1.

### Metabolomics

A total of $1 \times 10^8$ ESCs were collected in three biological replicates. Metabolite extraction and LC–MS/MS measurements were done as previously reported by us[63,64]. Methyl-*tert*-butyl ester (Sigma-Aldrich), methanol, ammonium acetate and water were used for metabolite extraction. Subsequent separation was performed on an LC instrument (1290 series UHPLC, Agilent), coupled to a triple quadrupole hybrid ion trap mass spectrometer QTrap 6500 (Sciex), as reported previously[65].

**Data analysis.** The metabolite identification was based on the correct retention time, up to three multiple reaction monitorings and a matching multiple reaction monitoring ion ratio of tuned pure metabolites as a reference. Relative quantification was performed using Multi-QuantTM software v.2.1.1 (Sciex), and all peaks were reviewed manually. Only the average peak area of the transition was used for calculations. Normalization was based on cell number of the samples and subsequently by internal standards. Metabolites with a $P < 0.05$ and absolute $\log_2 FC > 0.75$ were regarded as statistically significant.

**Embryo metabolomics sample prep.** To distinguish the different experimental conditions, embryos were fluorescently labelled with MitoSpy Green (E4.5), MitoSpyRed (D8 mTORi) and 4′,6-diamidino-2-phenylindole (DAPI; D15 mTORi + c). Embryos were washed 3× with phosphate-buffered saline (PBS), 3× with 100 mM ammonium acetate (Merck, 1.01116.1000), deposited onto microscope slides, desiccated for 1 h and stored at −80 °C. Sample slides were desiccated for 1 h in a vacuum chamber and matrix deposition was performed using a TM Sprayer (HTX Technologies) according to the protocol. Matrix for negative mode was 1,5-diaminonaphthalene (Sigma) at 7 mg ml$^{-1}$ sprayed at a constant rate of 0.05 ml min$^{-1}$ for seven passes. For positive mode, 2,5-dihydroxybenzoic acid (Sigma) at 15 mg ml$^{-1}$ was sprayed at a constant rate of 0.07 ml min$^{-1}$ for eight passes. Mass

spectrometry imaging was performed with the AP-SMALDI5-Orbitrap MS. The mass range was 100–500 $m/z$ for negative mode and 250–1200 $m/z$ for positive mode. After matrix coating, samples were imaged on an AP-SMALDI5 ion source (TransMIT) coupled to a Q Exactive Plus mass spectrometer (Thermo Fisher). Samples were imaged at 15 μM step size with attenuator level 28. Typical acquisition time was 1–2 h per sample, and the area covered was ~50–100 px$^2$. Raw data were analysed by converting to imzML and IBD format using Thermo Fisher ImageQuest software with centroiding. Results were annotated and interpretated using metaspace (https://metaspace2020.eu/).

**Embryo metabolomics data analysis.** A total of 26 ions with a false discovery rate <10% and embryonic localization were detected. Metabolite ion intensity was represented as a grey value in grey-scale images. To quantify each metabolite across the full embryo respectively, $z$-projections of all metabolites per spot were created and the embryos masked manually in ImageJ. The resulting masks were used to quantify the average intensity for each metabolite. Integrated intensity was calculated as average intensity multiplied by number of pixels per embryo. The $\log_2 FC$ over E4.5 was computed as integrated intensity (sample) divided by mean E4.5 integrated intensity per metabolite, which was then $\log_2$ transformed and visualized with a heatmap using ggplot2 (version 3.3.5).

### Seahorse analysis

A total of 35,000 cells were plated onto gelatin-coated cell culture mini-plates and incubated overnight at 37 °C and 5% $CO_2$. Seahorse analysis was performed following the manufacturer's recommendations for Mitostress, Glycostress and Mito Fuel Flex kits. An Agilent Seahorse XFp Extracellular Flux Analyzer was used for measurements.

**Data analysis.** Basal respiration and glycolysis rates, and glucose and fatty acid dependency values, were computed using the Seahorse 2019 software. Three technical and two biological replicates were performed. Data were plotted with ggplot2.

### FAO rate assay

Cells were treated with 50 mM etomoxir and/or 200 nM INK128 for 48 h as described in Results. Cells were trypsinized and plated on Matrigel (Sigma-Aldrich, CLS356237) pre-coated Ibidi μ-Slide 8 Well slides (Ibidi, 80821). After 30 min of attachment, cells were stained for 1 h with SPY650-DNA (SpiroChrome, SC501) in culture medium with inhibitors at 37 °C, washed twice with HEPES-buffered saline (Sigma-Aldrich, 40010) and stained for 2 h with 20 μM FAOBlue (Diag-noCine, FNK-FDV-0033) in DMEM with inhibitors at 37 °C in 20% $O_2$ and 5% $CO_2$. Cells were washed once with HEPES-buffered saline and imaged with 200 μl HEPES-buffered saline. Imaging was done on a Zeiss Plan-Apochromat 20×/0.8 objective on the Zeiss LSM880 Airy microscope using longitudinal-section magnetic mode. Images were processed using Fiji ImageJ2 (version 2.3.0) and quantified using Cell-Profiler (version 4.2.1).

### ESC/TSC derivation

ESCs were derived from E3.5 or D12 + c embryos of b6d2F1 females (Janvier Labs or produced in-house at the MPIMG, 8–12 weeks old) and b6casF1 males (produced in-house at the MPIMG, 2–6 months old) as described previously[66,67]. In brief, embryos were seeded onto mouse embryonic fibroblasts (MEFs) and allowed to attach for 3 days in knockout DMEM (Thermo Fisher, 10829018) supplemented with 20% knockout Serum Replacement (Thermo Fisher, 10828028), GlutaMAX (1.7 mM final, Thermo Fisher Scientific, 35050038), 1× non-essential amino acids (Thermo Fisher, 11140-035), 1× β-mercaptoethanol (Thermo Fisher, 21985023), 1× penicillin/streptomycin (Life Technologies, 15140148) and 1,000 U ml$^{-1}$ LIF. Cells were grown at 37 °C in 20% $O_2$ and 5% $CO_2$. After attachment, medium was refreshed every second

day, until the attached outgrowths were disaggregated. After forming colonies, cells were passaged and expanded.

TSCs were derived following previous protocols[68]. In brief, embryos were seeded onto MEFs and allowed to attach for 3 days in Roswell Park Memorial Institute 1640 medium with GlutaMAX (Thermo Fisher, 10438026), supplemented with 20% FBS (Thermo Fisher, 2206648RP), 1× sodium pyruvate (Thermo Fisher, 11360070), 1× β-mercaptoethanol (Thermo Fisher, 21985023) and 1× penicillin/streptomycin (Life Technologies, 15140148). Cells were grown at 37 °C in a 20% $O_2$ and 5% $CO_2$ incubator. Medium was refreshed every second day, until the attached outgrowths were disaggregated. After colony formation, cells were passaged and expanded.

### Electron microscopy
Diapaused (EDG7.5) or E4.5 CD1 (produced from 8–12-week-old CD1 females bred with 2–6-month-old CD1 males produced in house at the MPIMG) embryos were isolated and immediately fixed in 2.5% glutaraldehyde (grade I, Sigma) in PBS for 90 min at room temperature. The fixed embryos were subsequently embedded in 2% low melting point agarose (Biozym). Agarose was cut into cubes with an edge length of ~1 mm, containing one embryo per cube. The cubes were collected in 3-ml glass vials and stored overnight at 4 °C in PBS. Afterwards samples were post-fixed in 0.5% $OsO_4$ for 2.5 h at room temperature on a specimen rotator followed by four washes in dd$H_2O$ at room temperature for 5 min each. Samples were then incubated for 1 h in 0.1% tannic acid (Science Services) dissolved in 100 mM HEPES on a rotator, followed by three washes in dd$H_2O$ at room temperature for 10 min each. Samples were incubated in 2% uranyl acetate (Sigma-Aldrich, Merck) for 90 min at room temperature on a specimen rotator. Samples were washed in dd$H_2O$ and dehydrated through a series of increasing ethanol concentrations. Samples were incubated in a 1:1 mixture of propylene oxide and absolute ethanol for 5 min at room temperature followed by 2 × 10 min incubation in pure propylene oxide (Sigma-Aldrich, Merck) and a 30-min incubation in a 1:1 mixture of propylene oxide and Spurr resin (Low Viscosity Spurr Kit, Ted Pella, Plano GmbH) at room temperature. Samples were transferred to pure Spurr resin mixture and infiltrated overnight at 4 °C. Embryos were then polymerized at 60 °C for 48 h. Ultrathin sections (70 nm) were cut using a Leica UC7 ultramicrotome equipped with a 3-mm diamond knife (Diatome) and placed on 3.05-mm formvar carbon-coated TEM copper slot grids (Plano GmbH). Sections were post-contrasted with UranyLess EM stain and lead citrate (both from Science Services). Sections were imaged fully automatically at 4,400× nominal magnification using Leginon on a Tecnai Spirit transmission electron microscope (FEI) operated at 120 kV, which was equipped with a 4k × 4k F-416 CMOS camera (TVIPS). Acquired images were then stitched to a single montage using the TrakEM2 plugin[69] implemented in Fiji[70].

**Analysis.** Semi-automated analysis was done using the ZEN 3.4 Blue software (Zeiss). Blastocysts and subcellular structures were manually masked. The ZEN 3.4 Blue inbuilt function zone of influence analysis was carried out on the masked images to determine the number of mitochondria surrounding each LD (radius of 80 pixels). In cases where LDs occurred as clusters, binary dilation with a count of 15 was used to fuse the LDs in a cluster.

### IF staining
**Cell lines.** ESCs were directly cultured on the Nunc Lab-Tek II Chamber Slide System (Thermo Fisher Scientific, 154534). TSCs were trypsinized after culture and colonies were plated on Matrigel-coated (Sigma-Aldrich, CLS356237) Nunc Lab-Tek II Chamber Slides for 30 min. Cells were fixed for 10 min in 4% paraformaldehyde, then permeabilized for 15 min in 0.2% Triton X-100 (Sigma, T8787) in PBS. Permeabilized cells were incubated in blocking buffer (0.2% Triton X-100 in PBS, 2% BSA (BSA Fraction V 7.5%, Gibco, 15260-037) and 5% goat serum

(Jackson Immunoresearch/Dianova, 017-000-121)) for 1 h. Cells were incubated overnight at 4 °C with the following primary antibodies: Lamin B1 (Abcam, 16048), H3K9me2 (Abcam, 1220), phospho-mTOR (Abcam, ab131538), pS6 (CST, 4858 S), pAKT (CST,), OCT3/4 (Santa Cruz, sc-5279) and CDX2 (Biogenex, MU392A-UC). Cells were washed and incubated with the following secondary antibodies for 1 h at room temperature: donkey anti-rabbit AF647 (Thermo Fisher, A32795) and donkey anti-mouse AF488 (Thermo Fisher, A21202). Cells were washed, embedded and imaged with the Zeiss Plan-Apochromat 20×/0.8 or 63× 1.4 NA oil objectives on the Zeiss LSM880 Airy microscope using Airy scan. Airy processing (two-dimensional, strength of one) was done with the Zen Black software. Images were processed using Fiji ImageJ2 (version 2.3.0) and quantified using CellProfiler (version 4.2.1). Nuclei were identified as primary objects using the DAPI stain. The integrated intensities were normalized to the DAPI intensity or nuclear area. The signal intensity of H3K9me2, LAMIN B1 and DAPI were quantified per cell, using the multichannel plot profile of the BAR plugin in Fiji-2. The cell size was scaled, and H3K9me2 and LAMIN B1 intensity were plotted.

**Embryos.** Embryos were fixed, permeabilized and incubated in blocking buffer as described above, then incubated overnight at 4 °C with the following primary antibodies: Oct3/4 (Santa Cruz, sc-5279), H4K16ac (Millipore, 7329), c-Caspase3 (CST, 9661S), FoxO1 (CST, 2880T), Lamin B1 (Abcam, cat#. 16048), H3 (dimethyl K9) antibody (Abcam, 1220), phospho-mTOR (Abcam, ab131538), pS6 (CST, 4858S), pAKT (CST, 4060T), pAMPK (Abcam, ab23875), pACC (CST, 11818T), pULK (CST, 14202S), LC3B (CST, 83506S), LAMP1 (CST, 99437S), CPT1A (Abcam, ab128568) and SOX2 (R&D System, AF2018). Embryos were washed and incubated with the same secondary antibodies as above and donkey anti-goat AF594 (Thermo Fisher, A11058) for 1 h at room temperature. Embryos were washed and mounted on a microscope slide with a Secure-Seal Spacer (eight wells, 9 mm diameter and 0.12 mm deep, Thermo Fisher, S24737). LDs were stained using BODIPY, together with CellMask Deep Red (Thermo Fisher, A57245) and Hoechst 33342 (Thermo Fisher, H3570) in PBS. Imaging, processing and quantifications were done as described above. H3K9me2, LAMIN B1 and DAPI were quantified using the multichannel plot profile of the BAR plugin in Fiji-2. The cell size was scaled, and intensities were normalized for DAPI.

### Alkaline phosphatase staining
ESCs were grown on MEFs and allowed to form colonies. The Vector Red Substrate Kit, with alkaline phosphatase (Vector Labs, VEC-SK-5100), was used according to the manufacturer's instructions.

**Single embryo proteomics experimental setup.** Embryos were cultured in reduced KSOM, paused for 5 days via mTOR inhibition with 200 nM RapaLink-1 and supplemented with 1 mM L-carnitine, or treated with 50 μM etomoxir or 25 mM AS1842856 together with 1 mM L-carnitine.

**Sample preparation.** Mouse embryos (consisting of approximately 100–120 cells each) were isolated and transferred individually in low protein binding 96-well plates containing 5 μl lysis buffer (PreOmics). The plate was incubated at 95 °C for 2 min and centrifuged at 100$g$ for 1 min. Lysed proteins were digested with 5 ng trypsin (1 ng μl$^{-1}$ in ammonium bicarbonate, pH 8, Roche) at 37 °C for 3 h. Samples were acidified by adding 15 μl 1% formic acid in water and then put on a rocking platform at 300 rpm for 1 min to mix and centrifuged at 100$g$ for another minute.

**Run parameters.** The digests were loaded onto Evotip Pure (Evosep) tips according to the manufacturer's protocol. Peptide separation was carried out by nano-flow reverse phase LC (Evosep One, Evosep) using the Aurora Elite column (15 cm × 75 μm ID, C18 1.7 μm beads, IonOpticks) with the 40 samples a day method (Whisper_100_40SPD). The

LC system was online coupled to a timsTOF SCP mass spectrometer (Bruker Daltonics) applying the data-independent acquisition with parallel accumulation serial fragmentation method.

**Peptide analysis.** MS data were processed with Dia-NN (v1.8.1) and searched against an in silico-predicted mouse spectra library. The 'match between run' feature was used, and the mass search range was set to 400–1,000 $m/z$, with the MS1 and MS2 accuracy set to 10 and 15 ppm, respectively.

The DEP package (version 1.14.0) was used for analysis[60]. Potential contaminants were filtered, unique gene names were generated and only proteins that were quantified in all replicates of at least one condition were retained for further analysis. Samples with fewer than 3,000 identified proteins were excluded from further analysis, data were normalized and missing values were imputed as described above.

**KEGG pathway analysis.** Data were prepared and filtered as described above. $log_2FC$ and adjusted $P$ values for all conditions against E4.5 were computed. KEGG[61] pathways containing at least ten gene symbols were included in the divergence analysis. The pathway expression value was defined as the mean $log_2FC$ of proteins between any of the timepoints during entry and the 0 h control.

**Adult data RNA seq**
The following datasets were used: GSE126338 (GSM3596803, GSM3596804, GSM3596805, GSM3596806, GSM3596807, GSM3596831, GSM3596832, GSM3596833, GSM3596808 and GSM3596834 were omitted due to low mapping quality), PRJEB13002 (ERR1308085, ERR1308086, ERR1308087, ERR1308082, ERR1308083 and ERR1308084), GSE121589 (GSM4140537, GSM4140538, GSM4140539, GSM4140540, GSM4140541, GSM4140542, GSM4140556, GSM4140557, GSM4140558, GSM4140559 and GSM4140560), GSE101576 (GSM2706308, GSM2706309, GSM2706310, GSM2706311, GSM2706312, GSM2706313 and GSM2706314), GSE116997 (GSM3267160, GSM3267161, GSM3267162, GSM3267157, GSM3267158 and GSM3267159), GSE138243 (GSM4103308, GSM4103309, GSM4103310, GSM4103311, GSM4103316, GSM4103317, GSM4103318 and GSM4103319), GSE138884 (GSM4121228, GSM4121229, GSM4121230, GSM4121234, GSM4121235, GSM4121236, GSM4121237, GSM4121231, GSM4121232, GSM4121233, GSM4121238, GSM4121239, GSM4121240 and GSM4121241), GSE135705 (GSM4716555, GSM4716556, GSM4716557 and GSM4716558) and GSE168617 (GSM5151338, GSM5151339, GSM5151340 and GSM5151341).

Reads were aligned to the reference genome mm10 using STAR-2.5.3a (ref. 71) with default parameters except: --outFilterMultimapNmax 5, --outFilterMismatchNoverLmax 0.1, --chimSegmentMin 10 and --quantMode GeneCounts. Reads were counted using HTSeq 0.11.4 (ref. 72) using GRCm38.gtf file (Ensembl Release 101) and default parameters, except --nonunique all and -a 0.

Differentially expressed genes were identified using DESeq2 (ref. 73). To measure the effect of the conditions and to simultaneously control for batch differences the design parameter '~ batch + condition' was used. In addition, the six tissues were analysed separately. For tissues represented by more than one dataset, the design parameter '~ batch + condition' was used, while design parameter '~ condition' was used for tissues represented by one dataset.

The merged DESeq2 results were filtered for adjusted $P < 0.05$. Among them, the genes belonging to one of the following categories were selected: commonly upregulated genes (adjusted $P < 0.05$ and positive $log_2FC$ values in at least four out of six tissues), commonly downregulated genes (adjusted $P < 0.05$ and negative $log_2FC$ values in at least four out of six tissues) and tissue-specific genes (adjusted $P < 0.05$ only in the given tissue and not in the remaining ones). The heatmap (Fig. 7b), generated using R package 'ComplexHeatmap'[74], represents $log_2FC$ values from all DESeq2 runs for genes belonging to one of these categories.

**GO term and pathway enrichment analysis.** EnrichGO from the R package 'clusterProfiler'[75] were used with default parameters.

### Statistics and reproducibility
No statistical method was used to pre-determine sample size, but our sample sizes are similar to those reported in previous publications[13,76]. No data were excluded from the analyses. The experiments were not randomized and the investigators were not blinded to allocation during experiments and outcome assessment. Data distribution was assumed to be normal, but this was not formally tested. The precise number of biological replicates is indicated in the figure legends. A minimum of three different embryos were used for embryo stainings and electron microscopy. For image quantifications, precise numbers of quantified cells are provided in the figures or legends. Number of embryos used for survival analyses are provided in the figures or legends. Source data are provided with this study. Exact $P$ values that were not provided in the figures or legends are available in Supplementary Table 11.

### Reporting summary
Further information on research design is available in the Nature Portfolio Reporting Summary linked to this article.

### Data availability
UniProtKB and the mouse reference genome mm10 were used for peptide and read mapping. The proteomics datasets generated in this study have been deposited to the PRIDE[77] database and are available via ProteomeXchange with the identifiers PXD033750, PXD033798 and PXD041325. The metabolomics dataset has been deposited to peptideAtlas under the accession number PASS01758. Publicly available datasets used in this study are available under the accession numbers GSE126338, PRJEB13002, GSE121589, GSE101576, GSE116997, GSE138243, GSE138884, GSE135705 and GSE168617. All other data supporting the findings of this study are available from the corresponding author on reasonable request. Source data are provided with this paper.

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

## Acknowledgements

We thank members of the Bulut-Karslioglu Lab and B. Sözen for discussions and feedback, and MPIMG Transgenic Facility and Animal House, Mass Spectrometry and Microscopy Facilities for excellent technical support and discussions. We thank R. Buschow for help with electron microscopy image analysis, J. Shay, C. Mancini and B. Romberg for assistance, B. Lukaszewska-McGreal for proteome sample preparation, and R. Glauben, C. Yerinde and W. Kammerloher for guidance in Seahorse assays. This study was funded by the Swiss National Science Foundation (Early Postdoc Mobility fellowship P2EZP3_195682 and Postdoc Mobility fellowship P500PB_211046 to V.A.v.d.W.), the European Research Council Horizon 2020 (grant agreements 773089 to T.A. and 950349 to S.R.) and Horizon Europe (grant agreement 101101077 to T.A.), the Deutscher Akademischer Austauschdienst (DAAD PhD fellowship 91730547 to D.P.I.), the Max Planck Society (to M.S., B.F., E.G., D.M., T.M., S.R., M.V. and A.B.-K.), and the Alexander von Humboldt Foundation (Sofja Kovalevskaja Award to A.B.-K.).

## Author contributions

A.B.K. and V.A.v.d.W. conceived and developed the project. D.M. performed proteomics and metabolomics runs. M. Stötzel and B.F. performed electron microscopy sample preparation and imaging with input and supervision from T.M. M. Shahraz performed embryo metabolomics under the supervision of T.A. M. Stötzel performed electron microscopy and embryo metabolomics image analysis. D.P.I. performed Seahorse experiments. S.R. performed reversibility analysis. E.G. analysed adult stem cell datasets with supervision from M.V. V.A.v.d.W. performed all other experiments and analysis. A.B.K. supervised the project. V.A.v.d.W., M.S. and A.B.-K. wrote the paper with feedback from all authors.

## Funding

## Competing interests

The authors declare no competing interests.

## Additional information

**Extended data** is available for this paper at https://doi.org/10.1038/s41556-023-01325-3.

**Correspondence and requests for materials** should be addressed to Aydan Bulut-Karslioglu.

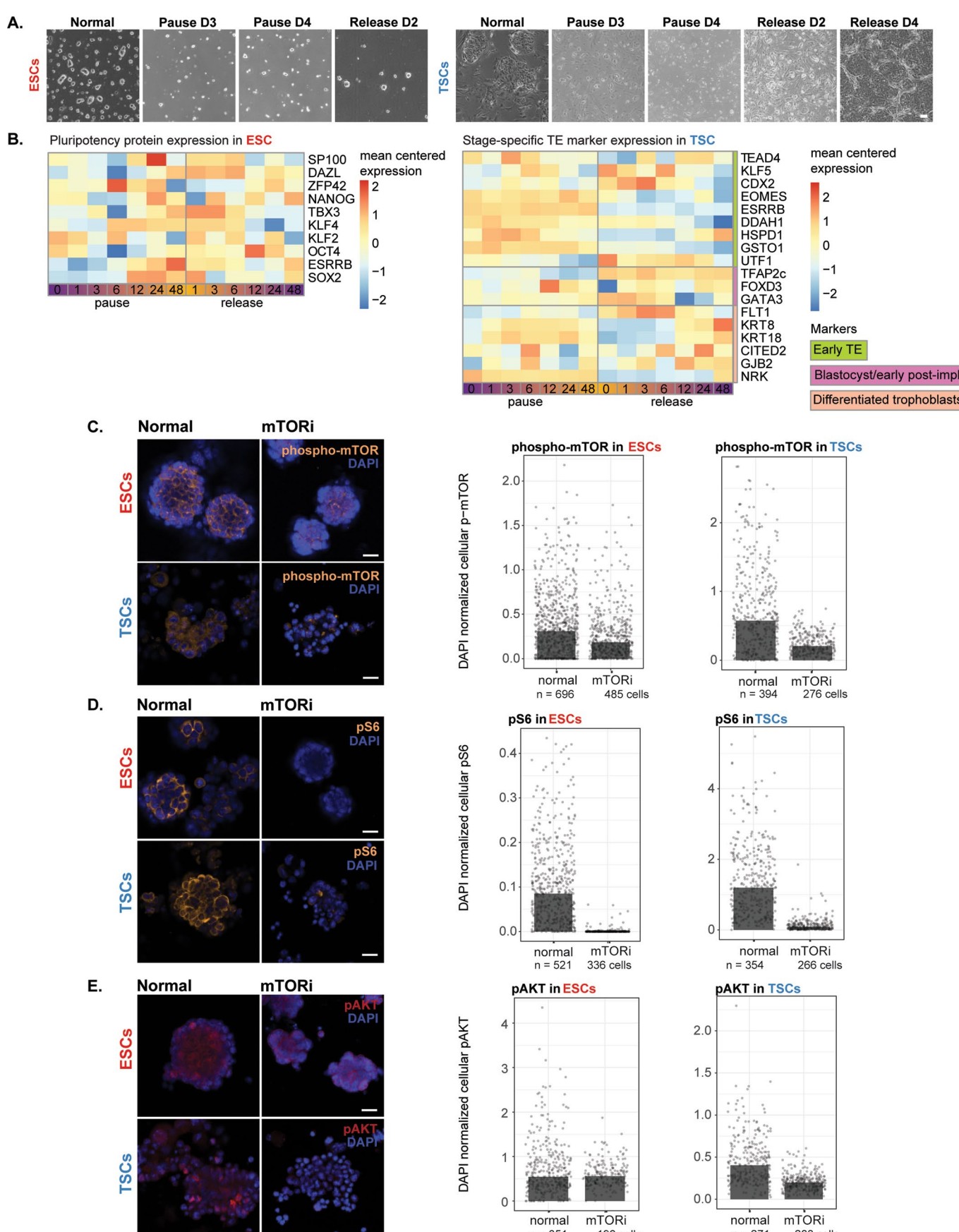

**Extended Data Fig. 1 | See next page for caption.**

**Extended Data Fig. 1 | Characterization of mTOR inhibition in ESCs and TSCs.**
**(a)** Bright field images of ESCs and TSCs during normal proliferation, at D3 and D4 of pausing and upon release from mTORi. Scale bar: 100 μm. **(b)** Mean protein expression (pause and release vs control t = 0) of pluripotency genes in ESCs and stage-specific TE marker expression in TSCs using a previously curated gene list[73]. (C-E) Levels of phospho-mTOR **(c)**, phospho-S6 **(d)**, and phospho-AKT **(e)** in normal and mTORi-treated (48 hours) ESCs and TSCs. Single-cell, DAPI-normalized quantifications are shown on right panels. n = number of cells. Two biological replicates were performed.

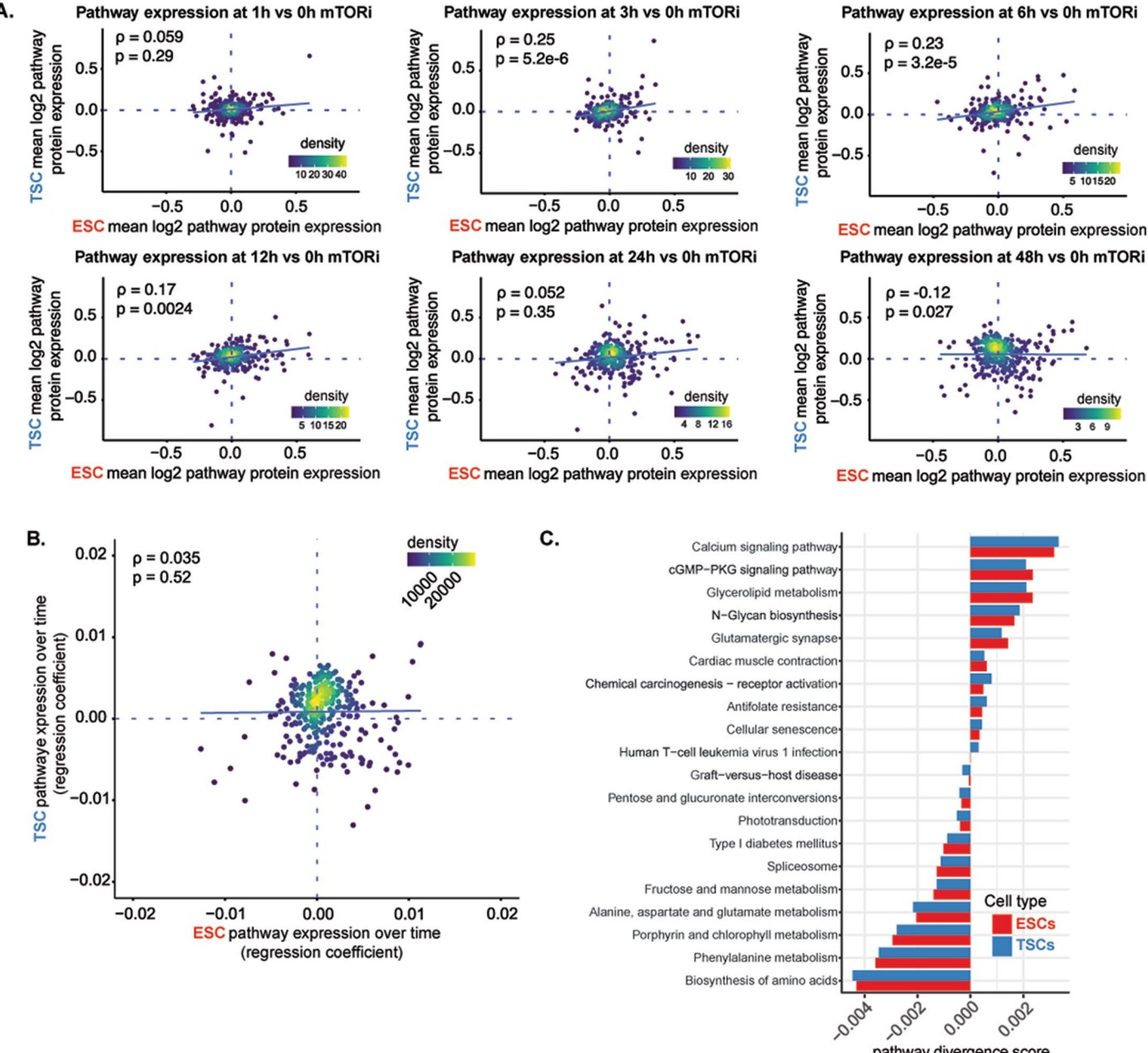

**Extended Data Fig. 2 | Pathway expression over time and commonly used pathways. (a)** Log2FC (mTORi/normal) of mean protein expression (mTORi/control t = 0) of each KEGG pathway at each time point in ESCs vs TSCs. Note that an initial concordant response can be seen for the first 12 h, which is lost afterwards. **(b)** Change in pathway expression over time in ESCs vs TSCs in response to mTORi treatment, computed by a linear regression of the mean KEGG pathway protein expression. **(c)** Concordantly altered KEGG pathways in ESCs and TSCs in response to mTORi.

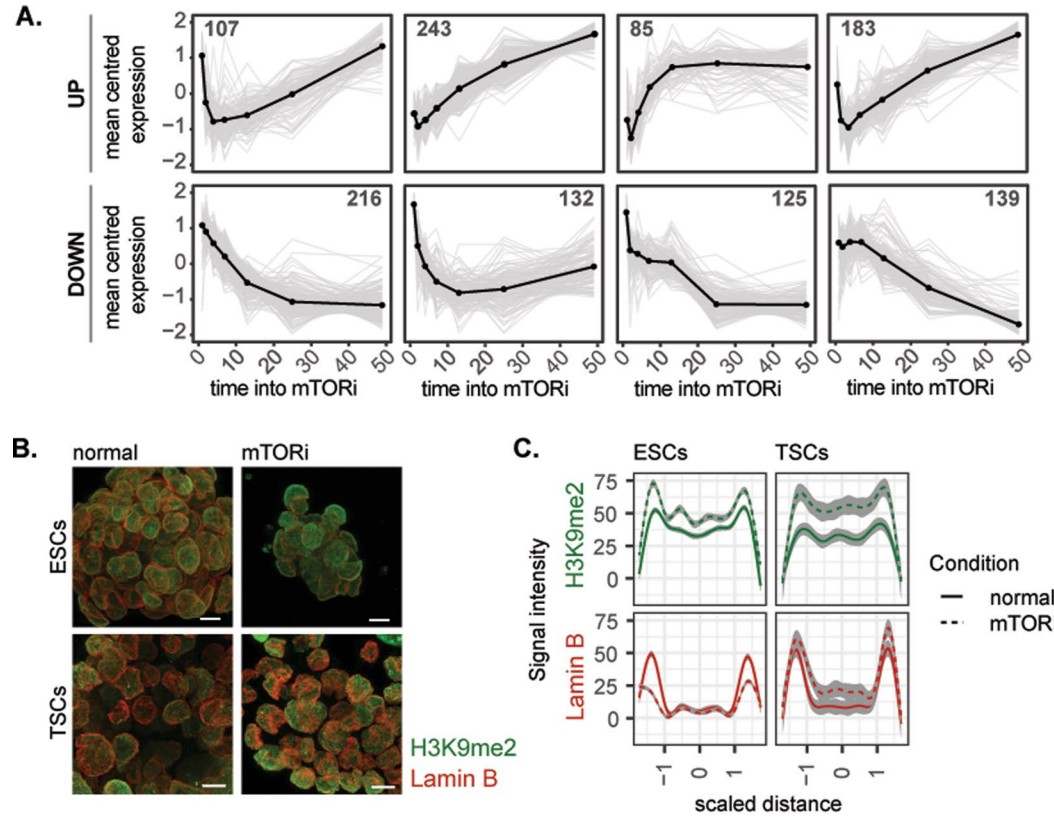

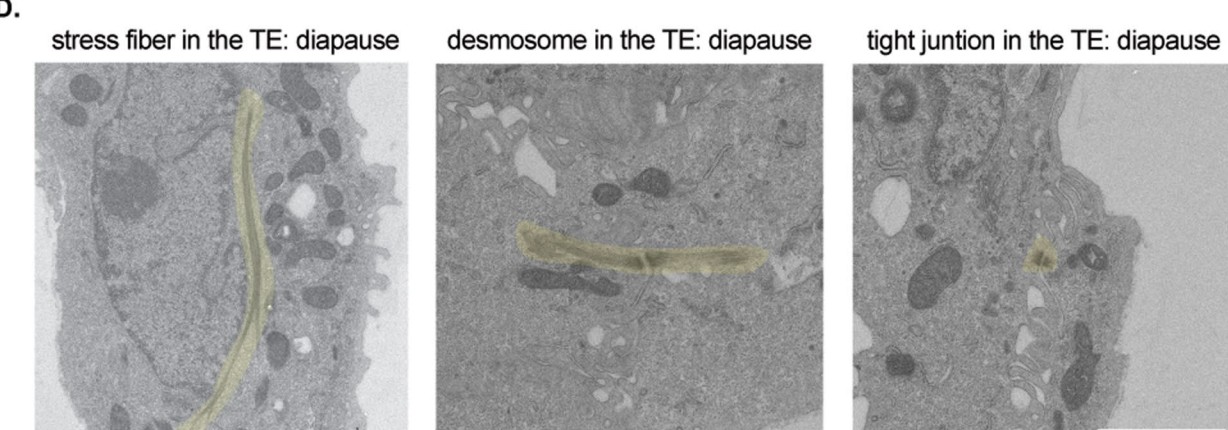

**Extended Data Fig. 3 | Altered protein expression in TSCs in response to mTORi treatment. (a)** K-means clustering of differentially expressed proteins identified with MetaboAnalyst 4.0. Spearman's rank correlation coefficient (ρ) and corresponding p value are shown in (A) and (B). **(b)** Representative images of H3K9me2 and Lamin B1 expression in normal and mTORi-treated ESCs and TSCs. Scale bar: 10 μm. **(c)** Quantifications of the H3K9me2 and Lamin B1 signal intensity along single-nucleus cross-sections of normal and mTORi-treated ESCs and TSCs (n = 15 cells, 48 h treatment). **(d)** Transmission electron microscopy images of cell-cell adhesion complexes and stress fibers. Structures are highlighted in yellow. Scale bar: 2 μm.

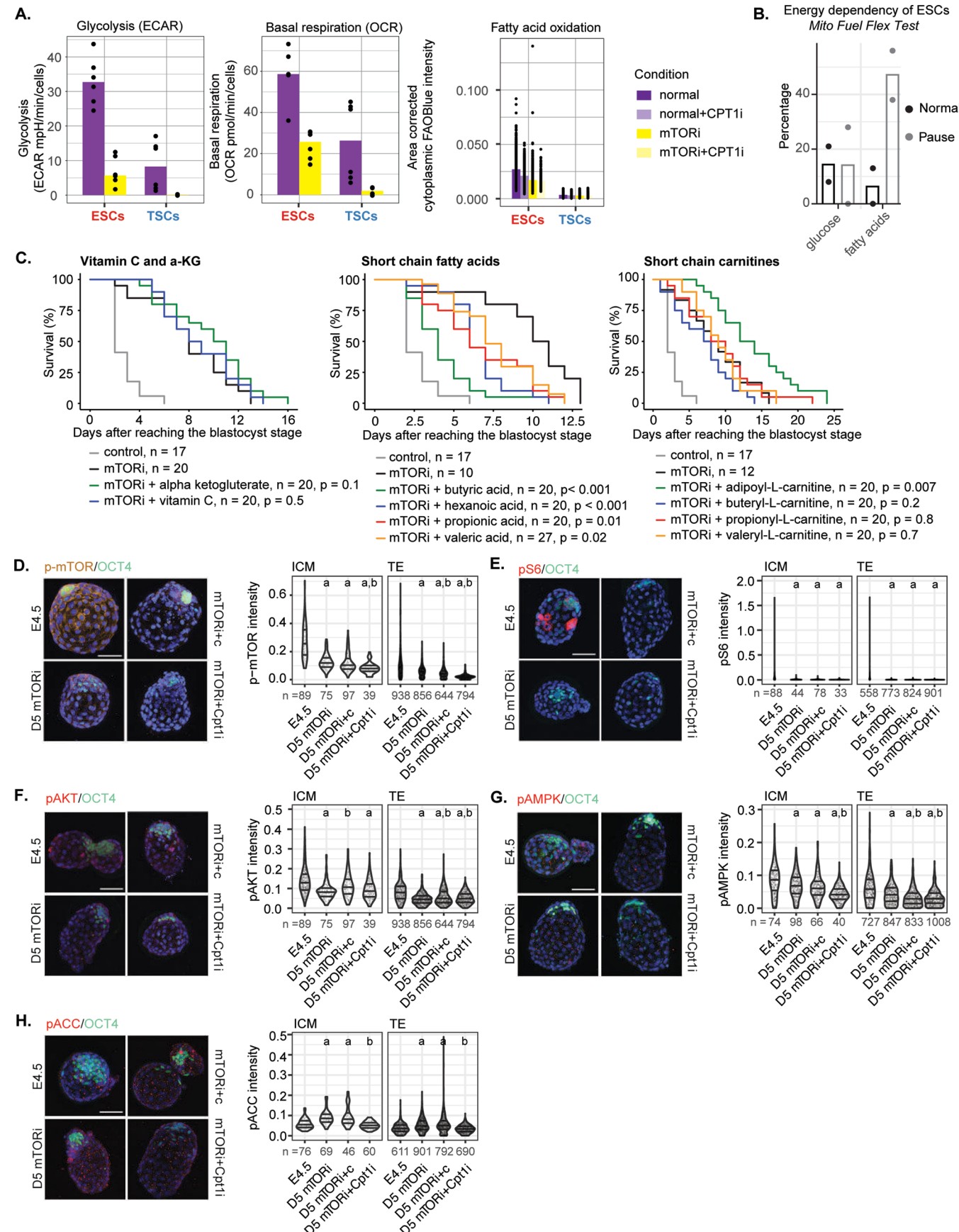

**Extended Data Fig. 4 | See next page for caption.**

**Extended Data Fig. 4 | Further characterization of stem cell and embryo metabolic profiles. (a)** Energetics of normal and mTORi-treated ESCs and TSCs. Seahorse glycolysis (ECAR) and basal respiration (OCR) assays were performed on normal or mTORi-treated cells (4 days). Three technical replicates of two biological replicates are shown. Fatty acid oxidation rate in normal and mTORi-treated (48 hours) ESCs and TSCs with or without the CPT1 inhibitor. FAO Blue reagent was used for measurement of FAO output on live cells. The area-corrected median cytoplasmic FAO Blue levels for two biological replicates are shown. **(b)** Energy preference of paused ESCs. Seahorse Mito Fuel Flex test was performed on normal and paused ESCs (4 days). Data from two biological replicates are shown. Average of two replicates are indicated with the bars.

**(c)** Survival curves or paused embryos supplemented with vitamin C, alpha-ketoglutarate, free short chain fatty acids, and short chain carnitine-conjugated fatty acids. n = number of embryos. Statistical test is the G-rho family test of Harrington and Fleming[89]. (D-H) IF staining against phospho-mTOR/OCT4 **(d)**, pS6/OCT4 **(e)**, pAKT/OCT4 **(f)**, pAMPK/OCT4 **(g)**, and pACC/OCT4 **(h)** in E4.5, D5 mTORi-only, mTORi+c, and mTORi+CPT1i embryos. Scale bar: 50 μm. Violin plots show the respective quantifications of the area-corrected cellular signal intensity. n = number of cells. a and b: $p < 0.05$ compared to E4.5 and D5 mTORi, respectively, computed with a one way ANOVA and Tukey HSD posthoc test. Minimum 4 embryos were used.

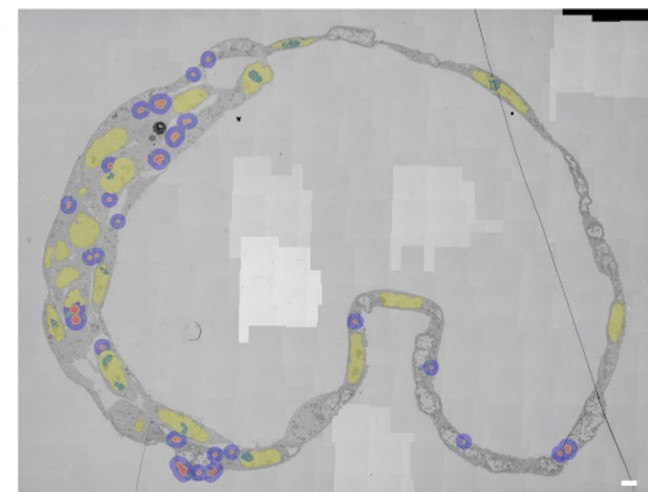

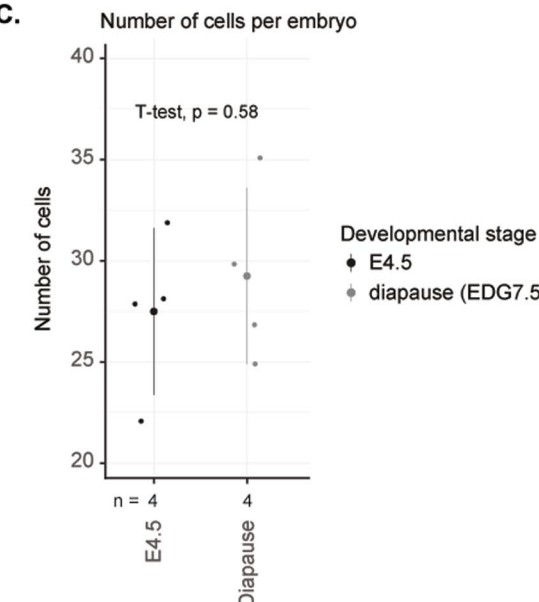

**Extended Data Fig. 5 | Detailed analysis of EM images. (a)** Example masking of the embryo (yellow), nucleus (light blue), LDs (green), and mitochondria (red). Scale bar: 2 μm. **(b)** Masking of the zone of influence to identify the number of mitochondria in close proximity to each LD or a cluster of LDs. Masked are the nucleus (yellow), the LDs (red), a watershed around the lipid droplets (orange), the zone of interest of 80 pixels around the watershed object containing the lipid droplets (purple), and the mitochondria in the zone of interest (green). Scale bar: 2 μm. **(c)** Number of analyzed cells per embryo, as quantified using the EM images. n = number of embryos per condition. Data are presented as median +/- standard deviation. Statistical test is student's T-test, two-sided.

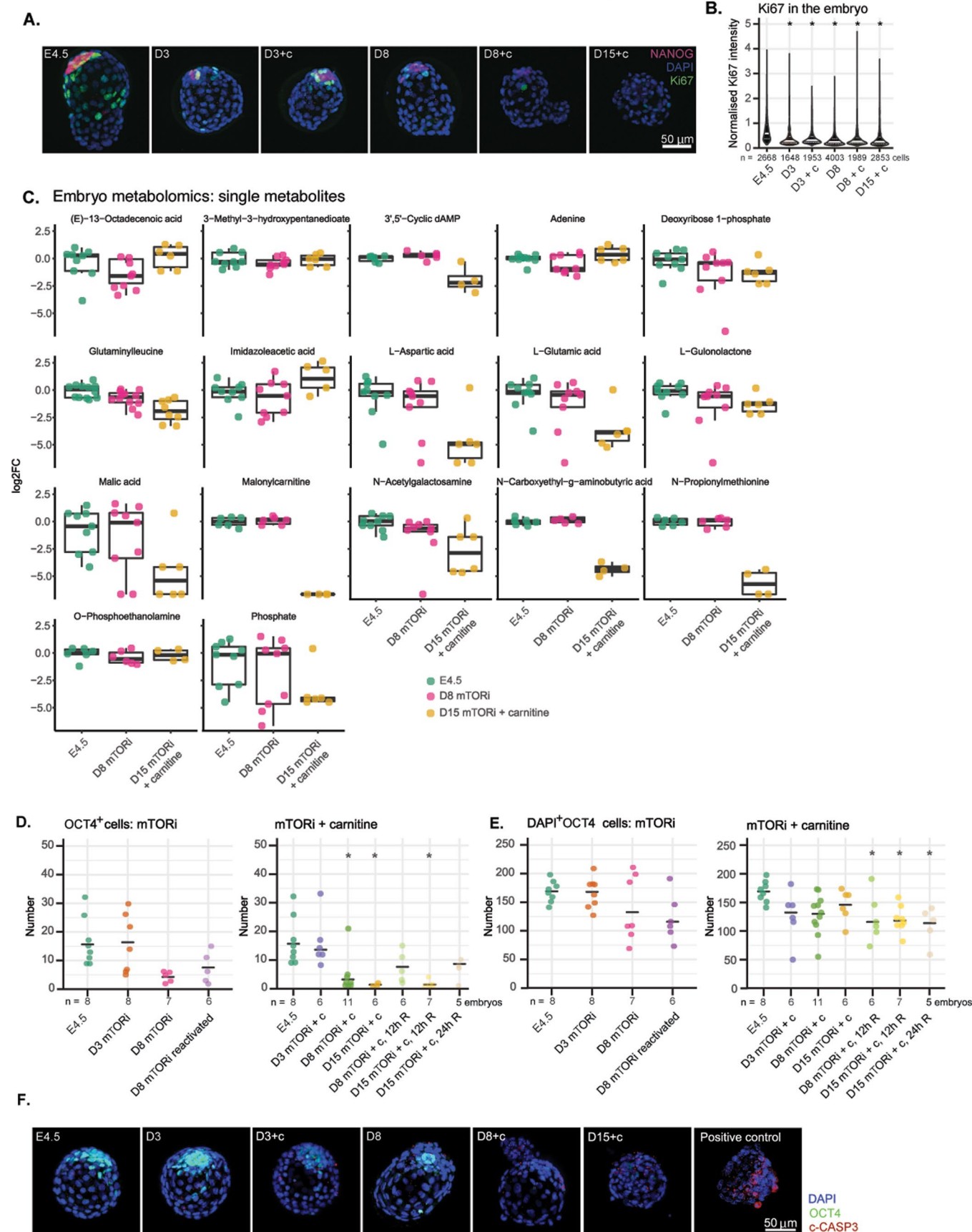

**Extended Data Fig. 6 | See next page for caption.**

**Extended Data Fig. 6 | Detailed characterization of carnitine-supplemented embryos. (a)** Representative IF images of embryos stained against Ki67 and NANOG in each condition (n = 8 embryos/condition). Scale bar = 50 μm. **(b)** Quantification of DAPI-normalized Ki67 intensities in the embryo. n = number of cells, *: p < 0.05, one way ANOVA, Dunnett's post hoc test. **(c)** Single embryo metabolomics quantification data per metabolite. Each dot represents an embryo. n = 4–12 embryos have been used for each measurement. **(d)** Number of OCT4-positive cells per embryo in each condition. n = number of embryos,

*: p < 0.05, one way ANOVA, Dunnett's post hoc test. **(e)** Number of TE cells per embryo in each condition. TE cells are defined as DAPI-positive and OCT4-negative. n = number of embryos, *: p < 0.05, one way ANOVA, Dunnett's post hoc test. **(f)** Representative IF images of embryos stained against cleaved-CASPASE3 and OCT4 in each condition (n = 5 embryos/condition). As positive control, E4.5 embryos were exposed to UV with a UV crosslinker. energy 4000 uJ/cm2 for 8 seconds and subsequently cultured for another 6 hours. Across conditions, some TE cells display cleaved-caspase3 staining. Scale bar = 50 μm.

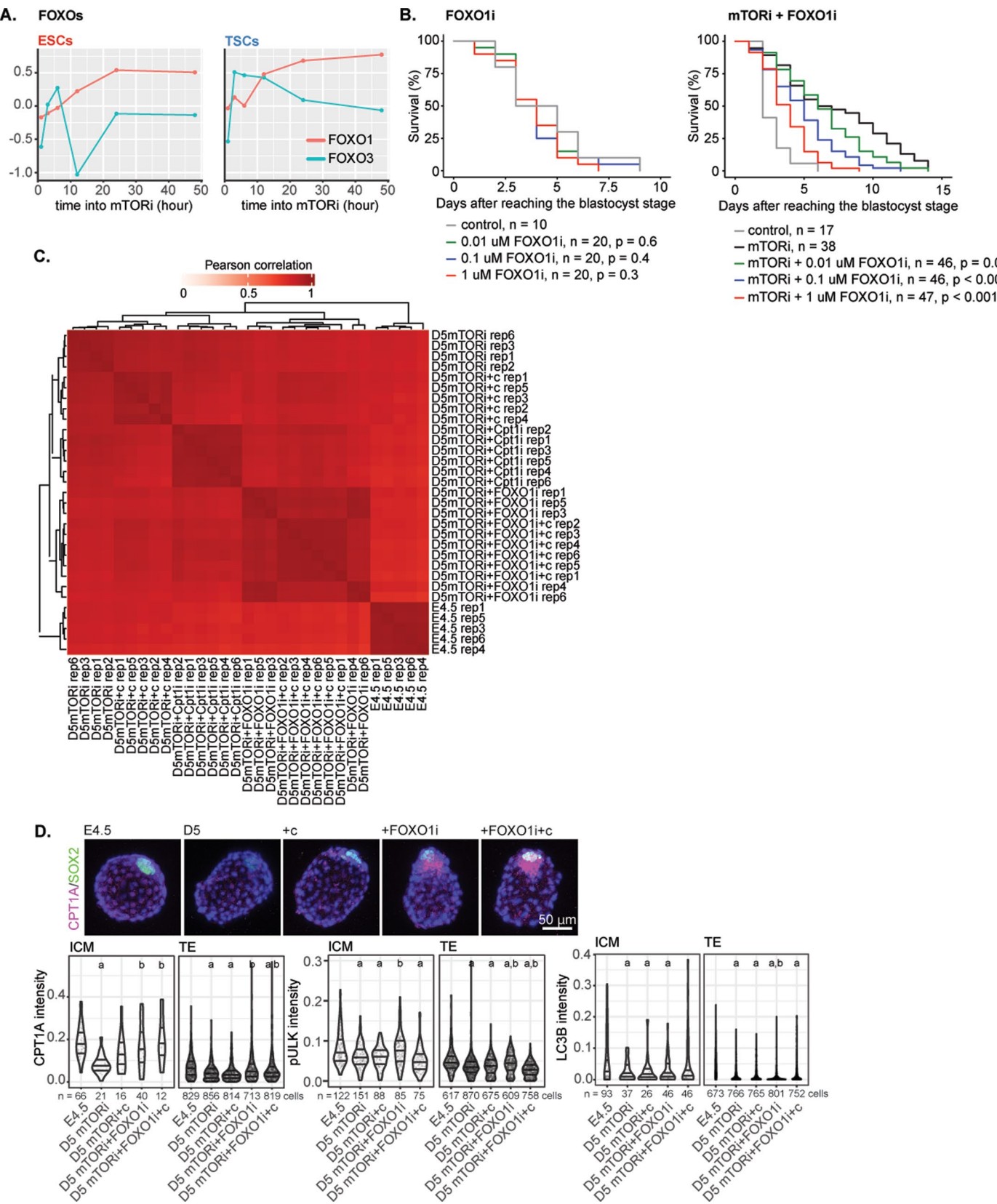

**Extended Data Fig. 7 | FOXO and CPT expression dynamics. (a)** FOXO1 and FOXO3 protein expression over time in ESCs and TSCs treated with mTORi. **(b)** Embryo survival curves in the shown conditions. n = number of embryos. Statistical test is the G-rho family test of Harrington and Fleming[89]. **(c)** Single embryo proteomics heatmap displaying the Pearson correlation between the individual samples using the expression values of the differentially expressed proteins. **(d)** IF staining for CPT1A and SOX2 in each condition. Embryos (n = 5/condition) were paused for 5 days with or without L-carnitine, FOXO11i and FOXO1i+carnitine. Representative embryos' z-projections are shown. Scale bar = 50 μm. Additionally, quantifications of area normalized CPT1A, pULK and LC3B are shown. a and b: p < 0.05 compared to E4.5 and D5 mTORi, respectively, computed with a one way ANOVA and Tukey HSD posthoc test.

# Reporting Summary

## Statistics

For all statistical analyses, confirm that the following items are present in the figure legend, table legend, main text, or Methods section.

| n/a | Confirmed | |
|---|---|---|
| ☐ | ☒ | The exact sample size (*n*) for each experimental group/condition, given as a discrete number and unit of measurement |
| ☐ | ☒ | A statement on whether measurements were taken from distinct samples or whether the same sample was measured repeatedly |
| ☐ | ☒ | The statistical test(s) used AND whether they are one- or two-sided *Only common tests should be described solely by name; describe more complex techniques in the Methods section.* |
| ☒ | ☐ | A description of all covariates tested |
| ☐ | ☒ | A description of any assumptions or corrections, such as tests of normality and adjustment for multiple comparisons |
| ☐ | ☒ | A full description of the statistical parameters including central tendency (e.g. means) or other basic estimates (e.g. regression coefficient) AND variation (e.g. standard deviation) or associated estimates of uncertainty (e.g. confidence intervals) |
| ☐ | ☒ | For null hypothesis testing, the test statistic (e.g. *F*, *t*, *r*) with confidence intervals, effect sizes, degrees of freedom and *P* value noted *Give P values as exact values whenever suitable.* |
| ☒ | ☐ | For Bayesian analysis, information on the choice of priors and Markov chain Monte Carlo settings |
| ☒ | ☐ | For hierarchical and complex designs, identification of the appropriate level for tests and full reporting of outcomes |
| ☐ | ☒ | Estimates of effect sizes (e.g. Cohen's *d*, Pearson's *r*), indicating how they were calculated |

*Our web collection on statistics for biologists contains articles on many of the points above.*

## Software and code

Policy information about availability of computer code

| | |
|---|---|
| Data collection | For ESC and TSC proteomics, LC-MS/MS was carried out by nanoflow reverse phase liquid chromatography (Dionex Ultimate 3000, Thermo Scientific) coupled online to a Q-Exactive HF Orbitrap mass spectrometer (Thermo Scientific). ESC metabolites were quantified by metabolite extraction and tandem LC-MS/MS measurements. For embryo metabolite detection, Mass Spectrometry Imaging was performed with the AP-SMALDI5-Orbitrap MS. For embryo proteomics, digests were loaded onto Evotip Pure (Evosep, Odense, Denmark), peptide separation was carried out by nanoflow reverse phase liquid chromatography (Evosep One, Evosep) using the Aurora Elite column (15 cm x 75 µm ID, C18 1.7 µm beads, IonOptics, Victoria, Australia). The LC system was coupled to the timsTOF SCP mass spectrometer (Bruker Daltonics, Bremen, Germany). The Agilent Seahorse XFp Analyzer was used for ESC and TSC energetics and the Mito Fuel Flex test. The Tecnai Spirit transmission electron microscope (FEI) operated at 120 kV, equipped with a 4kx4k F416 CMOS camera (TVIPS) was used for ultra-structure analysis of the embryos. The Zeiss Plan-Apochromat 20x/0.8 objective on the Zeiss LSM880 Airy microscope using Airy scan was used for lipid droplet quantifications, H4K16ac, OCT4, CDX2, FOXO1, c-CASPASE3, LaminB1, H3K9me2, KI67, p-mTOR, pS6, pAKT, pAMPK, pACC, pULK, LC3B, LAMP1, SOX2, and CPT1A. The Zeiss Plan-Apochromat 63x 1.4NA oil objective was used on the Zeiss LSM880 Airy microscope using Airy scan for LaminB1 and H3K9me2 imaging. Airy scan mode and image processing was done using Zen black software (version 2.3). |
| Data analysis | MaxQuant software (v1.6.10.43): Raw MS data processing. Dia-NN (v1.8.1): Peptide search for embryo proteomics. Data analysis and visualization: R (version 4.1.0), RStudio (version 1.3.1093 with R version 3.6.3) survminer (version 0.4.9) and survival (version 3.3-1) packages for survival curves Mass spec: Perseus (version 1.6.14.0), Dia-NN (v1.8.1), and DEP packages (version 1.14.0) Destiny package (version 3.8.0): diffusion maps and pseudotime calculation. |

MetaboAnalyst 4.0: time series analysis of proteomics data.
The R stats package "stats" (version 4.1.0): k-means clustering.
clusterProfiler (version 4.0.5): identification of enriched Biological Processes.
Plotting: ggplot2 (version 3.3.5) and R package 'ComplexHeatmap'
MultiQuantTM software (version 2.1.1): relative quantification metabolomics data.
pheatmap package (version 1.0.12): heatmaps
Thermo imagequest software (version 1.1): Thermo RAW data analysis.
https://metaspace2020.eu: annotation and interpretation of MALDI-imaging data.
ImageJ (version 1.53): metabolite quantification
Wave software (version 2.4): Seahorse Mitostress and Glyostress programs.
ZEN Blue software (version 3.4): electron microscopy image analysis.
Fiji ImageJ2 (version 2.3.0): confocal image processing.
CellProfiler (version 4.2.1): confocal image quantifications
STAR-2.5.3a and HTSeq 0.11.4: RNAseq read alignmnent and count
DESeq2: differential gene expression

For manuscripts utilizing custom algorithms or software that are central to the research but not yet described in published literature, software must be made available to editors and reviewers. We strongly encourage code deposition in a community repository (e.g. GitHub). See the Nature Portfolio guidelines for submitting code & software for further information.

## Data

Policy information about availability of data

All manuscripts must include a data availability statement. This statement should provide the following information, where applicable:

- Accession codes, unique identifiers, or web links for publicly available datasets
- A description of any restrictions on data availability
- For clinical datasets or third party data, please ensure that the statement adheres to our policy

UniProtKB and the mouse reference genome mm10 were used for peptide and read mapping. The proteomics datasets generated in this study have been deposited to the PRIDE database and are available via ProteomeXchange with the identifiers PXD033750, PXD033798, PXD041325. The metabolomics dataset has been deposited to peptideAtlas under the accession number PASS01758. Publicly available datasets used in this study are available under the accession numbers GSE126338, PRJEB13002, GSE121589, GSE101576, GSE116997, GSE138243, GSE138884, GSE135705, and GSE168617.

## Human research participants

Policy information about studies involving human research participants and Sex and Gender in Research.

| Reporting on sex and gender | N/A |
| --- | --- |
| Population characteristics | N/A |
| Recruitment | N/A |
| Ethics oversight | N/A |

Note that full information on the approval of the study protocol must also be provided in the manuscript.

# Field-specific reporting

Please select the one below that is the best fit for your research. If you are not sure, read the appropriate sections before making your selection.

☒ Life sciences    ☐ Behavioural & social sciences    ☐ Ecological, evolutionary & environmental sciences

For a reference copy of the document with all sections, see nature.com/documents/nr-reporting-summary-flat.pdf

# Life sciences study design

All studies must disclose on these points even when the disclosure is negative.

| | |
|---|---|
| Sample size | No statistical method was used to predetermine sample size, but our sample sizes are similar to those reported in previous publications13,76. The number of replicates for each analysis is indicated in the text or figure legends. For the ESC and TSC proteomics and ESC metabolomics experiment, three biological replicates were used. For the embryo proteomics experiment, four to five biological replicates were used. At least 5 embryos per condition were used for the MALDI-imaging experiments. Four embryos per condition were used for electron microscopy. For all confocal fluorescence imaging, at least 3 embryos were analyzed per condition. The number of analyzed embryos or cells is indicated in the figures. |
| Data exclusions | No data were excluded from the analysis. |
| Replication | The ESC and TSC proteomics and TSC metabolomics were performed with three biological replicates. The embryo proteomics experiment was performed with four to five biological replicates. The embryo supplementation experiments were performed once for supplements that did not improve developmental pausing length, and five times for carnitine supplementation experiments. The embryo staining experiments were performed with at least 3 biological replicates depending on the staining conditions. All attempts at replication were successful. |
| Randomization | For cell and embryo culture experiments, treatment groups were randomly attributed. |
| Blinding | No blinding was done because either the phenotype was obvious or treatment material needed to be refreshed often. |

# Reporting for specific materials, systems and methods

We require information from authors about some types of materials, experimental systems and methods used in many studies. Here, indicate whether each material, system or method listed is relevant to your study. If you are not sure if a list item applies to your research, read the appropriate section before selecting a response.

## Materials & experimental systems

| n/a | Involved in the study |
|---|---|
| ☐ | ☒ Antibodies |
| ☐ | ☒ Eukaryotic cell lines |
| ☒ | ☐ Palaeontology and archaeology |
| ☐ | ☒ Animals and other organisms |
| ☒ | ☐ Clinical data |
| ☒ | ☐ Dual use research of concern |

## Methods

| n/a | Involved in the study |
|---|---|
| ☒ | ☐ ChIP-seq |
| ☒ | ☐ Flow cytometry |
| ☒ | ☐ MRI-based neuroimaging |

# Antibodies

| | |
|---|---|
| Antibodies used | Primary antibodies for stainings (dilutions for embryos/ESCs and TSCs):<br>Oct3/4 mouse anti-mouse antibody (Santa Cruz, cat.# sc-5279): 1:50/1:100<br>H4K16ac rabbit anti-mouse antibody (Millipore, cat.# 7329): 1:100<br>c-Caspase3 rabbit anti-mouse antibody (Cell Signaling Technology, cat.# 9661S): 1:100<br>FoxO1 rabbit anti-mouse antibody (Cell Signaling Technology, cat.# 2880T): 1:50<br>Anti-Lamin B1 antibody - Nuclear Envelope Marker (Abcam, cat#. 16048): 1:100/1:400<br>Anti-Histone H3 (di methyl K9) antibody (Abcam, cat.# 1220): 1:100/1:200<br>Ki67 mouse anti-mouse antibody (BD Pharmingen, cat.# 556003): 1:100<br>pmTOR rabbit anti-human antibody (Abcam, cat.# ab131538): 1:100/1:200<br>pS6 rabbit anti-mouse antibody (Cell Signaling Technology, cat.# 4858S): 1:100/1:200<br>pAKT rabbit anti-mouse antibody (Cell Signaling Technology, cat.# 4060T): 1:100/1:200<br>pAMPK rabbit anti-human antibody (Abcam, cat.# ab23875): 1:100<br>pACC rabbit anti-mouse antibody: (Cell Signaling Technology, cat.# 11818T): 1:100<br>pULK rabbit anti-mouse antibody: (Cell Signaling Technology, cat.# 14202S): 1:100<br>LC3B mouse anti-mouse antibody: (Cell Signaling Technology, cat.# 83506S): 1:100<br>LAMP1 rabbit anti-mouse antibody (Cell Signaling Technology, cat.# 99437S): 1:100<br>CPT1A mouse anti-mouse antibody (Abcam, cat.# ab128568): 1:100<br>SOX2 goat anti-mouse antibody (R&D System, cat.# AF2018): 1:100<br>CDX2 mouse anti-mouse antibody (Biogenex, cat.# MU392A-UC): -/1:200<br><br>Secondary antibodies (dilution for embryos/ESCs and TSCs):<br>donkey anti-rabbit Alexa Fluor Plus 647 (Thermo Fisher Scientific, cat.# A32795): 1:200/1:1000 |

| | donkey anti-mouse Alexa Fluor 488 (Thermo Fisher Scientific, cat.# A21202): 1:200/1:1000 |
| | donkey anti-goat Alexa Fluor 594 (Thermo Fisher Scientific, cat.# A11058) 1:200 |

| Validation | All antibodies were previously validated by vendors: |
| | Oct3/4 mouse anti-mouse (Santa Cruz, cat.# sc-5279): https://www.scbt.com/p/oct-3-4-antibody-c-10?gclid=CjwKCAjwq5-WBhB7EiwAl-HEkuUFTBDEgirOz10qGUglWctHcjRfG2TUH1SdnmLe-GNXcklhDZNKLRoCJbMQAvD_BwE |
| | H4K16ac rabbit anti-mouse antibody (Millipore, cat.# 7329): https://www.merckmillipore.com/DK/en/product/Anti-acetyl-Histone-H4-Lys16-Antibody,MM_NF-07-329 |
| | c-Caspase3 rabbit anti-mouse antibody (Cell Signaling Technology, cat.# 9661S): https://www.cellsignal.com/products/primary-antibodies/cleaved-caspase-3-asp175-antibody/9661 |
| | FoxO1 rabbit anti-mouse antibody (Cell Signaling Technology, cat.# 2880T): https://www.cellsignal.com/products/primary-antibodies/foxo1-c29h4-rabbit-mab/2880?site-search-type=Products&N=4294956287&Ntt=2880t&fromPage=plp&_requestid=2248978 |
| | Anti-Lamin B1 antibody - Nuclear Envelope Marker (Abcam, cat#. 16048): https://www.abcam.com/lamin-b1-antibody-nuclear-envelope-marker-ab16048.html |
| | Anti-Histone H3 (di methyl K9) antibody (Abcam, cat.# 1220): https://www.abcam.com/histone-h3-di-methyl-k9-antibody-mabcam-1220-chip-grade-ab1220.html |
| | Ki67 mouse anti-mouse (BD Pharmingen, cat.# 556003): https://www.bdbiosciences.com/en-be/products/reagents/microscopy-imaging-reagents/immunofluoroscence-reagents/purified-mouse-anti-ki-67.556003 |
| | pmTOR rabbit anti-human (Abcam, cat.# ab131538): https://www.abcam.com/products/primary-antibodies/mtor-phospho-s2448-antibody-ab131538.html |
| | pS6 rabbit anti-mouse antibody (Cell Signaling Technology, cat.# 4858S): https://www.cellsignal.com/products/primary-antibodies/phospho-s6-ribosomal-protein-ser235-236-d57-2-2e-xp-rabbit-mab/4858?_requestid=723559 |
| | pAKT rabbit anti-mouse antibody (Cell Signaling Technology, cat.# 4060T): https://www.cellsignal.com/products/primary-antibodies/phospho-akt-ser473-d9e-xp-rabbit-mab/4060 |
| | pAMPK rabbit anti-human antibody (Abcam, cat.# ab23875): https://www.abcam.com/products/primary-antibodies/ampk-alpha-1-phospho-t183--ampk-alpha-2-phospho-t172-antibody-ab23875.html |
| | pACC rabbit anti-mouse: (Cell Signaling Technology, cat.# 11818T): https://www.cellsignal.com/products/primary-antibodies/phospho-acetyl-coa-carboxylase-ser79-d7d11-rabbit-mab/11818 |
| | pULK rabbit anti-mouse antibody: (Cell Signaling Technology, cat.# 14202S): https://www.cellsignal.com/products/primary-antibodies/phospho-ulk1-ser757-d7o6u-rabbit-mab/14202 |
| | LC3B mouse anti-mouse antibody: (Cell Signaling Technology, cat.# 83506S): https://www.cellsignal.com/products/primary-antibodies/lc3b-e5q2k-mouse-mab/83506 |
| | LAMP1 rabbit anti-mouse antibody (Cell Signaling Technology, cat.# 99437S): https://www.cellsignal.com/products/primary-antibodies/lamp1-e5n9z-rabbit-mab/99437 |
| | CPT1A mouse anti-mouse antibody (Abcam, cat.# ab128568): https://www.abcam.com/products/primary-antibodies/cpt1a-antibody-8f6ae9-ab128568.html |
| | SOX2 goat anti-mouse antibody (R&D System, cat.# AF2018): https://www.rndsystems.com/products/human-mouse-rat-sox2-antibody_af2018 |
| | CDX2 mouse anti-mouse antibody (Biogenex, cat.# MU392A-UC): https://biogenex.com/wp-content/uploads/2019/11/932-392M-EN.pdf |

# Eukaryotic cell lines

Policy information about cell lines and Sex and Gender in Research

| Cell line source(s) | ESCs: E14 cells - mouse embryonic stem cells (received from S. Kinkley Lab, MPIMG). |
| | TSCs: mouse trophoblast stem cells (received from M. Zernicka-Goetz Lab). |
| | MEFs: mouse embryonic fibroblasts (derived in house from CD1 embryos). |

| Authentication | Cell lines used in this study were sextyped. |

| Mycoplasma contamination | Cell lines tested frequently negative for mycoplasma contamination. |

| Commonly misidentified lines (See ICLAC register) | Cell lines used in this paper are not listed in the ICLAC register. |

# Animals and other research organisms

Policy information about studies involving animals; ARRIVE guidelines recommended for reporting animal research, and Sex and Gender in Research

| Laboratory animals | CD1 or b6d2F1 were used. All animal experiments were performed according to local animal welfare laws and approved by the local authority Landesamt für Gesundheit und Soziales (license numbers ZH120, G0284/18, G021/19, and G0243/18). |

| Wild animals | No wild animals were used in this study. |

| Reporting on sex | Sex of the embryos has not been considered in the anaylses. |

| Field-collected samples | No field-collected samples were used in this study. |

| Ethics oversight | All animal experiments were performed according to local animal welfare laws and approved by local authorities (covered by LaGeSo licenses ZH120, G0284/18, G021/19, and G0243/18). |

Note that full information on the approval of the study protocol must also be provided in the manuscript.

