## [Peer Review File · Nature Cell Biology]

Peer Review Information

Journal: Nature Cell Biology

Manuscript Title: FOXO1-mediated lipid metabolism maintains mammalian embryos in dormancy

Corresponding author name(s): Dr Aydan Bulut-Karslioglu

Editorial Notes:

Reviewer Comments & Decisions:

Decision Letter, initial version:

*Please delete the link to your author homepage if you wish to forward this email to co-authors.

Dear Aydan,

Your manuscript, "Metabolic enhancement of mammalian developmental pausing", has now been seen by 3 referees, who are experts in early development, fatty acid oxidation, epigenetics (referee 1); diapause, metabolism (referee 2); and early mammalian development, lipid metabolism (referee 3). As you will see from their comments (attached below) they find this work of potential interest, but have raised substantial concerns, which in our view would need to be addressed with considerable revisions before we can consider publication in Nature Cell Biology.

Nature Cell Biology editors discuss the referee reports in detail within the editorial team, including the chief editor, to identify key referee points that should be addressed with priority, and requests that are overruled as being beyond the scope of the current study. To guide the scope of the revisions, I have listed these points below. I should stress that the referees' concerns point to a premature dataset and these points would need to be addressed with experiments and data, and reconsideration of the study for this journal and re-engagement of referees would depend on strength of these revisions.

In particular, it would be essential to:

(A) Provide further mechanistic characterization, insight and validation at the points of the study indicated by:

Referee 1:

"Through proteomics analysis coupled with computational simulation, they identified lipid degradation processes (e.g., FAO) as one of the adaptive pathways to support the mTORi paused state. The proteomics results should be validated. The importance of FAO activity should be functionally tested in ESCs. The conclusion that FAO is likely the main energy pathway in paused pluripotent cells (line 171-172) is not well supported without functional testing using FAO inhibitors, or FAO inhibitors in conjunction with mTOR inhibitors. The authors should also perform Seahorse analysis or FAO rate assay after FAO/mTORi inhibition in ESCs. The results from ESCs need to be compared with the phenotype observed in embryos. In addition, does extensive chromatin reorganization also occur in paused ESCs, just like in paused embryos? A parallel study between ESCs and embryos will greatly strengthen the claim and rule out the embryo phenotype is exclusively from changes in epiblast".

"The authors identified FOXO1 as a critical regulator for Carnitine functions in mTORi paused embryos. Does addition of Carnitine affect FOXO1 expression? Is there a feedback regulation that depends on mTOR inhibition? This should be tested".

Referee 2:

"A theme that is repeated in the description of these results is that mTor treatment results in chromatin reorganization. Speculation in lines 151-153 reiterates this view, but without supporting data. Further explanation of the data and/or the reasoning behind this concept would be useful".

"Similarly, data in Figure 4F seem to indicate that there is a greater number of lipid droplets prior to mTOR induced pause than after. Pause is characterized by FA degradation (Figure 1F). Is there new uptake of lipids associated with the state of cellular pause?"

"It is stated that Foxo1 plays a critical role in cellular dormancy (lines 286 et seq.). There is some speculation about its role in the discussion, but, overall no mechanism is proposed".

Referee 3:

"Since changes in mTOR activity by mTORi are key in this study, the phosphorylation levels of mTOR and its downstream factors (such as S6 and 4EBP) in ESCs and TSCs must be confirmed in each experiment. Especially in experiments with carnitine and Etmoxir (Figure 3), in addition to changes in mTOR activity, phosphorylation levels of AMPK and ACC should also be examined by Western Blotting

and immunostaining".

"The interpretation of the results in Fig. 1E is somewhat puzzling. Why would inhibition of mTOR, a negative regulator of autophagy, conversely suppress autophagy in ESCs? Is it necessary to temporarily suppress autophagy activity in order to maintain dormancy? examining the levels of autophagy and mTOR activity over time after the addition of mTORi may provide answers to these questions. In this connection, suppression of the SNARE pathway, which is involved in the formation of autolysosomes, should also be considered".

"FOXO1 is known to induce autophagy (Nat Cell Biol. 2010 12(7):665-75, PMID: 20543840), therefore it is necessary to examine whether inhibition of FOXO1 alters autophagy activity. Related to the above comment, if inhibition of FOXO1 suppresses autophagy induction, it needs to be tested whether it is transient or autophagy-independent".

(B) Address the issues raised by referee 1 regarding the claims made about TSCs and their involvement in diapause.

"Authors reasoned that since ESCs and TSCs have divergent response to mTORi, they can be used as surrogates to identify essential pathways that specifically regulate the survival of mTORi-induced dormancy. This rationale is somewhat flawed since ESC and TSC are very different both epigenetically and metabolically and their cultural conditions are also different. For example, why ESC vs. TSC comparison is more meaningful than ESC vs. MEF? MEFs also don't enter dormancy upon mTORi inhibition".

"Related to point 1, previous publications indicated that TE contributes to entry and release from embryo diapause. Although TSCs do not enter into dormancy upon mTORi inhibition in vitro, it does not mean they play no roles in embryos diapause. Multiple pathways, e.g. Hippo, were perturbed in TSCs after mTORi treatment, their functions in embryo diapause or their extension should be tested or at least discussed to have a more comprehensive understanding of diapause regulation".

(C) Provide a more thorough functional characterization of the cultured embryos with prolonged survival, as indicated by:

Referee 1:

"One major concern is that while paused embryos by mTORi + L-carnitine treatment had prolonged survival (~ 34 days as claimed), there is no functional testing of these embryos to demonstrate that they are developmentally competent, healthy and able to give birth to live animals with no defects. It is very important to test embryos that survived beyond 10 days when most mTORi treated embryos died (Fig 3E). Since extension of survival by mTORi is the main point of the paper, this is essential to support the main conclusions".

Referee 2:

"Embryos treated with mTor and carnitine persisted longer in culture. The important question relates to whether the embryos could continue development after washout of these two factors. It would see from Figure 5B and the text in lines 253 et seq. that recovery of development was possible, but that it was substantially slower in those embryos that were in the paused state for the longer times. Is there any evidence relevant to the mechanism of differential temporal recovery?"

(D) All other referee concerns pertaining to strengthening existing data, providing controls, methodological details, clarifications and textual changes, should also be addressed.

(E) Finally please pay close attention to our guidelines on statistical and methodological reporting (listed below) as failure to do so may delay the reconsideration of the revised manuscript. In particular please provide:

- a Supplementary Figure including unprocessed images of all gels/blots in the form of a multi-page pdf file. Please ensure that blots/gels are labeled and the sections presented in the figures are clearly indicated.
- a Supplementary Table including all numerical source data in Excel format, with data for different figures provided as different sheets within a single Excel file. The file should include source data giving rise to graphical representations and statistical descriptions in the paper and for all instances where the figures present representative experiments of multiple independent repeats, the source data of all repeats should be provided.

We would be happy to consider a revised manuscript that would satisfactorily address these points, unless a similar paper is published elsewhere, or is accepted for publication in Nature Cell Biology in the meantime.

- ensure that it conforms to our format instructions and publication policies (see below and <https://www.nature.com/nature/for-authors>).
- provide a point-by-point rebuttal to the full referee reports verbatim, as provided at the end of this letter.
- provide the completed Reporting Summary (found here <https://www.nature.com/documents/nr-reporting-summary.pdf>). This is essential for reconsideration of the manuscript will be available to editors and referees in the event of peer review. For more information see <http://www.nature.com/authors/policies/availability.html> or contact me.

When submitting the revised version of your manuscript, please pay close attention to our [href="https://www.nature.com/nature-portfolio/editorial-policies/image-integrity">Digital Image Integrity Guidelines](https://www.nature.com/nature-portfolio/editorial-policies/image-integrity). and to the following points below:

Nature Cell Biology is committed to improving transparency in authorship. As part of our efforts in this direction, we are now requesting that all authors identified as 'corresponding author' on published papers create and link their Open Researcher and Contributor Identifier (ORCID) with their account on the Manuscript Tracking System (MTS), prior to acceptance. ORCID helps the scientific community achieve unambiguous attribution of all scholarly contributions. You can create and link your ORCID from the home page of the MTS by clicking on 'Modify my Springer Nature account'. For more information please visit www.springernature.com/orcid.

This journal strongly supports public availability of data. Please place the data used in your paper into a public data repository, or alternatively, present the data as Supplementary Information. If data can only be shared on request, please explain why in your Data Availability Statement, and also in the correspondence with your editor. Please note that for some data types, deposition in a public repository is mandatory - more information on our data deposition policies and available repositories appears below.

[Redacted]

We would like to receive a revised submission within six months.

We hope that you will find our referees' comments, and editorial guidance helpful. Please do not hesitate to contact me if there is anything you would like to discuss.

Best wishes,

Stelios

Stylios Lefkopoulos, PhD
He/him/his
Associate Editor
Nature Cell Biology
Springer Nature
Heidelberger Platz 3, 14197 Berlin, Germany

E-mail: stylios.lefkopoulos@springernature.com
Twitter: @s_lefkopoulos

Reviewers' Comments:

Reviewer #1:
Remarks to the Author:
Comments:

In the previous study, the authors found that inhibition of mTOR (mTORi) induces mouse blastocyst pausing ex vivo, mimicking embryonic diapause in mammals. However, the ex vivo paused embryos

collapse over time. In the study of “Metabolic enhancement of mammalian developmental pausing” by Weijden et al., authors performed proteomics studies to identify pathways downstream of mTORi that may extend the blastocyst pause duration. They hypothesized that since TSCs do not enter dormancy after mTOR inhibition, the differential proteomic changes in ESC vs TSC will point to essential pathways for durable dormancy. They reported that not surprisingly, ESCs and TSCs had divergent response to mTORi inhibition. Among them, lipid and amino acid degradation pathways were among multiple pathways in ESCs. Authors further showed that enhancing lipid usage via supplementing fatty acid oxidation (FAO) metabolite L-carnitine prolongs mTORi-induced embryo pausing ex vivo up to 34 days and this is mediated by FOXO1. While the manuscript addresses an important question with significance in developmental biology, there are several major concerns that need be addressed:

Major concerns:

1. Authors reasoned that since ESCs and TSCs have divergent response to mTORi, they can be used as surrogates to identify essential pathways that specifically regulate the survival of mTORi-induced dormancy. This rationale is somewhat flawed since ESC and TSC are very different both epigenetically and metabolically and their cultural conditions are also different. For example, why ESC vs. TSC comparison is more meaningful than ESC vs. MEF? MEFs also don't enter dormancy upon mTORi inhibition.
2. Through proteomics analysis coupled with computational simulation, they identified lipid degradation processes (e.g., FAO) as one of the adaptive pathways to support the mTORi paused state. The proteomics results should be validated. The importance of FAO activity should be functionally tested in ESCs. The conclusion that FAO is likely the main energy pathway in paused pluripotent cells (line 171-172) is not well supported without functional testing using FAO inhibitors, or FAO inhibitors in conjunction with mTOR inhibitors. The authors should also perform Seahorse analysis or FAO rate assay after FAO/mTORi inhibition in ESCs. The results from ESCs need to be compared with the phenotype observed in embryos. In addition, does extensive chromatin reorganization also occur in paused ESCs, just like in paused embryos? A parallel study between ESCs and embryos will greatly strengthen the claim and rule out the embryo phenotype is exclusively from changes in epiblast.
3. Related to point 1, previous publications indicated that TE contributes to entry and release from embryo diapause. Although TSCs do not enter into dormancy upon mTORi inhibition in vitro, it does not mean they play no roles in embryos diapause. Multiple pathways, e.g. Hippo, were perturbed in TSCs after mTORi treatment, their functions in embryo diapause or their extension should be tested or at least discussed to have a more comprehensive understanding of diapause regulation.
2. One major concern is that while paused embryos by mTORi + L-carnitine treatment had prolonged survival (~ 34 days as claimed), there is no functional testing of these embryos to demonstrate that they are developmentally competent, healthy and able to give birth to live animals with no defects. It is very important to test embryos that survived beyond 10 days when most mTORi treated embryos died (Fig 3E). Since extension of survival by mTORi is the main point of the paper, this is essential to support the main conclusions.
4. The previous study by Khoa et al, cited in the paper, showed that FAO inhibition leads to ESC pausing in vitro, which can be rescued by FAO metabolites. This is consistent with down regulation of FAO pathways in diapaused embryos. How does this reconcile with the results here that mTORi inhibition led to increase of FAO activity? Does that mean mTORi induced ESC dormancy is different from naturally occurring embryonic diapause in animals? A more thorough discussion may be warranted.
5. Why D15 mTORi + Carnitine embryos were used in Figure 5, but not later day embryos (e.g. day

20 or 30)? Although embryo survival decreased over time, choosing longer time point would further strengthen the arguments.

6. The authors identified FOXO1 as a critical regulator for Carnitine functions in mTORi paused embryos. Does addition of Carnitine affect FOXO1 expression? Is there a feedback regulation that depends on mTOR inhibition? This should be tested.

Minor concerns:

1. Figure legends lack essential information. Labels in the Figures are sometimes hard to follow. For examples, line 99-101 mentioned about parental ESCs in Fig. 1C, but it is difficult to find where they are. Fig. S1B, gene or protein expression?

2. Line 147-148, difference of H3K9me2 in EPI and TE needs to be better explained. Quantifications of Fig. 1F, G should be added.

Reviewer #2:

Remarks to the Author:

The manuscript authored by V. van der Weijden describes experiments to characterise the mechanisms by which inhibition of the mTor pathway in stem cells and embryos induces proliferative quiescence. Based on comparison of paused ICM and trophoblast derived stem cells, lipid metabolism pathways were identified as potential factors in the maintenance of the mitotically inactive state. The most interesting aspect of the manuscript is the extension of embryonic diapause and the viability of embryos in this state by addition of carnitine to enhance fatty acid oxidation. The approach is appropriate and valid, and the data are robust. Thus, manuscript provides timely new insight into the phenomenon of arrest in cell differentiation essential to understanding embryonic stem cells and embryonic diapause. Although the manuscript has been well-prepared, there are some issues that could be addressed.

1. It is stated that the real time and pseudotime results depicted in Figure 1C largely agree. It might be useful to plot these data in a similar way so they can be readily compared between methods of analysis. What might be the physiological significance of pseudotime in this context?

2. Figure 2A is used to describe immediate vs adaptive response to mTor inhibition. The differences between the two conditions are not abundantly obvious, as, in some cases the responses are quite similar.

3. A theme that is repeated in the description of these results is that mTor treatment results in chromatin reorganization. Speculation in lines 151-153 reiterates this view, but without supporting data. Further explanation of the data and/or the reasoning behind this concept would be useful.

4. The description of Figure 2E in the legend is not adequate, and it is not clear from the figure what the image is showing.

5. Figure 3A-B: Heat maps of gene expression are presented. It is not clear from the legend whether these are proteomic or some other sort of data. It's a minor point.

6. As noted above, the exciting aspect of this study is that carnitine supplementation can maintain diapause. What is the source of the carnitine regulating diapause in embryos that in this state in vivo, or in vitro? In the same vein, line 190 alludes to the idea that the source of lipids that maintain diapause is intracellular. Is this speculation, or are there data to support this proposition?

7. Similarly, data in Figure 4F seem to indicate that there is a greater number of lipid droplets prior to mTOR induced pause than after. Pause is characterized by FA degradation (Figure 1F). Is there new uptake of lipids associated with the state of cellular pause?

8. A percentage of the epiblast structures form rosettes, indicative of the onset of epithelial polarity. This is an interesting finding that does not receive much discussion beyond noting that carnitine

supplementation appears to increase the frequency of rosette formation.

9. It is stated that Foxo1 plays a critical role in cellular dormancy (lines 286 et seq.). There is some speculation about its role in the discussion, but, overall no mechanism is proposed.

10. Embryos treated with mTor and carnitine persisted longer in culture. The important question relates to whether the embryos could continue development after washout of these two factors. It would see from Figure 5B and the text in lines 253 et seq. that recovery of development was possible, but that it was substantially slower in those embryos that were in the paused state for the longer times. Is there any evidence relevant to the mechanism of differential temporal recovery?

11. There are constraints on length of the text for this journal. Given the exciting nature of the results, i.e. that diapause can be further extended, and that a clear mechanism has been identified in carnitine, the discussion seems not to do justice to the data. Perhaps it could be rewritten with strong focus on mechanisms and significance of the data, and a bit less speculation.

Reviewer #3:

Remarks to the Author:

In this study, the authors performed proteomic and metabolomic analyses of ESCs and TSCs to clarify the mechanism of maintaining dormancy of blastocyst-derived ESCs in vitro by mTOR inhibition, and found significant differences in the properties of ESCs and TSCs. In particular, they demonstrated that the metabolism of lipids and amino acids is reduced in dormant ESCs, thus, long-term and stable dormancy is possible by compensating for these functions. Furthermore, the authors showed that Foxo1-mediated lipid metabolism affects dormancy. It is particularly interesting that there is such a difference in the properties of ESCs and TSCs derived from the same blastocyst and that mTOR inhibition can maintain the integrity of ESCs for a long time in vitro. Overall, the authors' interpretation of the experimental results is reasonable, and most of their claims seem to be supported by the experimental results. However, for some experiments, the current data are insufficient to assess their validity. In particular, since mTOR is involved in a variety of cellular functions, it would be desirable to carefully examine the level of activation of mTOR and related signaling pathways in each experiment.

Major comments:

Since changes in mTOR activity by mTORi are key in this study, the phosphorylation levels of mTOR and its downstream factors (such as S6 and 4EBP) in ESCs and TSCs must be confirmed in each experiment. Especially in experiments with carnitine and Etmozir (Figure 3), in addition to changes in mTOR activity, phosphorylation levels of AMPK and ACC should also be examined by Western Blotting and immunostaining.

The interpretation of the results in Fig. 1E is somewhat puzzling. Why would inhibition of mTOR, a negative regulator of autophagy, conversely suppress autophagy in ESCs? Is it necessary to temporarily suppress autophagy activity in order to maintain dormancy? Examining the levels of autophagy and mTOR activity over time after the addition of mTORi may provide answers to these questions. In this connection, suppression of the SNARE pathway, which is involved in the formation of autolysosomes, should also be considered.

Depending on the energy level of the cell, LDs are known to interact with various organelles. In Figure 4, it is worth analyzing whether the differences in localization with lysosomes in addition to mitochondria correlate with the size and distribution of LDs observed in epiblasts and TEs.

FOXO1 is known to induce autophagy (Nat Cell Biol. 2010 12(7):665-75, PMID: 20543840), therefore it is necessary to examine whether inhibition of FOXO1 alters autophagy activity. Related to the above

comment, if inhibition of FOXO1 suppresses autophagy induction, it needs to be tested whether it is transient or autophagy-independent.

Minor comments:

line 180: it would be more helpful to the reader if the specific name of the inhibitor is mentioned (except for mTORi). e.g., Etomoxir for CPT1 inhibitor, etc. FOXO1 inhibitor as well.

In Fig. 5, the authors examine H4K16ac levels, but would like an explanation as to why they focused only on this acetylation.

line 213: The authors mention that the fixation procedure affects BODIPY staining, but in this reviewer's experience, this problem could be resolved by restaining the fixed cells with BODIPY after fixation (just before fluorescence observation).

Several terms such as Thermo, Thermo Fisher, Themo, and Themo Fisher are mixed in the Materials and Methods.

Line 262: D3 + C embryo should be D3 + c (lowercase).

ABSTRACT AND MAIN TEXT – please follow the guidelines that are specific to the format of your

manuscript, as listed in our Guide to Authors (http://www.nature.com/ncb/pdf/ncb_gta.pdf) Briefly, Nature Cell Biology Articles, Resources and Technical Reports have 3500 words, including a 150 word abstract, and the main text is subdivided in Introduction, Results, and Discussion sections. Nature Cell Biology Letters have up to 2500 words, including a 180 word introductory paragraph (abstract), and the text is not subdivided in sections.

Methods should be written concisely, but should contain all elements necessary to allow interpretation and replication of the results. As a guideline, Methods sections typically do not exceed 3,000 words. The Methods should be divided into subsections listing reagents and techniques. When citing previous methods, accurate references should be provided and any alterations should be noted. Information must be provided about: antibody dilutions, company names, catalogue numbers and clone numbers for monoclonal antibodies; sequences of RNAi and cDNA probes/primers or company names and catalogue numbers if reagents are commercial; cell line names, sources and information on cell line identity and authentication. Animal studies and experiments involving human subjects must be reported in detail, identifying the committees approving the protocols. For studies involving human subjects/samples, a statement must be included confirming that informed consent was obtained. Statistical analyses and information on the reproducibility of experimental results should be provided in a section titled "Statistics and Reproducibility".

All Nature Cell Biology manuscripts submitted on or after March 21 2016 must include a Data availability statement as a separate section after Methods but before references, under the heading "Data Availability". For Springer Nature policies on data availability see <http://www.nature.com/authors/policies/availability.html>; for more information on this particular

policy see <http://www.nature.com/authors/policies/data/data-availability-statements-data-citations.pdf>. The Data availability statement should include:

- Accession codes for primary datasets (generated during the study under consideration and designated as "primary accessions") and secondary datasets (published datasets reanalysed during the study under consideration, designated as "referenced accessions"). For primary accessions data should be made public to coincide with publication of the manuscript. A list of data types for which submission to community-endorsed public repositories is mandated (including sequence, structure, microarray, deep sequencing data) can be found here <http://www.nature.com/authors/policies/availability.html#data>.
- Unique identifiers (accession codes, DOIs or other unique persistent identifier) and hyperlinks for datasets deposited in an approved repository, but for which data deposition is not mandated (see here for details <http://www.nature.com/sdata/data-policies/repositories>).
- At a minimum, please include a statement confirming that all relevant data are available from the authors, and/or are included with the manuscript (e.g. as source data or supplementary information), listing which data are included (e.g. by figure panels and data types) and mentioning any restrictions on availability.
- If a dataset has a Digital Object Identifier (DOI) as its unique identifier, we strongly encourage including this in the Reference list and citing the dataset in the Methods.

We recommend that you upload the step-by-step protocols used in this manuscript to the Protocol Exchange. More details can be found at www.nature.com/protocolexchange/about.

All imaging data should be accompanied by scale bars, which should be defined in the legend. Cropped images of gels/blots are acceptable, but need to be accompanied by size markers, and to retain visible background signal within the linear range (i.e. should not be saturated). The boundaries of panels with low background have to be demarked with black lines. Splicing of panels should only be considered if unavoidable, and must be clearly marked on the figure, and noted in the legend with a statement on whether the samples were obtained and processed simultaneously. Quantitative comparisons between samples on different gels/blots are discouraged; if this is unavoidable, it should only be performed for samples derived from the same experiment with gels/blots were processed in parallel, which needs to be stated in the legend.

Figures should be provided at approximately the size that they are to be printed at (single column is 86 mm, double column is 170 mm) and should not exceed an A4 page (8.5 x 11"). Reduction to the scale that will be used on the page is not necessary, but multi-panel figures should be sized so that the whole figure can be reduced by the same amount at the smallest size at which essential details in each panel are visible. In the interest of our colour-blind readers we ask that you avoid using red and green for contrast in figures. Replacing red with magenta and green with turquoise are two possible

colour-safe alternatives. Lines with widths of less than 1 point should be avoided. Sans serif typefaces, such as Helvetica (preferred) or Arial should be used. All text that forms part of a figure should be rewritable and removable.

SUPPLEMENTARY INFORMATION – Supplementary information is material directly relevant to the conclusion of a paper, but which cannot be included in the printed version in order to keep the manuscript concise and accessible to the general reader. Supplementary information is an integral part of a Nature Cell Biology publication and should be prepared and presented with as much care as the main display item, but it must not include non-essential data or text, which may be removed at the editor's discretion. All supplementary material is fully peer-reviewed and published online as part of the HTML version of the manuscript. Supplementary Figures and Supplementary Notes are

appended at the end of the main PDF of the published manuscript.

The total number of Supplementary Figures (not including the “unprocessed scans” Supplementary Figure) should not exceed the number of main display items (figures and/or tables (see our Guide to Authors and March 2012 editorial <http://www.nature.com/ncb/authors/submit/index.html#suppinfo>; <http://www.nature.com/ncb/journal/v14/n3/index.html#ed>). No restrictions apply to Supplementary Tables or Videos, but we advise authors to be selective in including supplemental data.

GUIDELINES FOR EXPERIMENTAL AND STATISTICAL REPORTING

REPORTING REQUIREMENTS – We are trying to improve the quality of methods and statistics reporting in our papers. To that end, we are now asking authors to complete a reporting summary that collects information on experimental design and reagents. The Reporting Summary can be found here <https://www.nature.com/documents/nr-reporting-summary.pdf> If you would like to reference the guidance text as you complete the template, please access these flattened versions at <http://www.nature.com/authors/policies/availability.html>.

Author Rebuttal to Initial comments

We thank the Reviewers for their valuable input. During the revision period we have extensively addressed all Reviewer comments. To summarize briefly:

- we addressed the mechanism of FOXO1 function by adding extensive new datasets that comprise of single-embryo mass spectrometry and fluorescence imaging. These data show that FOXO1 function is necessary to divert the cellular metabolism to FAO instead of glycolysis, oxphos, and autophagy,*
- we more closely investigated mTOR pathway activity by staining specific downstream targets,*
- we corroborated the chromatin reorganization aspect by amending single-nucleus spatial analyses,*
- we addressed the concerns regarding the use of TSCs by performing comparative immunofluorescence and metabolism analysis in stem cells and embryos,*
- we addressed the concerns about the reactivation and developmental potential of the carnitine-supplemented embryos by efficient derivation of several ESCs and TSCs from day 12+c embryos.*
- and finally, we addressed all other concerns regarding clarity in text and figures through appropriate revisions of these.*

In addition to these revisions addressing the raised issues, we also extended the paper to include an exciting new dataset suggesting that FAO/FOXO1 may be a general component of stem cell dormancy not only in embryonic but also adult tissues (Fig. 7). For this we analyzed publicly available high-quality RNA-seq datasets from multiple adult tissues containing direct comparisons of quiescent and active stem cell transcriptomes. We identified a small set of 295 genes that are commonly upregulated in multiple dormant/quiescent cells. Among these, we observed a high enrichment of lipid metabolism and lysosome-related genes, including FOXO1. We believe that this new analysis provides a gateway linking our findings to regulation of dormancy in general and thus present it as the last figure.

We believe that the revision experiments have significantly improved the quality of the manuscript and substantiated FOXO1's significance for dormancy in embryos and potential for pan-dormancy regulation in adult tissues. For this reason we revised the title to 'FOXO1-mediated lipid metabolism maintains mammalian embryos in dormancy'. We do hope that the Reviewers' concerns are all addressed and that the paper is now ready for publication.

Reviewer #1:

Remarks to the Author:

Comments:

In the previous study, the authors found that inhibition of mTOR (mTORi) induces mouse blastocyst pausing ex vivo, mimicking embryonic diapause in mammals. However, the ex vivo paused embryos collapse over time. In the study of "Metabolic enhancement of mammalian developmental pausing" by Weijden et al., authors performed proteomics studies to identify pathways downstream of mTORi that may extend the blastocyst pause duration. They hypothesized that since TSCs do not enter dormancy after mTOR inhibition, the differential proteomic changes in ESC vs TSC will point to essential pathways for durable dormancy. They reported that not surprisingly, ESCs and TSCs had divergent response to mTORi inhibition. Among them, lipid and amino acid degradation pathways were among multiple pathways in ESCs. Authors further showed that enhancing lipid usage via supplementing fatty acid oxidation (FAO) metabolite L-carnitine prolongs mTORi-induced embryo pausing ex vivo up to 34 days and this is mediated by FOXO1. While the manuscript addresses an important question with significance in developmental biology, there are several major concerns that need be addressed:

Major concerns:

1. Authors reasoned that since ESCs and TSCs have divergent response to mTORi, they can be used as surrogates to identify essential pathways that specifically regulate the survival of mTORi-induced dormancy. This rationale is somewhat flawed since ESC and TSC are very different both epigenetically and metabolically and their cultural conditions are also different. For example, why ESC vs. TSC comparison is more meaningful than ESC vs. MEF? MEFs also don't enter dormancy upon mTORi inhibition.

The rationale of comparing the response of ESCs vs. TSCs is rooted in our previous findings that the inner cell mass (ICM) and trophoectoderm (TE) of mTORi-paused embryos show different sensitivity to mTORi treatment (Bulut-Karslioglu et al. 2016). As can also be observed in Fig. 5A of this manuscript, the ICM reacts strongly to mTORi and reduces genomic activity (readout: H4K16ac staining). In contrast, the TE of mTORi-only embryos does not successfully shut down completely. We reasoned that this differential sensitivity and inefficient shutdown of TE may underlie the deterioration and collapse of mTORi-paused embryos in vitro. To find out the pathways that are efficiently used by pluripotent cells and potentially not by the TE, we profiled and compared proteome profiles of ESCs and TSCs, which are in vitro derivatives of the ICM/epiblast and the TE. While we agree that it would be better to directly do this comparison in the embryo, this tissue-specific readout would require the use of a large number of embryos. In addition, we did not have the low-input mass spectrometry technology that we now used for single-embryo proteomics profiling at the time when we started the project. Overall, our embryo experiments validate the predictions that we initially made by using the ESC/TSC proteome data. Therefore, although the Reviewer is right that ESCs and TSCs are epigenetically and metabolically

different, we find that their response to mTORi reflects their tissue of origin in the embryo. As MEFs have a much later developmental origin and do not represent the tissues found in the blastocyst, we do not think this would be an appropriate in vitro model to understand the cell type specific regulation of dormancy at the blastocyst stage. We have now clarified our rationale in the text (line 92): “To isolate ESC-specific pathways that may allow their successful entry into pause and that can be used to improve TSC/TE pausing, we performed a quantitative, time-resolved analysis of global ESC and TSC proteomes during the cellular transition in and out of pausing (denoted pause and release from here onwards respectively, Fig. 1A).”

3. Related to point 1, previous publications indicated that TE contributes to entry and release from embryo diapause. Although TSCs do not enter into dormancy upon mTORi inhibition in vitro, it does not mean they play no roles in embryos diapause. Multiple pathways, e.g. Hippo, were perturbed in TSCs after mTORi treatment, their functions in embryo diapause or their extension should be tested or at least discussed to have a more comprehensive understanding of diapause regulation.

The TE definitely plays a role in natural diapause. Kamemizu and Fujimori have previously shown the distinct dynamics of the ICM and TE both during entry into and exit from diapause of mouse embryos (Kamemizu and Fujimori 2019). Our main aim here was to improve our mTORi-induced in vitro diapause system. The importance of TE function is indeed reflected in our results, since we show that balancing TE metabolism results in overall better dormancy in the embryo and a longer period of in vitro diapause. In the future, it will be an exciting possibility to address the crosstalk between the ICM and TE, as we now mention in the discussion: “We propose that a concerted communication between the Epi and TE may result in active chromatin rewiring, followed by a metabolic shift to FAO, which is critical in the establishment of dormancy.”

2. Through proteomics analysis coupled with computational simulation, they identified lipid degradation processes (e.g., FAO) as one of the adaptive pathways to support the mTORi paused state. The proteomics results should be validated. The importance of FAO activity should be functionally tested in ESCs. The conclusion that FAO is likely the main energy pathway in paused pluripotent cells (line 171-172) is not well supported without functional testing using FAO inhibitors, or FAO inhibitors in conjunction with mTOR inhibitors. The authors should also perform Seahorse analysis or FAO rate assay after FAO/mTORi inhibition in ESCs. The results from ESCs need to be compared with the phenotype observed in embryos. In addition, does extensive chromatin reorganization also occur in paused ESCs, just like in paused embryos? A parallel study between ESCs and embryos will greatly strengthen the

claim and rule out the embryo phenotype is exclusively from changes in epiblast.

We thank the Reviewer for these suggestions, which improved the stem cell to embryo comparisons. In the first version of the paper, we have included Seahorse ECAR and OCR assays that showed a significant reduction in the glycolysis and OxPhos capacity of paused ESCs compared to normal counterparts (Fig. S4A). Below we show the actual readouts of these experiments as well, which show the drastic reduction in both ECAR and OCR activities. As the reviewer requested, we have now performed a FAO rate assay as well (new Fig. S4A). In this assay, the FAOBlue substrate is converted through fatty acid oxidation in a product that is excitable at 405 nm. The results first of all show that TSCs show a 10-fold lower FAO level compared to ESCs, possibly explaining the significant enhancement that carnitine supplementation provides. The FAO level of paused ESCs is significantly higher than both proliferating and mTORi-treated TSCs, even though it is lower than normal ESCs. Importantly, FAO rate decreases when CPT1 is inhibited in paused ESCs, showing that active FAO takes place. In contrast, the FAO rate in paused TSCs is extremely low (background level) and is not sensitive to CPT1 inhibition. These results show that paused ESCs do perform FAO at an appreciable rate, while TSCs do not.

To further test the relative usage of fatty acids and glucose in paused ESCs, we now also performed a Seahorse Mito Fuel Flex test. In this test, cells are challenged with inhibitors against the three major major mitochondrial fuels pyruvate, glutamate, and fatty acids and their utilization of and dependency on each source is measured. This experiment showed a clearly increased dependence of paused ESCs on fatty acids compared to glucose, establishing FAO as the main energy pathway (fatty acid dependency: on average 6.5% in normal vs. 47% in paused ESCs; glucose dependency: 14.5% in normal vs 14% in pause, Fig. S4B).

Regarding the chromatin reorganization, as the Reviewer suggested, we now performed H3K9me2 and Lamin B1 stainings also in ESCs and TSCs (see new Fig. S3B). In addition, we quantified these signals in single nuclei of ICM/TE and ESCs/TSCs to be able to more thoroughly show spatial chromatin reorganization within the nucleus and compare stem cells to embryos (new. Fig 2H and S3C). We do show that both ICM cells and ESCs show extensive accumulation of H3K9me2 at the nuclear borders and this signal overlaps with Lamin B1, suggesting heterochromatin formation at the nuclear envelope. These results rule out the possibility that the embryo changes are exclusively due to changes in the epiblast, and also show that the ESC response reflects its tissue of origin. In contrast to ICM/ESCs, we do not observe this extensive chromatin reorganization in the TE/TSCs, validating our proteomics results.

2. One major concern is that while paused embryos by mTORi + L-carnitine treatment had prolonged survival (~ 34 days as claimed), there is no functional testing of these embryos to demonstrate that they are developmentally competent, healthy and able to give birth to live animals with no defects. It is very important to test embryos that survived beyond 10 days when most mTORi treated embryos died (Fig 3E). Since extension of survival by mTORi is the main point of the paper, this is essential to support the main conclusions.

In the first version of this paper, we have extensively addressed the reactivation potential of the paused embryos including the quantification of tissue-specific cell numbers. These experiments showed that the cells do in principle reactivate and express tissue-appropriate marker genes.

To further test the reversibility of pausing, we used embryos that survived beyond 10 days to derive ESCs and TSCs (day 12+c embryos, Fig. 5G). Carnitine-supplemented embryos gave rise to both ESCs and TSCs at similar or higher efficiency compared to normal E3.5 blastocysts (45% of day 12+c embryos vs 47% of E3.5 embryos gave rise to ESCs; 40% of day 12+c embryos vs 27% of E3.5 embryos gave rise to TSCs). These results indicate that carnitine-supplemented embryos are able to reactivate and proliferate and that stemness of embryonic and extraembryonic cells are not affected.

To further corroborate the developmental competence, we did attempt numerous embryo transfers. However, we faced systematic problems with the further development of not only +c embryos but also the mTORi-only embryos that previously efficiently gave rise to live, fertile mice (Bulut-Karslioglu et al 2016). We believe that either the (home-made) culture media or the batch of mTOR inhibitor has impurities that affect embryonic development, even though the blastocyst maintenance is not affected. The reduced KSOM is critical for facilitation by carnitine, therefore we cannot replace it with commercial media. Despite these problems, we consistently got implantation sites (see Figure) showing that the embryos reactivate and implant efficiently. So far, however we were able to generate only 9 live mice from mTORi-only (day 4) embryos and only 1 mouse from day 12+c embryos. Even though this, in principle, shows that carnitine-supplementation allows further development, we refrained from using these data in the paper as we believe that the transfers are overall inefficient due to the mentioned problems. In the revised paper, we explicitly bring this point up in the Discussion to draw attention to the need for further testing of developmental competence despite highly efficient ESC/TSC generation. In addition, we now changed the title of the manuscript to 'FOXO1-mediated lipid metabolism maintains mammalian embryos in dormancy' to stress the mechanistic part of dormancy regulation.

4. The previous study by Khoa et al, cited in the paper, showed that FAO inhibition leads to ESC pausing in vitro, which can be rescued by FAO metabolites. This is consistent with down regulation of FAO pathways in diapaused embryos. How does this reconcile with the results here that mTORi inhibition led to increase of FAO activity? Does that mean mTORi induced ESC dormancy is different from naturally occurring embryonic diapause in animals? A more thorough discussion may be warranted.

We thank the Reviewer for prompting us to clarify this issue. Khoa et al propose that Mof deletion induces a quiescent state in ESCs and that FAO inhibition pauses embryos for 5 days. The semi-quiescent state of Mof KO ESCs only happens in 2i culture conditions and not serum/LIF, which suggests that it is not an upstream regulator of pausing as mTOR (which pauses cells in both conditions). Furthermore, the Mof KO cells show only moderate reductions in transcription, translation, and histone acetylation (indeed only the Mof target H4K16ac is depleted), which argues against a stable dormant/quiescent state. Finally, FAOi 'only' pauses embryos for 5 days, which is significantly shorter than our optimized system here. The authors do not show effective downregulation of FAO pathways in diapaused embryos, since this conclusion is solely based on transcriptome data. We believe that FAOi by Khoa et al lowered the basal metabolic rate in blastocysts, thereby initially inducing a diapause-like state, which was however not

stable beyond 5 days. In our system, inhibition of mTOR greatly lowers the glycolysis and oxphos levels in cells (see point 2 above), and the cells and embryos switch to the use of lipids. The depletion of lipid droplets in *in vivo* diapaused embryos has been shown recently (Arena et al. 2021), which suggests that mTORi-diapause is regulated similarly to *in vivo* diapause.

In addition, we would like to note that we added the new Fig. 7 to the paper to probe the significance of our findings in dormant cells in adult tissues. We downloaded publicly available datasets from multiple tissues and reanalyzed these to investigate whether FAO could play an important role also in dormancy in adult cells. We identified a set of 295 commonly upregulated genes in dormant adult cells. Strikingly, FOXO1 as well as several lysosomal and FAO genes are upregulated in multiple tissues in the dormant state (Fig. 7B). Lipid metabolism has previously been shown to play important roles in maintaining stem cells (Knobloch et al. 2017; Yue et al. 2022; Dong et al. 2021; Ito et al. 2012; Giger et al. 2020), and we propose here that FOXO1-mediated FAO may be part of a pan-dormancy signature. We have now expanded the Discussion to elaborate on these points.

5. Why D15 mTORi + Carnitine embryos were used in Figure 5, but not later day embryos (e.g. day 20 or 30)? Although embryo survival decreased over time, choosing longer time point would further strengthen the arguments.

We used D15 mTORi + carnitine embryos, as this represents the median survival for this condition and allows us to robustly probe the temporal dynamics of pausing without wasting a large number of embryos. We have clarified this in line 302: “The ICM shows a more dynamic response compared to TE, with gradual reduction in H4K16ac until the median survival at D8 of mTORi and D15 of mTORi+carnitine embryos, respectively (Fig. 5C-D)”. Since H4K16ac is practically absent in D15 mTORi+c embryos (Fig. 5B), which are also metabolically very silent (Fig. 5E), we worked with the assumption that dormancy is stabilized. In addition, embryos paused for longer periods of time significantly downregulate OCT4 expression, making it impossible to investigate the tissue-specificity.

6. The authors identified FOXO1 as a critical regulator for Carnitine functions in mTORi paused embryos. Does addition of Carnitine affect FOXO1 expression? Is there a feedback regulation that depends on mTOR inhibition? This should be tested.

We thank the reviewer for the suggestion. We quantified FOXO1 expression in embryos cultured *in vitro* for two days, starting from E3.5, in either normal medium or medium supplemented with carnitine. Our stainings and quantifications show that carnitine supplementation does not affect FOXO1 expression. Interestingly, FOXO1 inhibition increases CPT1 expression (Fig. S7D), suggesting a feedback regulation between these two factors and corroborating FOXO1's involvement in FOA. In the manuscript, we emphasize on the link between FOXO1 and its involvement in CPT1A expression.

Minor concerns:

1. Figure legends lack essential information. Labels in the Figures are sometimes hard to follow. For examples, line 99-101 mentioned about parental ESCs in Fig. 1C, but it is difficult to find where they are. Fig. S1B, gene or protein expression?

We apologize for the unclarity. We now clarified this in the text: “Interestingly, the pause-release samples show less variability than those at the beginning of the entry into pausing, suggesting that paused pluripotent cells may have more uniform pluripotency characteristics (Fig. 1C)”.

Fig. S1B shows protein expression, which is now corrected on the Figure.

We also expanded the other figure legends and hope that they are clearer now.

2. Line 147-148, difference of H3K9me2 in EPI and TE needs to be better explained. Quantifications of Fig. 1F, G should be added.

We now added quantifications of H3K9me2 in the EPI and TE in single-nucleus cross-sections and compared this additionally to the pattern observed in ESCs and TSCs (Figs. 2H-I, S3B-C). In paused epiblast and ESCs, we see an increase in H3K9me2 close to the nuclear lamina, indicative of extensive changes in the spatial chromatin landscape and heterochromatin formation. In the TE and TSCs, there is a tendency of H3K9me2 increase, but not exclusively close to nuclear lamina. We now elaborated these points in the text, which reads as follows: “The heterochromatin mark H3K9me2 is differentially enriched in the epiblast compared to the TE in diapause, highlighting the tissue-specific chromatin reorganization (Fig. 2F-G). Quantification of H3K9me2 and Lamin B1 intensity in single-nucleus cross-sections showed that the H3K9me2 specifically accumulates at the nuclear envelope in epiblast but not TE cells (Fig. 2H-I), corroborating the heterochromatinization observed in electron microscopy images. Paused ESCs show similar H3K9me2 accumulation at the nuclear envelope and recapitulate the chromatin reorganization seen in the diapaused embryo, while TSCs show less specificity (Fig. S3B-C)”.

Reviewer #2:

Remarks to the Author:

The manuscript authored by V. van der Weijden describes experiments to characterise the mechanisms by which inhibition of the mTor pathway in stem cells and embryos induces proliferative quiescence.

Based on comparison of paused ICM and trophoblast derived stem cells, lipid metabolism pathways were identified as potential factors in the maintenance of the mitotically inactive state. The most interesting aspect of the manuscript is the extension of embryonic diapause and the viability of embryos in this state by addition of carnitine to enhance fatty acid oxidation. The approach is appropriate and valid, and the data are robust. Thus, manuscript provides timely new insight into the phenomenon of arrest in cell differentiation essential to understanding embryonic stem cells and embryonic diapause. Although the manuscript has been well-prepared, there are some issues that could be addressed.

1. It is stated that the real time and pseudotime results depicted in Figure 1C largely agree. It might be useful to plot these data in a similar way so they can be readily compared between methods of analysis. What might be the physiological significance of pseudotime in this context?

We apologize for the lack in clarity and tried to resolve this issue by indicating what is plotted in the figure legend. On the x-axis, we have the computed pseudotime, while the y-axis indicates the biological time (culture time in hours (+1 for imputation)). The black dotted line indicates the regression curve. In Fig. 1C and D, we see that the pseudotime and biological time correlate for ESCs, but not TSCs. As the pseudotime orders samples according to the similarity level of their proteomes, we believe that a high correlation between pseudotime and biological time indicates progressive changes to the proteome and thus a concerted dormancy trajectory. The corresponding section of the main text now reads: "To further investigate cell type-specific proteome dynamics, we constructed a pseudotime trajectory using the time-series proteome datasets. The pseudotime analysis also revealed distinct trajectories of ESCs and TSCs during pause and release (Fig. 1C, D). For ESCs, pseudotime and biological time largely correlate during pause and release (Fig. 1C), indicating progressive changes in the proteome."

2. Figure 2A is used to describe immediate vs adaptive response to mTor inhibition. The differences between the two conditions are not abundantly obvious, as, in some cases the responses are quite similar.

We agree that some patterns are similar, yet our rationale was to distinguish the molecular responses that happen immediately upon mTORi and that persist over the 48-hour pausing period from those that develop at later time points. For this reason, we named them immediate and adaptive responses. To clarify this, we have now amended the text: "We categorized the clusters as immediate (unidirectional response over time) or adaptive (changing response over time) based on the temporal expression patterns"

3. A theme that is repeated in the description of these results is that mTor treatment results in chromatin reorganization. Speculation in lines 151-153 reiterates this view, but without supporting data. Further explanation of the data and/or the reasoning behind this concept would be useful.

Our proteomics data suggest that ESCs undergo chromatin organization as immediate and adaptive response to mTORi, while TSCs do not. To support this point, we now stained normal and mTORi-treated ESCs and TSCs against H3K9me2 and Lamin B1 in addition to the existing stainings of the same in embryos (Fig. S3B, 2F-G). To directly address the 'chromatin reorganization' part, we quantified H3K9me2 in the EPI and TE in single-nucleus cross-sections and compared this additionally to the pattern observed in ESCs and TSCs (Figs. 2H-I, S3B-C). In paused epiblast and ESCs, we see an

increase in H3K9me2 close to the nuclear lamina, indicative of extensive changes in the spatial chromatin landscape and heterochromatin formation. In the TE and TSCs, there is a tendency of H3K9me2 increase, but not exclusively close to nuclear lamina. We now elaborated these points in the text, which reads as follows: “The heterochromatin mark H3K9me2 is differentially enriched in the epiblast compared to the TE in diapause, highlighting the tissue-specific chromatin reorganization (Fig. 2F-G). Quantification of H3K9me2 and Lamin B1 intensity in single-nucleus cross-sections showed that the H3K9me2 specifically accumulates at the nuclear envelope in epiblast but not TE cells (Fig. 2H-I), corroborating the heterochromatinization observed in electron microscopy images. Paused ESCs show similar H3K9me2 accumulation at the nuclear envelope and recapitulate the chromatin reorganization seen in the diapaused embryo, while TSCs show less specificity (Fig. S3B-C).”

4. The description of Figure 2E in the legend is not adequate, and it is not clear from the figure what the image is showing.

We apologize for the inadequate figure legend and now improved this: “(E) Transmission electron microscopy images of selected areas with one or more nuclei in the epiblast and TE of E4.5 and diapaused embryos. The nucleus is denoted with “N” and visible as the area with an electron dense periphery, while the nucleolus is denoted with “n”. Scale bar: 2 μ m.”

5. Figure 3A-B: Heat maps of gene expression are presented. It is not clear from the legend whether these are proteomic or some other sort of data. It’s a minor point.

We improved the figure legends to clarify which data are presented. The heatmap in Figure 3A, shows metabolite abundances, while that in Figure 3B shows protein expression levels. The figure legend now reads as follows: “(A) Differentially enriched metabolites detected by LC-MS/MS (p -value <0.05 and absolute $\log_2FC >0.75$) in paused (7 days) vs normal ESCs. Mean-centered data of three biological replicates are shown.

(B) Expression levels of differentially expressed fatty acid oxidation and glycolysis-related proteins in ESCs. Mean-centered protein expression of three biological replicates is shown.”

6. As noted above, the exciting aspect of this study is that carnitine supplementation can maintain diapause. What is the source of the carnitine regulating diapause in embryos that in this state in vivo, or in vitro? In the same vein, line 190 alludes to the idea that the source of lipids that maintain diapause is intracellular. Is this speculation, or are there data to support this proposition?

Carnitine is one of the metabolites in the uterine fluid, which is abundantly present in the in vivo microenvironment of diapaused embryos (e.g., in the European roe deer (van der Weijden et al. 2021)). In addition to uptake from the outside, cells can also synthesize carnitine via degradation of branched-chain amino acids (BCAA). Supplementation of embryos with BCAAs, and particularly valine and isoleucine also enhance in vitro pausing, albeit at much lower levels compared to direct carnitine supplementation (Fig. 3E). This suggests that carnitine is mainly taken from outside the cells. In vitro, embryo culture medium does not contain carnitine and thus we supplement the reduced KSOM with 1 mM L-carnitine. Inhibition of fatty acid uptake SLC27A1 inhibition and synthesis via FASN inhibition did not alter maximum survival (Fig. 3C and E) in our system. In fact, SLC27A1 inhibition enhances pausing

efficiency (median duration), likely by promoting the degradation of intracellular lipids. Likewise, supplementing the culture with short chain free fatty acids also did not enhance pausing (Fig. S4C), thus we deduce that only stored cellular lipids are used in paused cells.

7. Similarly, data in Figure 4F seem to indicate that there is a greater number of lipid droplets prior to mTOR induced pause than after. Pause is characterized by FA degradation (Figure 1F). Is there new uptake of lipids associated with the state of cellular pause?

Since direct supplementation of embryos with short chain free fatty acids does not enhance pausing and the number of LDs decrease over time, we deduce that only stored cellular lipids are used in paused cells. However, there is clearly carnitine uptake (due to enhanced survival with carnitine supplementation). Our data also show that carnitine-coupled fatty acids can be taken up (enhancement via adipoyl-L-carnitine, Fig. S4D). Therefore, an exciting direction for us in the future is to further enhance in vitro pausing by supplementing the paused embryos with a renewable and usable exogenous lipid source.

8. A percentage of the epiblast structures form rosettes, indicative of the onset of epithelial polarity. This is an interesting finding that does not receive much discussion beyond noting that carnitine supplementation appears to increase the frequency of rosette formation.

It has been recently shown that a subset of diapaused blastocysts form rosettes and that this is mediated via the WNT pathway (Fan et al. 2020). Other groups have shown that lipid consumption leads to a naive-to-primed pluripotency transition (Sperber et al. 2015; Cornacchia et al. 2019). In the context of our work, we mentioned the increase in rosette formation and raise the possibility that the earlier onset and high frequency of polarization in carnitine supplemented embryos at D3 points to a direct link between fatty acid oxidation and WNT pathway activity. This link, to our knowledge, has not been shown before in the context of pluripotency regulation. Furthermore, our results suggest that the carnitine supplemented paused embryos may better represent the morphological properties of in vivo diapaused embryos. We address these points in the discussion of the manuscript. However, since tissue morphology and polarization are not the central focus of this paper, we did not pursue the link between FAO and morphology experimentally.

9. It is stated that Foxo1 plays a critical role in cellular dormancy (lines 286 et seq.). There is some speculation about its role in the discussion, but, overall no mechanism is proposed.

We thank the Reviewer for prompting us to address the mechanism of FOXO1 function in more detail. We have now addressed the role of FOXO1 in relation to FAO with new experiments. For this we performed a single-embryo proteomics experiment as well as IF stainings to compare the responses of mTORi-only, mTOR+carnitine, mTORi+carnitine+FOXO1i, mTORi+FOXO1i, and mTORi+CPT1i embryos. This extensive experiment allowed us to dissect the pathways that are dependent on FOXO1, CPT1/FOA, and both FOXO1 and CPT1 (Fig. 6C). We show that FOXO1 and CPT1 together regulate lysosomal activity (likely lysosomal lipid catabolism), and that FOXO1 promotes autophagy independent of CPT1. Our results suggest that FOXO1 inhibition derepresses glycolysis and OxPhos and that the embryos do not utilize FOA when FOXO1 is inhibited (Fig. 6D). Upregulation of CPT1 in response to FOXO1 inhibition (Fig. S7D) indicate a feedback mechanism between these two factors (we interpret the upregulation as

compensation for low FOA output in FOXO1i). As a model, we propose that carnitine enhances developmental pausing through mitochondrial fatty acid oxidation and that this is dependent on FOXO1-mediated lysosomal function.

In addition, we would like to note that we added the new Fig. 7 to the paper to probe the significance of our findings in dormant cells in adult tissues. We downloaded publicly available datasets from multiple tissues and reanalyzed these to investigate whether FAO could also play an important role in adult stem cells. We identified a set of 295 commonly upregulated genes in dormant adult cells. Strikingly, FOXO1 as well as several lysosomal and FAO genes are upregulated in multiple tissues in the dormant state (Fig. 7B). Lipid metabolism has previously been shown to play important roles in maintaining stem cells (Knobloch et al. 2017; Yue et al. 2022; Dong et al. 2021; Ito et al. 2012; Giger et al. 2020), and we propose here that FOXO1-mediated FAO may be part of a pan-dormancy signature. We have now expanded the Discussion to elaborate on these points.

10. Embryos treated with mTor and carnitine persisted longer in culture. The important question relates to whether the embryos could continue development after washout of these two factors. It would see from Figure 5B and the text in lines 253 et seq. that recovery of development was possible, but that it was substantially slower in those embryos that were in the paused state for the longer times. Is there any evidence relevant to the mechanism of differential temporal recovery?

Temporal dynamics of the reactivation of *in vivo* diapause embryos has previously been studied by Kamemizu and Fujimori, 2019 (Kamemizu and Fujimori 2019). Here, the authors retransferred embryos that have been diapause for 1 or 7 days and staged the development of resulting fetuses based on the development of limbs. Embryos paused for 7 days showed a delay of embryonic development of about 0.5 to 1.0 day compared with the case of embryos paused for 1 day. This is in line with their findings that embryos paused for 7 days need 13 hours to re-enter the cell cycle, compared to only 5.4 hours in embryos paused for 1 day. These results show that *in vivo* diapause embryos, just like our mTORi+carnitine embryos, take longer to reactivate as the dormancy period gets longer. The mechanism however is not resolved and we can only speculate that the embryos in deeper dormancy would reactivate later than semi-dormant ones because resuming full capacity RNA and protein production and replenishing protein pools will require more time.

To further test the reversibility of pausing, we used embryos that survived beyond 10 days to derive ESCs and TSCs (day 12+c embryos, Fig. 5G). Carnitine-supplemented embryos gave rise to both ESCs and TSCs at similar or higher efficiency compared to normal E3.5 blastocysts (45% of day 12+c embryos vs 47% of E3.5 embryos gave rise to ESCs; 40% of day 12+c embryos vs 27% of E3.5 embryos gave rise to TSCs). These results indicate that carnitine-supplemented embryos are able to reactivate and proliferate and that stemness of embryonic and extraembryonic cells are not affected.

11. There are constraints on length of the text for this journal. Given the exciting nature of the results, i.e. that diapause can be further extended, and that a clear mechanism has been identified in carnitine, the

discussion seems not to do justice to the data. Perhaps it could be rewritten with strong focus on mechanisms and significance of the data, and a bit less speculation.

We appreciate the Reviewer's feedback. We now revised the Discussion in this direction. We do however find the future implications exciting and for this reason still mention the larger implications in the current Discussion.

////////////////////////////////////

Reviewer #3:
Remarks to the Author:

In this study, the authors performed proteomic and metabolomic analyses of ESCs and TSCs to clarify the mechanism of maintaining dormancy of blastocyst-derived ESCs *in vitro* by mTOR inhibition, and found significant differences in the properties of ESCs and TSCs. In particular, they demonstrated that the metabolism of lipids and amino acids is reduced in dormant ESCs, thus, long-term and stable dormancy is possible by compensating for these functions. Furthermore, the authors showed that Foxo1-mediated lipid metabolism affects dormancy. It is particularly interesting that there is such a difference in the properties of ESCs and TSCs derived from the same blastocyst and that mTOR inhibition can maintain the integrity of ESCs for a long time *in vitro*. Overall, the authors' interpretation of the experimental results is reasonable, and most of their claims seem to be supported by the experimental results. However, for some experiments, the current data are insufficient to assess their validity. In particular, since mTOR is involved in a variety of cellular functions, it would be desirable to carefully examine the level of activation of mTOR and related signaling pathways in each experiment.

Major comments:

Since changes in mTOR activity by mTORi are key in this study, the phosphorylation levels of mTOR and its downstream factors (such as S6 and 4EBP) in ESCs and TSCs must be confirmed in each experiment. Especially in experiments with carnitine and Etmozir (Figure 3), in addition to changes in mTOR activity, phosphorylation levels of AMPK and ACC should also be examined by Western Blotting and immunostaining.

We thank the Reviewer for these suggestions, which prompted us to more carefully dissect the involved pathways. We performed the suggested experiments, which can be found in new Fig. S1 and S4). We preferred immunofluorescence stainings over western blotting because in our experience, IF on stem cell colonies gives reliable results, while disruption of colonies and cell lysis leads to inconsistencies. We show that the mTOR pathway is effectively inhibited in both ESCs and TSCs, with significant decrease in phospho-mTOR, phospho-S6 (mTORC1 target) and phospho-AKT (mTORC2 target) shown by our quantifications (new Fig. S1C-E). We performed these stainings also directly on embryos and again show the effective inhibition of mTOR pathway activity both in the ICM and TE (Fig. S4D-F). Carnitine and CPT1i (Etomoxir)-treated embryos maintain low mTOR pathway activity levels, although a slight upregulation of pAKT is seen. As suggested, we also stained for phospho-AMPK and -ACC. L-carnitine supplementation reduced phospho-AMPK in the TE (Fig. S4G). Although pAMPK is also reduced in CPT1i embryos, these do not survive more than 10 days in culture (Fig. 3E). Phosphorylation of ACC, which inhibits its activity to convert acetyl-coA to malonyl-coA, slightly increased with mTORi and mTORi in conjunction with carnitine supplementation (Fig S4H), suggesting that carnitine could enhance CPT1

activity by lifting its inhibition by malonyl-coA (McGarry, Leatherman, and Foster 1978). Overall, carnitine supplementation prolongs the duration of *in vitro* pausing largely independent of mTOR pathway activity, yet possibly in feedback with AKT, AMPK, and ACC.

The interpretation of the results in Fig. 1E is somewhat puzzling. Why would inhibition of mTOR, a negative regulator of autophagy, conversely suppress autophagy in ESCs? Is it necessary to temporarily suppress autophagy activity in order to maintain dormancy? Examining the levels of autophagy and mTOR activity over time after the addition of mTORi may provide answers to these questions. In this connection, suppression of the SNARE pathway, which is involved in the formation of autolysosomes, should also be considered.

We have previously shown the functional involvement of autophagy in mTORi-mediated pausing (Bulut-Karslioglu et al. 2016). In these experiments, blocking autophagy during mTOR inhibition compromised the efficiency of pausing. Since carnitine supplementation changes the energetics of embryos, we agree that it is worth revisiting the role of autophagy. In addition, as the Reviewer mentions below, FOXO1 is known to induce autophagy, and we show that carnitine-enhancement of in vitro pausing is dependent on FOXO1.

To understand the role of autophagy during mTORi-mediated pausing, we paused embryos and supplemented them with carnitine or treated with FOXO1i, then stained against pULK, the variant that is phosphorylated by mTOR and causes inactivation of the autophagy pathway. We found that pULK levels in the ICM are decreased in mTORi and mTORi+carnitine compared to E4.5, but increased when FOXO1 is inhibited compared to mTORi (Fig. 6E and S7D). This suggests that the autophagy pathway is active in mTORi and mTORi+c embryos, and that FOXO1i reduces the autophagy activity. Interestingly, we observed an accumulation of lysosomes (via LAMP1 staining) in the TE of mTORi-only paused embryos, which was not the case in carnitine supplemented paused embryos (Fig. 6E). As a model, we propose that carnitine enhances developmental pausing through mitochondrial fatty acid oxidation and that this is dependent on FOXO1-mediated lysosomal function.

Depending on the energy level of the cell, LDs are known to interact with various organelles. In Figure 4, it is worth analyzing whether the differences in localization with lysosomes in addition to mitochondria correlate with the size and distribution of LDs observed in epiblasts and TEs.

We found it hard to identify lysosomes in EM data. To address the Reviewer's point from a different perspective, we investigated lysosomes in relation with FAO and FOXO1 activity in embryos by LAMP1 staining as described above (Fig. 6E). We observe an accumulation of lysosomes on D5 of mTORi, which is not observed if the paused embryos are supplemented with carnitine. This indeed suggests a different energy balance and usage in normal paused embryos and carnitine supplemented paused embryos. With our single embryo proteomics approach, we show that carnitine, CPT1i and FOXO1 regulate the metabolic state of paused embryos (Fig. 6D). We now show that carnitine enhances developmental pausing through FOXO1's role in mediating lysosomal lipid catabolism and mitochondrial fatty acid oxidation. Strikingly, we find FOXO1 and lysosomal genes to be commonly upregulated in dormant adult cells (Fig. 7B). This indicates that FOXO1 may play an essential role in the metabolic regulation of cellular dormancy in both embryonic and adult stem cells.

FOXO1 is known to induce autophagy (Nat Cell Biol. 2010 12(7):665-75, PMID: 20543840), therefore it is necessary to examine whether inhibition of FOXO1 alters autophagy activity. Related to the above comment, if inhibition of FOXO1 suppresses autophagy induction, it needs to be tested whether it is transient or autophagy-independent.

FOXO inhibition indeed reduces the levels of autophagy through an increase in autophagy inhibitory phosphorylation of the autophagy inducer ULK (Fig. 6E and S7D). Given the reduced accumulation of lysosomes in FOXO1 inhibited embryos, it seems that FOXO is required for both autophagy and formation of lysosomes. As these embryos have a lower capacity to use their lipid reserves compared to the mTORi+c embryos, we speculate that this results in a faster degradation of the FOXO embryos due to a lack of energy required to sustain during dormancy.

Minor comments:

line 180: it would be more helpful to the reader if the specific name of the inhibitor is mentioned (except for mTORi). e.g., Etmoxir for CPT1 inhibitor, etc. FOXO1 inhibitor as well.

Thank you for the suggestion, we have added the specific inhibitor names throughout the text.

In Fig. 5, the authors examine H4K16ac levels, but would like an explanation as to why they focused only on this acetylation.

We have previously shown that H4K16ac shows a tissue specific downregulation in the paused embryo (Bulut-Karslioglu et al. 2016). A hallmark of dormancy is hypotranscription and hypotranslation. H4K16ac is correlated with transcriptional activity. Moreover, fatty acid oxidation provides cells with Acetyl-CoA, used for histone acetylation (McDonnell et al. 2016). Therefore, we chose to investigate H4K16ac levels to address whether lipid usage influences global histone acetylation.

line 213: The authors mention that the fixation procedure affects BODIPY staining, but in this reviewer's experience, this problem could be resolved by restaining the fixed cells with BODIPY after fixation (just before fluorescence observation).

We thank the Reviewer for this suggestion. During the optimization of our protocol, we wished to have combined an IF against an epiblast marker and the bodipy staining to allow for a tissue specific analysis. We tried the bodipy staining both prior to and after fixation but saw a vast reduction in the number of detected lipid droplets with both methods compared to the live stainings. Therefore, we decided to use live stainings, constraining the data interpretation to the embryo and cell level.

Several terms such as Thermo, Thermo Fisher, Themo, and Themo Fisher are mixed in the Materials and Methods.

We have rectified this in the revised version of the manuscript.

Line 262: D3 + C embryo should be D3 + c (lowercase).

Thank you for catching this mistake, we corrected it in the revised version of the manuscript.

References

- Arena, Roberta, Simona Bisogno, Łukasz Gąsior, Joanna Rudnicka, Laura Bernhardt, Thomas Haaf, Federica Zacchini, Michał Bochenek, Kinga Fic, Ewelina Bik, Małgorzata Barańska, Anna Bodzoń-Kułakowska, Piotr Suder, Joanna Depciuch, Artur Gurgul, Zbigniew Polański, and Grażyna E. Ptak. 2021. 'Lipid droplets in mammalian eggs are utilized during embryonic diapause', *Proceedings of the National Academy of Sciences*, 118: e2018362118.
- Bulut-Karslioglu, A., S. Biechele, H. Jin, T. A. Macrae, M. Hejna, M. Gertsenstein, J. S. Song, and M. Ramalho-Santos. 2016. 'Inhibition of mTOR induces a paused pluripotent state', *Nature*, 540: 119-23.
- Cornacchia, Daniela, Chao Zhang, Bastian Zimmer, Sun Young Chung, Yujie Fan, Mohamed A. Soliman, Jason Tchieu, Stuart M. Chambers, Hardik Shah, Daniel Paull, Csaba Konrad, Michelle Vincendeau, Scott A. Noggle, Giovanni Manfredi, Lydia W. S. Finley, Justin R. Cross, Doron Betel, and Lorenz Studer. 2019. 'Lipid Deprivation Induces a Stable, Naive-to-Primed Intermediate State of Pluripotency in Human PSCs', *Cell Stem Cell*, 25: 120-36.e10.
- Dong, Q., M. Zavortink, F. Foldi, S. Golenkina, T. Lam, and L. Y. Cheng. 2021. 'Glial Hedgehog signalling and lipid metabolism regulate neural stem cell proliferation in *Drosophila*', *EMBO Rep*, 22: e52130.
- Fan, Rui, Yung Su Kim, Jie Wu, Rui Chen, Dagmar Zeuschner, Karina Mildner, Kenjiro Adachi, Guangming Wu, Styliani Galatidou, Jianhua Li, Hans R. Schöler, Sebastian A. Leidel, and Ivan Bedzhov. 2020. 'Wnt/Beta-catenin/Esrrb signalling controls the tissue-scale reorganization and maintenance of the pluripotent lineage during murine embryonic diapause', *Nature Communications*, 11: 5499.
- Giger, Sonja, Larisa V. Kovtonyuk, Sebastian G. Utz, Mergim Ramosaj, Werner J. Kovacs, Emanuel Schmid, Vassilios Ioannidis, Melanie Greter, Markus G. Manz, Matthias P. Lutolf, Sebastian Jessberger, and Marlen Knobloch. 2020. 'A Single Metabolite which Modulates Lipid Metabolism Alters Hematopoietic Stem/Progenitor Cell Behavior and Promotes Lymphoid Reconstitution', *Stem Cell Reports*, 15: 566-76.
- Ito, Keisuke, Arkaitz Carracedo, Dror Weiss, Fumio Arai, Ugo Ala, David E. Avigan, Zachary T. Schafer, Ronald M. Evans, Toshio Suda, Chih-Hao Lee, and Pier Paolo Pandolfi. 2012. 'A PML-PPAR- δ pathway for fatty acid oxidation regulates hematopoietic stem cell maintenance', *Nature Medicine*, 18: 1350-58.
- Kamemizu, C., and T. Fujimori. 2019. 'Distinct dormancy progression depending on embryonic regions during mouse embryonic diapause', *Biol Reprod*, 100: 1204-14.
- Knobloch, M., G. A. Pilz, B. Ghesquière, W. J. Kovacs, T. Wegleiter, D. L. Moore, M. Hruzova, N. Zamboni, P. Carmeliet, and S. Jessberger. 2017. 'A Fatty Acid Oxidation-Dependent Metabolic Shift Regulates Adult Neural Stem Cell Activity', *Cell Rep*, 20: 2144-55.
- McDonnell, Eoin, Scott B. Crown, Douglas B. Fox, Betül Kitir, Olga R. Ilkayeva, Christian A. Olsen, Paul A. Grimsrud, and Matthew D. Hirschey. 2016. 'Lipids Reprogram Metabolism to Become a Major Carbon Source for Histone Acetylation', *Cell Reports*, 17: 1463-72.
- McGarry, J. D., G. F. Leatherman, and D. W. Foster. 1978. 'Carnitine palmitoyltransferase I. The site of inhibition of hepatic fatty acid oxidation by malonyl-CoA', *J Biol Chem*, 253: 4128-36.
- Sperber, Henrik, Julie Mathieu, Yuliang Wang, Amy Ferreccio, Jennifer Hesson, Zhuojin Xu, Karin A. Fischer, Ariketh Devi, Damien Detraux, Haiwei Gu, Stephanie L. Battle, Megan Showalter, Cristina Valensisi, Jason H. Bielias, Nolan G. Ericson, Lilyana Margaretha, Aaron M. Robitaille,

- Daciana Margineantu, Oliver Fiehn, David Hockenbery, C. Anthony Blau, Daniel Raftery, Adam A. Margolin, R. David Hawkins, Randall T. Moon, Carol B. Ware, and Hannele Ruohola-Baker. 2015. 'The metabolome regulates the epigenetic landscape during naive-to-primed human embryonic stem cell transition', *Nature Cell Biology*, 17: 1523-35.
- van der Weijden, V. A., J. T. Bick, S. Bauersachs, A. B. Rüegg, T. B. Hildebrandt, F. Goeritz, K. Jewgenow, P. Giesbertz, H. Daniel, E. Derisoud, P. Chavatte-Palmer, R. M. Bruckmaier, B. Drews, and S. E. Ulbrich. 2021. 'Amino acids activate mTORC1 to release roe deer embryos from decelerated proliferation during diapause', *Proc Natl Acad Sci U S A*, 118.
- Yue, Feng, Stephanie N. Oprescu, Jiamin Qiu, Lijie Gu, Lijia Zhang, Jingjuan Chen, Naagarajan Narayanan, Meng Deng, and Shihuan Kuang. 2022. 'Lipid droplet dynamics regulate adult muscle stem cell fate', *Cell Reports*, 38: 110267.

Second Author Rebuttal to Initial comments

REVIEWER #1

1. In the revised manuscript, the authors have provided additional data to address my previous concerns. However, some of my concerns, particularly the major concern #3, remain unresolved (see below).

We thank the reviewer for their time and critical evaluation of the manuscript. To specifically address the concern regarding the stemness of the derived embryonic and extraembryonic cells, we now included IF stainings for stemness markers.

2. Regarding major concern point #1, the authors did a very good job when using H4K16ac as a marker to distinguish the differential response of ICM and TEs to mTORi. Since they used ESCs and TSCs as in vitro models for ICM and TE, the expression of H4K16ac should also be tested in ESCs and TSCs treated with mTORi +/- L-Carnitine. This will strengthen the rationale of using ESCs and TSCs in the current study. Also in Fig. 5 the authors should include CDX2 to mark the TE, which is critical for data quantification. DAPI alone is not sufficient in this case.

The Reviewer correctly points out the rationale of using ESCs and TSCs as in vitro derivatives of the epiblast and TE. In the manuscript, per the Reviewer's request, we have now established the shared basis of the response of used stem cells and their tissues of origin (epiblast-ESCs and TE-TSCs) to mTORi-induced dormancy. These include:

- a) the same pattern of chromatin reorganization in the epiblast and ESCs, which differs from the TE-TSCs (Figure 2H-I, S3B-C);*
- b) Usage of FAO pathway and the response of stem cells and embryo to Cpt1 inhibition (blocking FAO, Figure 3E, S4A);*
- c) altered adhesion properties of TSCs and TEs (Figure 2C and S3D).*

We believe that these extensive experiments place ESCs and TSCs as a reliable representation of the embryo and its altered regulation during diapause entry. However, the composition of culture media, the growth rates of stem cells and embryos, as well as the complexity of intercellular regulation are drastically different, therefore we think direct carnitine supplementation is best suited to embryo culture.

Regarding the quantifications of the embryo stainings in Fig. 5 and the request to include CDX2 to mark the TE: We used OCT4 as a marker of the ICM, which includes both the epiblast and the PrE. Thus, by definition, OCT4-negative cells correspond to the TE. This is a commonly used approach in the analysis of preimplantation embryos (see e.g. Bao et al, NCB (2022)¹). In addition, it has been shown that a DNA stain (Hoechst in this case) gives the same pattern of CDX2 expression in the TE of diapaused embryos (Figure 2, Kamemizu et al, Biology of Reproduction (2019)²). We guess that the Reviewer alludes to the ICM cells transiently losing OCT4 expression in D15+c embryos, which may be miscategorized as TE cells. This is however only a very minor proportion of the 868 cells that we quantified in this experiment, and thus cannot skew the results.

3. Regarding major concern point #3 to demonstrate in vivo survival of paused embryos, the author indicated that they encountered technical problems in such that they cannot reproduce their previous published results for mTORi-treated embryos (Bulut-Karslioglu et al. 2016), which is very problematic. Authors cited the previous work to indicate developmental competency of mTORi induced development diapause. Unable to reproduce this finding raises the question for the foundation of current work. The Figure showed in the rebuttal letter did not adequately indicate that the embryos can reactivate and implant efficiently. It is not shown whether there are embryos inside the decidua and whether the embryos have developed appropriately. The inability of mTORi+c embryos to efficiently give rise healthy mice (compared to mTORi alone) dramatically reduces the significance, as well as the conceptual advance of the current manuscript.

We would like to note that the previous findings are reproduced. As we mentioned in the previous point-by-point response, we were able to generate 9 live-born, healthy pups from mTORi-only embryos (see Figure below). As also mentioned, we have used home-made reduced KSOM media

for the experiments in this manuscript. Even though this media is meticulously prepared, it may inevitably not reach the commercial standards of mouse and human embryo culture media. Even with commercial media, it has been shown that mouse embryos isolated directly from the uterus perform superior to embryos cultured *in vitro* in normal embryo culture conditions (no mTORi) after embryo transfer, underlying the inefficiency introduced by *in vitro* culture (see Figure 2, Schwarzer et al, Human Reproduction (2012)³). Furthermore, we

believe that embryo reactivation conditions would need to be optimized for embryos supplemented with carnitine, as we have shown that they reactivate later than mTORi-only embryos (Figure 5A). Such an optimization requires the use of a large number of animals. Due to these reasons, and even though initial experiments produced one pup from D12+c embryos; we found it more appropriate to derive several stem cell lines from D12+c embryos (Figure 5G). As we have shown in the revised manuscript, the ESC derivation efficiency of D12+c embryos is equivalent to normal blastocysts and the efficiency of TSC derivation from D12+c embryos exceeds that of normal blastocysts. Since the mainly impacted tissue in carnitine-supplemented embryos is the TE, this dataset lends support to the idea that the reactivation capacity of this tissue is not deteriorated.

In the revision, authors argued that mTORi+L-carnitine-treated embryos can successfully derive ESCs and TSCs, comparable to mTORi-treated embryos. However, the ES derivation efficiency shown in Fig. 5G is very low (45% for control embryos), as compared to published studies for normal E3.5 embryos (often in the range of 80-90%). The ESC derivation was only 47% for 12+ embryos. Notably, the ESC derivation efficiency from mTORi-treated embryos was >50% in Bulut-Karslioglu et al. 2016. The lower efficiency of D12+c embryos to give rise to ESC does not support mTORi+L-carnitine extend during of healthy and normal embryos. In addition, the authors had overstated when saying '...that stemness of embryonic and extraembryonic cells are not affected'. Without testing respective stemness markers for derived ESC and TSC and set up different lineage differentiation assays, this statement is invalid.

ESC derivation efficiencies vary based on the strain^{4,5}, blastocyst production method⁶, and culture conditions^{4,7,8}. For derivation of ESCs and TSCs, we here used b6d2F1 x b6casF1 embryos that were generated via IVF. These embryos were retrieved from frozen stocks at the 2C-stage, thawed, cultured to blastocyst stage and afterwards used for ESC/TSC derivation. The combination of IVF and freeze-thaw do result in lower efficiency of stem cell derivation, however we generated several independent lines.

We note that ESC derivation efficiency from D12+c embryos is not lower than the side-by-side E3.5 control here (47% vs 45%). We also note that the TSC derivation efficiency from D12+c embryos is higher than that of E3.5 controls (40% vs 27%). Since the mainly impacted tissue in carnitine-supplemented embryos is the TE, this dataset lends support to the idea that the reactivation capacity of this tissue is not deteriorated. Successful implantation of these embryos, as mentioned in the previous point-by-point response, corroborates this point.

The Reviewer correctly points out that stemness markers need to be tested. To address this point, we now provide OCT4 and CDX2 stainings for the derived ESCs and TSCs, respectively, along with corresponding single-nucleus quantifications (new Figure 5H and below). In addition we also provide alkaline phosphatase assay for ESCs. We show that ESCs derived from E3.5 and D12+c embryos are both alkaline phosphatase-positive and have similar OCT4 expression levels (Figure 5H). TSCs derived from D12+c embryos have significantly higher CDX2 expression compared to E3.5 controls (Figure 5H). Therefore, we conclude that the stemness markers of embryonic and extraembryonic tissues are not affected and even improved for TSCs in our system.

Regarding my concern point #6, expression of CPT1A (Fig. S7D) is very unclear. Where is this protein expressed? For example, in c+ embryo, I cannot clearly see the CPT1 signals in the ICM (marked by SOX2), but it was shown to increase in quantification (compared to the D5 mTORi)? CPT1A localization needs to be clearly demonstrated.

We had to compress the file for uploading, which led to a lower quality image and this may have made the staining pattern unclear for the Reviewer. Here we provide a single stack showing CPT1A staining. The protein is cytoplasmic as expected and is expressed both in the ICM and the TE. Our image quantifications are done in an unbiased way via CellProfiler and then normalized to object area, therefore we deem them more reliable compared to observation by eye. We now further clarified the CPT1 expression pattern in the text, which reads as follows: "Moreover, FOXO1i-treated embryos retain cytoplasmic CPT1 expression in both the ICM and TE (Fig. S7D), likely as a compensatory mechanism as they cannot switch to FAO."

Finally, the new Figure 7 showed large variations of FAO expression in different tissues. The pathway analyses can be skewed by the huge up regulation in hepatic tissues, for example, which have abnormally high FAO gene expression. It needs to be careful to draw overly broad conclusions based on RNA datasets alone.

We would like to note that the 'common up' genes shown in Figure 7B are significantly differentially expressed in at least 4 out of 6 tissues. Thus, by design, these genes cannot be skewed by hepatic tissues. Genes that are significantly differentially expressed only in the hepatic tissue are annotated as 'hepatic-enriched' in Figure 7B and are not included in the pathway analysis.

REVIEWER #2

I believe that the authors have successfully addressed my previous concerns. The manuscript is clearly original, and the methods are appropriate. The results are credible. I particularly liked the new information that the arrested embryos were capable of implantation into the uterus. I believe it to be a solid contribution to the understanding of the enigma of embryonic diapause.

We thank the reviewer for their positive feedback and helping us improve our manuscript.

REVIEWER #3

The authors provided thoughtful responses to the review with revisions that included new data and requested analyses. These changes made the manuscript even better and more impactful.

We thank the reviewer for their positive feedback and helping us improve our manuscript.

One minor comment:

The authors used reduced KSOM medium for blastocyst culture. Would the dormancy period be shorter if conventional KSOM medium was used? It would be helpful to our readers if the authors could describe the reasons for their choice of reduced KSOM medium.

Embryos cultured in reduced KSOM without mTORi show improved survival compared to those cultured in conventional KSOM (see Figure below). Even though this extension of survival is marginal, it indicates that removing amino acids is already beneficial for survival in the absence of mTORi, likely by reducing mTOR activity. As we aimed to investigate the effect of carnitine on pausing efficiency, and since carnitine can be generated in cells by amino acid breakdown, we decided to exclude the amino acids from the KSOM. We now explained the rationale to use reduced KSOM in the main manuscript: "As L-carnitine can be generated within the cells via amino acid breakdown, we cultured embryos in reduced KSOM, i.e. devoid of any amino acids (Table S1)."

ADDITIONAL COMMENTS ON THE POINTS RAISED BY REVIEWER #1

The request by reviewer #1 to create live animals using dormant embryonic cells was not addressed, as the embryo transfer experiments were not successful. While I certainly understand the importance of this experiment, I feel it is not necessary to investigate this part in that depth, since the major message of the study is the new metabolic pathway (FOXO1/FAO axis) involved in the maintenance of cellular dormancy, which has been sufficiently demonstrated.

The authors should address the rest of the comments by reviewer #1 as best they can, though, even if only at the text level.

We appreciate the additional feedback from reviewer #3 regarding the comments by reviewer #1. We now further addressed the comments in the text and added data illustrating the stemness of the embryonic and extraembryonic cells derived from D12+c embryos.

References

1. Bao, M., Cornwall-Scoones, J., Sanchez-Vasquez, E., Cox, A.L., Chen, D.-Y., Jonghe, J.D., Shadkhoo, S., Hollfelder, F., Thomson, M., Glover, D.M., et al. (2022). Stem cell-derived synthetic embryos self-assemble by exploiting cadherin codes and cortical tension. *Nat. Cell Biol.* 24, 1341–1349. 10.1038/s41556-022-00984-y.

2. Kamemizu, C., and Fujimori, T. (2018). Distinct dormancy progression depending on embryonic regions during mouse embryonic diapause†. *Biol Reprod* 100, 1204–1214. 10.1093/biolre/iox017.
3. Schwarzer, C., Esteves, T.C., Araúzo-Bravo, M.J., Gac, S.L., Nordhoff, V., Schlatt, S., and Boiani, M. (2012). ART culture conditions change the probability of mouse embryo gestation through defined cellular and molecular responses. *Hum. Reprod.* 27, 2627–2640. 10.1093/humrep/des223.
4. Yagi, M., Kishigami, S., Tanaka, A., Semi, K., Mizutani, E., Wakayama, S., Wakayama, T., Yamamoto, T., and Yamada, Y. (2017). Derivation of ground-state female ES cells maintaining gamete-derived DNA methylation. *Nature* 548, 224–227. 10.1038/nature23286.
5. Czechanski, A., Byers, C., Greenstein, I., Schrode, N., Donahue, L.R., Hadjantonakis, A.-K., and Reinholdt, L.G. (2014). Derivation and characterization of mouse embryonic stem cells from permissive and nonpermissive strains. *Nat. Protoc.* 9, 559–574. 10.1038/nprot.2014.030.
6. Wang, Z. (2011). Derivation of mouse embryonic stem cell lines from blastocysts produced by fertilization and somatic cell nuclear transfer. *Methods Mol. Biol. (Clifton, NJ)* 770, 529–549. 10.1007/978-1-61779-210-6_21.
7. Umehara, H., Kimura, T., Ohtsuka, S., Nakamura, T., Kitajima, K., Ikawa, M., Okabe, M., Niwa, H., and Nakano, T. (2007). Efficient Derivation of Embryonic Stem Cells by Inhibition of Glycogen Synthase Kinase-3. *STEM CELLS* 25, 2705–2711. 10.1634/stemcells.2007-0086.
8. Lee, K.-H., Chuang, C.-K., Guo, S.-F., and Tu, C.-F. (2012). Simple and Efficient Derivation of Mouse Embryonic Stem Cell Lines Using Differentiation Inhibitors or Proliferation Stimulators. *Stem Cells Dev.* 21, 373–383. 10.1089/scd.2011.0021.

Decision Letter, first revision:

Dear Aydan,

Thank you for submitting your revised manuscript "FOXO1-mediated lipid metabolism maintains mammalian embryos in dormancy" (NCB-A48823A). As I have already communicated to you via e-mail, the original reviewers re-reviewed your manuscript and, while reviewer #2 found that their concerns had been addressed, reviewer #1 continued to raise some concerns and reviewer #3 had a remaining minor point. Thank you for providing a response to these last comments of the referees and a revised manuscript to us via e-mail. We have now asked reviewer #3 (see 'ADDITIONAL COMMENTS' in their report) to comment on your response to their last minor point and cross-comment on your responses to the comments offered by reviewer #1 in the last round of review (please see all reports below). Taken everything together, we have decided that we'll be happy in principle to publish your manuscript in Nature Cell Biology, pending minor revisions to satisfy the referees' final requests (that is, except for everything you have added so far in the revised version which you e-mailed to us, to please acknowledge in the discussion of your text that the developmental potential of paused embryos in this manuscript is to be further explored in future studies) and to comply with our editorial and formatting guidelines.

If the current version of your manuscript is in a PDF format, please email us a copy of the file in an editable format (Microsoft Word or LaTeX)-- we cannot proceed with PDFs at this stage.

Thank you again for your interest in Nature Cell Biology. Please do not hesitate to contact me if you have any questions.

Best wishes,
Stelios

Stylianos Lefkopoulos, PhD
He/him/his
Senior Editor, Nature Cell Biology
Springer Nature
Heidelberger Platz 3, 14197 Berlin, Germany

E-mail: stylianos.lefkopoulos@springernature.com
Twitter: @s_lefkopoulos
LinkedIn: [linkedin.com/in/stylianos-lefkopoulos-81b007a0](https://www.linkedin.com/in/stylianos-lefkopoulos-81b007a0)

Reviewer #1 (Remarks to the Author):

In the revised manuscript, the authors have provided additional data to address my previous concerns. However, some of my concerns, particularly the major concern #3, remain unresolved (see below).

Regarding major concern point #1, the authors did a very good job when using H4K16ac as a marker to distinguish the differential response of ICM and TEs to mTORi. Since they used ESCs and TSCs as in vitro models for ICM and TE, the expression of H4K16ac should also be tested in ESCs and TSCs treated with mTORi +/- L-Carnitine. This will strengthen the rationale of using ESCs and TSCs in the current study. Also in Fig. 5 the authors should include CDX2 to mark the TE, which is critical for data quantification. DAPI alone is not sufficient in this case.

Regarding major concern point #3 to demonstrate in vivo survival of paused embryos, the author indicated that they encountered technical problems in such that they cannot reproduce their previous published results for mTORi-treated embryos (Bulut-Karslioglu et al. 2016), which is very problematic. Authors cited the previous work to indicate developmental competency of mTORi induced development diapause. Unable to reproduce this finding raises the question for the foundation of current work. The Figure showed in the rebuttal letter did not adequately indicate that the embryos can reactivate and implant efficiently. It is not shown whether there are embryos inside the decidua and whether the embryos have developed appropriately. The inability of mTORi+c embryos to efficiently give rise healthy mice (compared to mTORi alone) dramatically reduces the significance, as well as the conceptual advance of the current manuscript.

In the revision, authors argued that mTORi+L-carnitine-treated embryos can successfully derive ESCs and TSCs, comparable to mTORi-treated embryos. However, the ES derivation efficiency shown in Fig. 5G is very low (45% for control embryos), as compared to published studies for normal E3.5 embryos (often in the range of 80-90%). The ESC derivation was only 47% for 12+ embryos. Notably, the ESC derivation efficiency from mTORi-treated embryos was >50% in Bulut-Karslioglu et al. 2016. The lower efficiency of D12+c embryos to give rise to ESC does not support mTORi+L-carnitine extend during of healthy and normal embryos. In addition, the authors had overstated when saying '...that stemness of embryonic and extraembryonic cells are not affected'. Without testing respective stemness markers for derived ESC and TSC and set up different lineage differentiation assays, this statement is invalid.

Regarding my concern point #6, expression of CPT1A (Fig. S7D) is very unclear. Where is this protein expressed? For example, in c+ embryo, I cannot clearly see the CPT1 signals in the ICM (marked by SOX2), but it was shown to increase in quantification (compared to the D5 mTORi)? CPT1A localization needs to be clearly demonstrated.

Finally, the new Figure 7 showed large variations of FAO expression in different tissues. The pathway analyses can be skewed by the huge up regulation in hepatic tissues, for example, which have abnormally high FAO gene expression. It needs to be careful to draw overly broad conclusions based on RNA datasets alone.

Reviewer #2 (Remarks to the Author):

I believe that the authors have successfully addressed my previous concerns. The manuscript is clearly original, and the methods are appropriate. The results are credible. I particularly liked the new

information that the arrested embryos were capable of implantation into the uterus. I believe it to be a solid contribution to the understanding of the enigma of embryonic diapause.

Reviewer #3 (Remarks to the Author):

The authors provided thoughtful responses to the review with revisions that included new data and requested analyses. These changes made the manuscript even better and more impactful.

One minor comment:

The authors used reduced KSOM medium for blastocyst culture. Would the dormancy period be shorter if conventional KSOM medium was used? It would be helpful to our readers if the authors could describe the reasons for their choice of reduced KSOM medium.

ADDITIONAL COMMENTS

The request by reviewer #1 to create live animals using dormant embryonic cells was not addressed, as the embryo transfer experiments were not successful. While I certainly understand the importance of this experiment, I feel it is not necessary to investigate this part in that depth, since the major message of the study is the new metabolic pathway (FOXO1/FAO axis) involved in the maintenance of cellular dormancy, which has been sufficiently demonstrated. However, it I did find it important that the authors address the rest of the comments by reviewer #1 as best they can, even if only at the text level, which the authors have now done.

I think the authors responded appropriately to the reviewers' comments (including reviewer #1 and me) on the first revision.

With regard to the generation of living mice from dormant stem cells, I think the authors were able to compensate by adding data using many newly established cell lines (and also confirmed the reproducibility of previous experiments).

Through the revision process, I feel that the authors have sufficient data to support their claims.

Decision Letter, final checks:

Our ref: NCB-A48823A

27th October 2023

Dear Dr. Bulut-Karslioglu,

Thank you for your patience as we've prepared the guidelines for final submission of your Nature Cell Biology manuscript, "FOXO1-mediated lipid metabolism maintains mammalian embryos in dormancy" (NCB-A48823A). Please carefully follow the step-by-step instructions provided in the attached file, and add a response in each row of the table to indicate the changes that you have made. Please also check and comment on any additional marked-up edits we have proposed within the text. Ensuring that each point is addressed will help to ensure that your revised manuscript can be swiftly handed over to our

production team.

In recognition of the time and expertise our reviewers provide to Nature Cell Biology's editorial process, we would like to formally acknowledge their contribution to the external peer review of your manuscript entitled "FOXO1-mediated lipid metabolism maintains mammalian embryos in dormancy". For those reviewers who give their assent, we will be publishing their names alongside the published article.

Nature Cell Biology offers a Transparent Peer Review option for new original research manuscripts submitted after December 1st, 2019. As part of this initiative, we encourage our authors to support increased transparency into the peer review process by agreeing to have the reviewer comments, author rebuttal letters, and editorial decision letters published as a Supplementary item. When you submit your final files please clearly state in your cover letter whether or not you would like to participate in this initiative. Please note that failure to state your preference will result in delays in accepting your manuscript for publication.

Cover suggestions

COVER ARTWORK: We welcome submissions of artwork for consideration for our cover. For more information, please see our guide for cover artwork.

Nature Cell Biology has now transitioned to a unified Rights Collection system which will allow our Author Services team to quickly and easily collect the rights and permissions required to publish your work. Approximately 10 days after your paper is formally accepted, you will receive an email in providing you with a link to complete the grant of rights. If your paper is eligible for Open Access, our Author Services team will also be in touch regarding any additional information that may be required to arrange payment for your article.

Please note that *Nature Cell Biology* is a Transformative Journal (TJ). Authors may publish their research with us through the traditional subscription access route or make their paper immediately open access through payment of an article-processing charge (APC). Authors will not be required to make a final decision about access to their article until it has been accepted. Find out more about Transformative Journals

Authors may need to take specific actions to achieve compliance with funder and institutional open access mandates. If your research is supported by a funder that requires

immediate open access (e.g. according to Plan S principles) then you should select the gold OA route, and we will direct you to the compliant route where possible. For authors selecting the subscription publication route, the journal's standard licensing terms will need to be accepted, including self-archiving policies. Those licensing terms will supersede any other terms that the author or any third party may assert apply to any version of the manuscript.

Please use the following link for uploading these materials:
[Redacted]

Best regards,

Aimee Frier
Staff
Nature Cell Biology

On behalf of

Stylios Lefkopoulos, PhD
He/him/his
Senior Editor, Nature Cell Biology
Springer Nature
Heidelberger Platz 3, 14197 Berlin, Germany

E-mail: stylios.lefkopoulos@springernature.com
Twitter: [@s_lefkopoulos](https://twitter.com/s_lefkopoulos)
LinkedIn: [linkedin.com/in/stylios-lefkopoulos-81b007a0](https://www.linkedin.com/in/stylios-lefkopoulos-81b007a0)

Reviewer #1:

Remarks to the Author:

In the revised manuscript, the authors have provided additional data to address my previous concerns. However, some of my concerns, particularly the major concern #3, remain unresolved (see below).

Regarding major concern point #1, the authors did a very good job when using H4K16ac as a marker

to distinguish the differential response of ICM and TEs to mTORi. Since they used ESCs and TSCs as in vitro models for ICM and TE, the expression of H4K16ac should also be tested in ESCs and TSCs treated with mTORi +/- L-Carnitine. This will strengthen the rationale of using ESCs and TSCs in the current study. Also in Fig. 5 the authors should include CDX2 to mark the TE, which is critical for data quantification. DAPI alone is not sufficient in this case.

Regarding major concern point #3 to demonstrate in vivo survival of paused embryos, the author indicated that they encountered technical problems in such that they cannot reproduce their previous published results for mTORi-treated embryos (Bulut-Karslioglu et al. 2016), which is very problematic. Authors cited the previous work to indicate developmental competency of mTORi induced developmental diapause. Unable to reproduce this finding raises the question for the foundation of current work. The Figure showed in the rebuttal letter did not adequately indicate that the embryos can reactivate and implant efficiently. It is not shown whether there are embryos inside the decidua and whether the embryos have developed appropriately. The inability of mTORi+c embryos to efficiently give rise healthy mice (compared to mTORi alone) dramatically reduces the significance, as well as the conceptual advance of the current manuscript.

In the revision, authors argued that mTORi+L-carnitine-treated embryos can successfully derive ESCs and TSCs, comparable to mTORi-treated embryos. However, the ES derivation efficiency shown in Fig. 5G is very low (45% for control embryos), as compared to published studies for normal E3.5 embryos (often in the range of 80-90%). The ESC derivation was only 47% for 12+ embryos. Notably, the ESC derivation efficiency from mTORi-treated embryos was >50% in Bulut-Karslioglu et al. 2016. The lower efficiency of D12+c embryos to give rise to ESC does not support mTORi+L-carnitine extend during of healthy and normal embryos. In addition, the authors had overstated when saying '...that stemness of embryonic and extraembryonic cells are not affected'. Without testing respective stemness markers for derived ESC and TSC and set up different lineage differentiation assays, this statement is invalid.

Regarding my concern point #6, expression of CPT1A (Fig. S7D) is very unclear. Where is this protein expressed? For example, in c+ embryo, I cannot clearly see the CPT1 signals in the ICM (marked by SOX2), but it was shown to increase in quantification (compared to the D5 mTORi)? CPT1A localization needs to be clearly demonstrated.

Finally, the new Figure 7 showed large variations of FAO expression in different tissues. The pathway analyses can be skewed by the huge up regulation in hepatic tissues, for example, which have abnormally high FAO gene expression. It needs to be careful to draw overly broad conclusions based on RNA datasets alone.

Reviewer #2:

Remarks to the Author:

I believe that the authors have successfully addressed my previous concerns. The manuscript is clearly original, and the methods are appropriate. The results are credible. I particularly liked the new information that the arrested embryos were capable of implantation into the uterus. I believe it to be a solid contribution to the understanding of the enigma of embryonic diapause.

Reviewer #3:

Remarks to the Author:

The authors provided thoughtful responses to the review with revisions that included new data and requested analyses. These changes made the manuscript even better and more impactful.

One minor comment:

The authors used reduced KSOM medium for blastocyst culture. Would the dormancy period be shorter if conventional KSOM medium was used? It would be helpful to our readers if the authors could describe the reasons for their choice of reduced KSOM medium.

ADDITIONAL COMMENTS

The request by reviewer #1 to create live animals using dormant embryonic cells was not addressed, as the embryo transfer experiments were not successful. While I certainly understand the importance of this experiment, I feel it is not necessary to investigate this part in that depth, since the major message of the study is the new metabolic pathway (FOXO1/FAO axis) involved in the maintenance of cellular dormancy, which has been sufficiently demonstrated. However, it I did find it important that the authors address the rest of the comments by reviewer #1 as best they can, even if only at the text level, which the authors have now done.

I think the authors responded appropriately to the reviewers' comments (including reviewer #1 and me) on the first revision.

With regard to the generation of living mice from dormant stem cells, I think the authors were able to compensate by adding data using many newly established cell lines (and also confirmed the reproducibility of previous experiments).

Through the revision process, I feel that the authors have sufficient data to support their claims.

Author Rebuttal, first revision:

Final Decision Letter:

Dear Aydan,

I am pleased to inform you that your manuscript, "FOXO1-mediated lipid metabolism maintains mammalian embryos in dormancy", has now been accepted for publication in Nature Cell Biology. Congratulations to you and the whole team!

Please note that *Nature Cell Biology* is a Transformative Journal (TJ). Authors may publish their research with us through the traditional subscription access route or make their paper immediately open access through payment of an article-processing charge (APC). Authors will not be required to make a final decision about access to their article until it has been accepted. Find out more about

Transformative Journals

Authors may need to take specific actions to achieve compliance with funder and institutional open access mandates. If your research is supported by a funder that requires immediate open access (e.g. according to Plan S principles) then you should select the gold OA route, and we will direct you to the compliant route where possible. For authors selecting the subscription publication route, the journal's standard licensing terms will need to be accepted, including self-archiving policies. Those licensing terms will supersede any other terms that the author or any third party may assert apply to any version of the manuscript.

If you have not already done so, we strongly recommend that you upload the step-by-step protocols used in this manuscript to the Protocol Exchange (www.nature.com/protocolexchange), an open online resource established by Nature Protocols that allows researchers to share their detailed experimental know-how. All uploaded protocols are made freely available, assigned DOIs for ease of citation and are fully searchable through nature.com. Protocols and Nature Portfolio journal papers in which they are used can be linked to one another, and this link is clearly and prominently visible in the online versions of both papers. Authors who performed the specific experiments can act as primary authors for the Protocol as they will be best placed to share the methodology details, but the Corresponding Author of the present research paper should be included as one of the authors. By uploading your Protocols to Protocol Exchange, you are enabling researchers to more readily reproduce or adapt the methodology you use, as well as increasing the visibility of your protocols and papers. You can also establish a dedicated page to collect your lab Protocols. Further information can be found at www.nature.com/protocolexchange/about

With kind regards,
Stelios

Stylianos Lefkopoulos, PhD
He/him/his
Senior Editor, Nature Cell Biology
Springer Nature

Heidelberger Platz 3, 14197 Berlin, Germany

E-mail: stylianos.lefkopoulos@springernature.com

Twitter: @s_lefkopoulos

LinkedIn: [linkedin.com/in/stylianos-lefkopoulos-81b007a0](https://www.linkedin.com/in/stylianos-lefkopoulos-81b007a0)

Click here if you would like to recommend Nature Cell Biology to your librarian

<http://www.nature.com/subscriptions/recommend.html#forms>